

# Islands for entanglement negativity

**Jaydeep Kumar Basak[1], Debarshi Basu[1], Vinay Malvimat[2*],
Himanshu Parihar[1] and Gautam Sengupta [1]**

**1** Department of Physics, Indian Institute of Technology Kanpur, 208016, India
**2** Indian Institute of Science Education and Research, Homi Bhabha Rd, Pashan,
Pune 411 008, India

⋆ vinaymmp@gmail.com

## Abstract

We advance two alternative proposals for the island contributions to the entanglement negativity of various pure and mixed state configurations in quantum field theories coupled to semiclassical gravity. The first construction involves the extremization of an algebraic sum of the generalized Renyi entropies of order half. The second proposal involves the extremization of the sum of the effective entanglement negativity of quantum matter fields and the backreacted area of a cosmic brane spanning the entanglement wedge cross section which also extremizes the generalized Renyi reflected entropy of order half. These proposals are utilized to obtain the island contributions to the entanglement negativity of various pure and mixed state configurations involving the bath systems coupled to extremal and non-extremal black holes in JT gravity demonstrating an exact match with each other. Furthermore, the results from both the proposals match precisely with the island contribution to half the Renyi reflected entropy of order half providing a strong consistency check. We then allude to a possible doubly holographic picture of our island proposals and provide a derivation of the first proposal by determining the corresponding replica wormhole contributions.

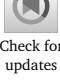
# 1  Introduction

The black hole information loss paradox is one of the most intriguing puzzles of modern theoretical physics [1–5]. This paradox arises during the process of black hole evaporation after a particular time called the Page time when the fine grained entropy of the radiation becomes greater than its coarse grained entropy. However, this is forbidden in any quantum system undergoing a unitary time evolution [6]. The recent resolution to this puzzle has revealed a novel formula for the fine-grained entropy of the Hawking radiation which involves extremization over regions in black hole spacetime known as the "*Islands*" [7–10][1], which is expressed as follows

$$S[\text{Rad}] = \min \left\{ \text{ext}_{\text{Is(Rad)}} \left[ S[\text{Rad} \cup \text{Is(Rad)}] + \frac{\text{Area}[\, \partial \, \text{Is(Rad)}\,]}{4 G_N} \right] \right\} . \tag{1}$$

---

[1]A different approach towards the resolution to the information loss puzzle has been explored in [11, 12]. According to this approach, a copy of the information inside the black hole is always available outside and hence the Page curve is trivial ( See [13] for a recent review.)

Rad in the above equation refers to the radiation which is modeled as a subsystem in the bath, Is(Rad) corresponds to the island for the radiation subsystem and $\partial$ Is(Rad) denotes its boundary. The island formula described above takes into consideration a missing saddle in Hawking's calculation arising due to spacetime "*replica wormholes*" in the gravitational path integral for the Renyi entanglement entropy, which dominates at late times during the black hole evaporation [14, 15]. The island formulation has led to a variety of fascinating directions involving exciting developments ranging from the information loss paradox in flat spacetime to puzzles in cosmology [16–49].

Although the island formula is applicable to generic spacetimes, it was inspired by the Ryu-Takayanagi (RT)/Hubeny-Rangamani-Takayanagi (HRT) formula for computing the entanglement entropy of a subsystem in a holographic $CFT_d$ [50–53], specifically its quantum corrected formula proposed in [54,55]. The proposal in [55], involves a co-dimension two surface known as the "*quantum extremal surface*" obtained by extremizing the *generalized entropy* which is defined as the sum of the area of the RT surface and the entanglement entropy of bulk quantum matter fields across the RT surface. If there are many such extremal surfaces, the one which leads the minimum generalized entropy has to be chosen. The island contribution to the generalized entropy is then computed utilizing eq.(1). The quantum extremal surface thus obtained characterizes the fine-grained entropy of the subsystems in quantum field theories, coupled to semi-classcial gravity. Recently an equivalent maximin procedure has been proposed to determine the quantum extremal surface [56]. Furthermore, the bulk domain of dependence of the HRT surface known as the "*entanglement wedge*" [57–59] played a vital role in determining the island for the Hawking radiation [9].

The fine grained entropy or the von Neumann entropy is a valid measure of the entanglement for bipartite systems in pure states, for example when we consider the bipartite system to be a black hole and the entire Hawking radiation emitted by it. Hence, the island construction for the entanglement entropy is sufficient as long as it is required to compute the Page curve or the quantities related to the bipartite entanglement between the black hole and the radiation. However, if we are interested in the structure of entanglement within the Hawking radiation, then we have to resort to mixed state entanglement or correlation measures. This is because the von Neumann entropy is neither a valid measure of the entanglement of a mixed state nor of its correlations. In the context of holography several such mixed state correlation and entanglement measures have been explored. Specifically, there is the entanglement of purification whose holographic dual was proposed as the minimal entanglement wedge cross section in [60, 61] ( See [62–65], for detailed studies of various aspects of the EWCS ). The island construction for the multipartite generalization of this quantity has been recently explored in [66]. Another significant measure in this context, is the reflected entropy proposed in [67], which is defined as the entanglement entropy of a subsystem and its copy in a canonically purified mixed state. Quite interestingly, an island construction for the reflected entropy has been proposed in [68,69]. These constructions have revealed interesting insights into the structure of correlations in within the Hawking radiation.

Note that in quantum information theory, both the entanglement of purification and the reflected entropy receive contributions from classical as well as quantum correlations. In contrast, a characteristic mixed state entanglement measure has to obey various axioms such as the monotonicity property under local operations and classical communication (LOCC) [70]. There are various such measures namely entanglement of formation, entanglement of distillation, concurrence etc. However, most of these measures are hard to compute for generic quantum states especially in extended systems such as quantum field theories. One of the computable measures which provides an upper bound on the distillable entanglement [2] is

---

[2]In quantum information theory the term "distillable entanglement" refers to the number of Bell pairs one may extract from a given quantum state utilizing only LOCC.

known as "*entanglement negativity*" proposed by Vidal and Werner in [71]. A replica technique was developed for this quantity in [72,73] and furthermore, it was utilized to compute the entanglement negativity for various mixed state configurations in two dimensional relativistic conformal field theories (CFT$_2$). A replica technique was also advanced to study the entanglement negativity in Galilean conformal field theories in [74]. The progress in [72,73], led to the interesting question of the holographic construction for the entanglement negativity, which for pure states was attempted in [75]. Following this, a holographic entanglement negativity proposal for the configuration involving a connected single interval in a zero and a finite temperature CFT$_2$ was presented in [76–78]. Furthermore, the holographic conjectures for the mixed states of the adjacent and the disjoint intervals were proposed in [79,80] and [81–83] respectively. The above mentioned holographic constructions for the entanglement negativity involved a specific algebraic sum of the areas of the co-dimension two extremal surfaces ( geodesics in AdS$_3$ ). The particular combination of which extremal surfaces appeared in the sum was determined by the mixed state in question. A plausible higher dimensional extension of the above mentioned holographic proposals and their applications to subsystems with rectangular strip like geometry were explored in [84–89].

An alternative proposal for the holographic entanglement negativity involves the minimal area of a backreacted cosmic brane on the EWCS [90]. For the specific subsystems involving spherical entangling surface, the effect of the backreaction may be determined explicitly, and the entanglement negativity in such cases is simply proportional to the area of the EWCS. For more generic scenarios, it was recently proposed in [91] that the holographic entanglement negativity is simply given by half of the Renyi reflected entropy of order half. The results from the two proposals i.e [90,91] and [76,79,81] described above, match for all the cases in the AdS$_3$/CFT$_2$ scenario except for the case of a single interval at a finite temperature. The reason for this mismatch was determined in [92] to be originating from an incorrect choice of the minimal EWCS for this configuration. Furthermore, upon computing the correct minimal EWCS, the result matches exactly with that determined from the combination of the bulk geodesics proposed in [76] and that obtained from the replica technique results in the large central charge limit in [78].

The above mentioned holographic proposals lead to the significant question concerning the island construction for the entanglement negativity. In this article, we address this extremely interesting issue by proposing two alternative constructions to determine the island contributions to the entanglement negativity of various pure and mixed state configurations in quantum field theories coupled to semiclassical gravity. The first one involves the extremization of an algebraic sum of generalized Renyi entropies of order half which is inspired by the holographic construction of [76,79,81]. Our second proposal is inspired by the quantum version of the holographic entanglement negativity construction described in [90]. This involves extremizing the sum of the area of a backreacted brane on the EWCS and the effective entanglement negativity of quantum matter fields coupled to semiclassical gravity. Furthermore, motivated by [91], we argue that the second proposal is equivalent to extremizing half the generalized Renyi reflected entropy of order half. We then apply our proposals to compute the island contributions to the entanglement negativity of several pure and mixed state configurations in non-gravitating bath systems coupled to extremal and non-extremal black holes in Jackiw-Teitelboim (JT) gravity coupled to matter described by a large-c CFT$_2$. Following the model in [8], we consider the bath to be in flat spacetime and impose transparent boundary conditions at the interface of the black hole and the bath. We compute the entanglement negativity of the pure and mixed state configurations involving two disjoint intervals, adjacent intervals, and the single interval. We demonstrate that the results from our two proposals match exactly in all the configurations considered. Furthermore, we show that results from both proposals match explicitly with the corresponding result obtained from extremizing half

the generalized Renyi reflected entropy of order half. This serves as a strong consistency check for both of our proposals[3]. Following this, we allude to a possible double holographic picture of our island proposals for the entanglement negativity. Finally, we provide a proof of our island proposal-I for the entanglement negativity of all the pure and mixed states considered in the present article, by determining the corresponding replica wormhole contribution through the techniques developed in [15, 35, 93].

The article is organized as follows: In section 2 we review the island constructions for the entanglement entropy and the reflected entropy briefly. In section 3 we propose our island constructions for the entanglement negativity of various pure and mixed state configurations. In section 4 we employ our island proposals to determine the entanglement negativity for the pure and mixed state configurations involving disjoint, adjacent and single intervals in a bath system coupled to an extremal black hole in JT gravity with the matter described by a large-$c$ CFT$_2$. In section 5 we apply our island constructions to obtain the entanglement negativity for various pure and mixed configurations in the bath system coupled to an eternal black hole in JT gravity with the matter. In section 6, we briefly describe a possible double holographic picture for our island proposals. In section 7 we provide a derivation of our island proposal-I for the entanglement negativity by considering the corresponding replica wormhole contribution. Finally, in section 8 we conclude with a summary of our results and discussions.

## 2 Review of the Island Constructions

In this section, we provide a concise review of the island construction for the entanglement entropy as described in [9] and that for the reflected entropy as developed in [68, 69].

### 2.1 Islands for the Entanglement Entropy

We begin by a brief review of the quantum Ryu-Takayanagi formula which inspired the island proposal for the entanglement entropy. The quantum corrected expression for the holographic entanglement entropy as described in [55] (as a modification to [54]) is expressed as follows

$$S(A) = \min_{X_A} \left\{ \text{ext}_{X_A} \left[ \frac{\text{Area}(X_A)}{4G_N} + S_{\text{bulk}}\left(\Sigma_{X_A}\right) \right] \right\}, \tag{2}$$

where $A$ denotes the subsystem in a holographic CFT, $X_A$ is a co-dimension 2 surface anchored to the subsystem-$A$, and $S_{\text{bulk}}\left(\Sigma_{X_A}\right)$ is the von Neumann entropy of the bulk quantum fields in the time slice of the entanglement wedge denoted as $\Sigma_{X_A}$. If there are many such extremal surfaces then the one with the minimum value has to be considered. The surface which is obtained by the above extremization followed by the minimization procedure is known as the quantum extremal surface (QES). The expression within the brackets in the above equation is known as the generalized entropy

$$S_{\text{gen}}(X) = \frac{\text{Area}(X)}{4G_N} + S_{bulk}\left(\Sigma_X\right). \tag{3}$$

Note that the definition of the generalized entropy is applicable to a co-dimension two surface ( denoted as X above ) in a generic spacetime and not restricted to holography [55].

Motivated by the quantum RT formula described above, it was proposed in [9], that the entanglement entropy of a region $A$ in a quantum field theory coupled to semiclassical gravity

---

[3]It would be very interesting to directly compute the entanglement negativity of various pure and mixed state configurations considered here, in a corresponding BCFT along the lines of [18, 25, 32, 48] for the entanglement entropy. This would serve as another strong consistency check for our island proposals and we are currently engaged in exploring this exciting issue.

is obtained by an expression analogous to the generalized entropy which is extremized over gravitational regions known as "islands"

$$S(A) = \min_{\text{Is}(A)} \left\{ \text{ext}_{\text{Is}(A)} \left[ \frac{\text{Area}[\partial \text{Is}(A)]}{4G_N} + S_{\text{eff}}[A \cup \text{Is}(A)] \right] \right\}. \tag{4}$$

Note that in the above equation Is($A$) is the island corresponding to the subsystem-$A$ and $\partial \text{Is}(A)$ denotes its boundary, "min" indicates that if there are more than one extremal surfaces one with the minimum generalized entropy is chosen, and $S_{\text{eff}}$ is the effective entanglement entropy of quantum matter fields coupled to semiclassical gravity. We emphasize here that although inspired by holography, the above expression is applicable to generic spacetimes and not restricted to asymptotically AdS gravitational configurations.

## 2.2 Islands for the Reflected Entropy

Having reviewed the island construction for entanglement entropy we now proceed to briefly describe the same for the reflected entropy. As explained in [67], the reflected entropy $S_R(A:B)$ of a bipartite system $AB$ involves the canonical purification of the given mixed state $\rho_{AB}$ by doubling its Hilbert space to define $\left| \sqrt{\rho_{AB}} \right\rangle_{ABA^*B^*}$. Note that $A^*$ and $B^*$ are the copies of the subsystems $A$ and $B$ respectively. The reflected entropy $S_R(A:B)$ is then defined as the von Neumann entropy of the subsystem $AA^*$ as follows

$$S_R(A:B) = S(AA^*). \tag{5}$$

The authors in [67], not only proposed this new measure in quantum information theory, but also developed a replica technique for it. The technique was then utilized to compute the reflected entropy of two disjoint intervals in a $\text{CFT}_2$. Furthermore, it was described in [94,95] that one may prepare a purified state $\left| \sqrt{\rho_{AB}} \right\rangle$ corresponding to a given mixed state $\rho_{AB}$ of a holographic CFT by gluing the entanglement wedge of $AB$ to the entanglement wedge of its CPT conjugate $A^*B^*$ along the RT/HRT surfaces of $AB$ and $A^*B^*$. Following this, it was described in [67], that the reflected entropy defined to be the von Neumann entropy of $AA^*$ in the state $\left| \sqrt{\rho_{AB}} \right\rangle$, is simply given by the area of RT/HRT surface of the subsystem $AA^*$ in this newly sewed manifold. It was then demonstrated that the RT/HRT surface of the subsystem $AA^*$ is twice the value of minimal EWCS. Following this, the authors of [67] developed a quantum corrected version of their proposal for a bipartite system-$AB$ in a holographic CFT which is given as follows [67]

$$S_R(A:B) = \min \text{ext}_Q \left\{ \frac{2\langle \text{Area}[Q = \partial a \cap \partial b] \rangle}{4G_N} \right\} + S_R^{\text{bulk}}(a:b)$$
$$= 2 \, \text{EWCS} + S_R^{\text{bulk}}(a:b). \tag{6}$$

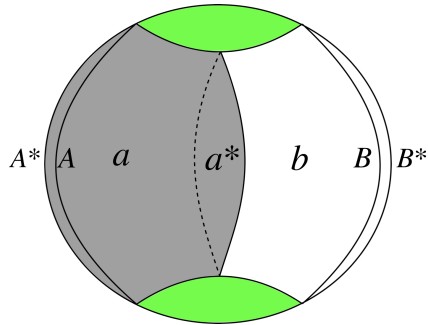

Figure 1: Schematic for the holographic construction of reflected entropy. Figure modified from [67,69]

In the above equation the leading term corresponds to twice the value of EWCS, $a$ and $b$ correspond to two bulk regions in the entanglement wedge of the subsystem $AB$ that are separated by the EWCS as depicted in fig.[1] and $S_R^{\text{bulk}}(a:b)$ denotes the effective reflected entropy of bulk matter fields in the semiclassical region $ab$.

Quite recently, an island construction for the reflected entropy was proposed in [68, 69] which may be stated as follows

$$S_R(A:B) = \min \text{ext}_{Q'} \left\{ \frac{2\text{Area}(Q' = \partial \text{Is}_R(A) \cap \partial \text{Is}_R(B))}{4G_N} \right.$$
$$\left. + S_R^{\text{eff}}(A \cup \text{Is}_R(A) : B \cup \text{Is}_R(B)) \right\}. \tag{7}$$

Observe that in the above equation $S_R^{\text{eff}}$ is the effective reflected entropy of bulk quantum matter fields on a fixed semiclassical gravitational region, $\text{Is}_R(A)$ and $\text{Is}_R(B)$ correspond to the reflected entropy islands for the subsystems $A$ and $B$ respectively. We emphasize here that $\text{Is}_R(A)$ and $\text{Is}_R(B)$ in general need not coincide with the respective entanglement entropy islands $\text{Is}(A)$ and $\text{Is}(B)$. However, they obey the condition that $\text{Is}_R(A) \cup \text{Is}_R(B) = \text{Is}(A \cup B)$.

## 3  Island Proposal for the Entanglement Negativity

In this section, we develop two alternative proposals to obtain the island contributions to the entanglement negativity for various pure and mixed state configurations in quantum field theories coupled to semiclassical gravity.

Before proceeding to describe our island constructions, let us note that entanglement negativity is defined for a bipartite system in a quantum state $\rho_{AB}$ as follows

$$\mathcal{E}(A:B) \equiv \log \left\| \rho_{AB}^{T_B} \right\|. \tag{8}$$

The Hilbert space $\mathcal{H}_{AB}$ of the bipartite system-$AB$ is assumed to be factorized as $\mathcal{H}_{AB} = \mathcal{H}_A \otimes \mathcal{H}_B$. In the above equation the superscript $T_B$ denotes the operation of partial transpose on the density matrix $\rho_{AB}$ which is defined as follows

$$\left\langle i_A, j_B \left| \rho_{AB}^{T_B} \right| k_A, l_B \right\rangle = \left\langle i_A, l_B \left| \rho_{AB} \right| k_A, j_B \right\rangle, \tag{9}$$

where $i_A, k_A$ and $j_B, l_B$ correspond to the basis states of the subsystem $A$ and $B$ respectively. Furthermore, $\left\| \rho_{AB}^{T_B} \right\|$ denotes the trace norm which is given by the absolute sum of the eigen values of the partially transposed density matrix.

### 3.1  Proposal-I: Islands for Entanglement Negativity from a Combination of Generalized Renyi Entropies

As discussed earlier, a replica technique proposed in [72, 73], was utilized to compute the entanglement negativity for various pure and mixed state configurations of a $\text{CFT}_2$. Following this, a holographic construction was advanced in [76, 78, 79] to determine the entanglement negativity of a holographic $\text{CFT}_2$ through a specific algebraic sum of the areas of extremal surfaces (lengths of geodesics in the dual bulk $\text{AdS}_3$). For example, the holographic entanglement negativity of two disjoint intervals $A$ and $B$ in proximity is given by [81]

$$\mathcal{E} = \frac{3}{16G_N} \left[ \mathcal{L}_{A \cup C} + \mathcal{L}_{B \cup C} - \mathcal{L}_{A \cup B \cup C} - \mathcal{L}_C \right] \tag{10}$$

$$= \frac{3}{4} \left[ S(A \cup C) + S(B \cup C) - S(A \cup B \cup C) - S(C) \right], \tag{11}$$

where $C$ denotes the interval sandwiched between $A$ and $B$, $\mathcal{L}_Y$ denotes the length of a geodesic anchored on the subsystem $Y$, and $G_N$ corresponds to the 3 dimensional gravitational constant. Note that in order to arrive at the last expression from the eq.(10), we have used the Ryu-Takayanagi proposal for holographic entanglement entropy which for a subsystem-$Y$ is given as [50–52]

$$S_Y = \frac{\mathcal{L}_Y}{4G_N}. \tag{12}$$

The numerical coefficient $\frac{3}{16G_N}$ in front of the area terms in eq.(10) has an important physical significance. In this context, it is crucial to recall that the holographic dual of the Renyi entropy of a subsystem-$A$ in a CFT is given by the area of a cosmic brane with a tension in the dual bulk $AdS$ spacetime [96]. This is expressed as follows

$$n^2 \frac{\partial}{\partial n} \left( \frac{n-1}{n} S^{(n)}(A) \right) = \frac{Area(\text{ cosmic brane }_n)}{4G_N}$$
$$n^2 \frac{\partial}{\partial n} \left( \frac{n-1}{n} \mathcal{A}^{(n)} \right) = Area(\text{ cosmic brane }_n), \tag{13}$$

where $S^{(n)}(A)$ is the $n^{th}$ Renyi entanglement entropy for subsystem-$A$ and the subscript $n$ the RHS indicates that the tension of the cosmic brane depends on the replica index. Note that $\mathcal{A}^{(n)}$ is related to $S^{(n)}$ as follows

$$S^{(n)} = \frac{\mathcal{A}^{(n)}}{4G_N}. \tag{14}$$

We will now utilize the following result which states that the quantity $\mathcal{A}^{(n)}$ related to the area of a back reacting cosmic brane is proportional to that of the corresponding cosmic brane with vanishing backreaction ($\mathcal{A}$) as described in [75, 90, 97]

$$\lim_{n \to 1/2} \mathcal{A}^{(n)} = \mathcal{X}_d^{hol} \mathcal{A}. \tag{15}$$

Observe that $\mathcal{X}_d$ in the above equation is a dimension dependent constant and the subscript $d$ denotes the dimension of the holographic $CFT_d$. Note that the above relation holds only for configurations involving entangling surfaces with spherical symmetry and $\mathcal{X}_d$ is explicitly known to be of the following form

$$\mathcal{X}_d = \frac{1}{2} x_d^{d-2} \left( 1 + x_d^2 \right) - 1 \tag{16}$$

$$x_d = \frac{2}{d} \left( 1 + \sqrt{1 - \frac{d}{2} + \frac{d^2}{4}} \right). \tag{17}$$

In the $AdS_3/CFT_2$ scenario this constant may be determined from the above expressions to be $\mathcal{X}_2 = \frac{3}{2}$. From the above discussion, it is clear that we may re-express the conjecture given in eq.(10) [81], as follows

$$\mathcal{E} = \frac{\mathcal{X}_2}{8G_N} \left[ \mathcal{L}_{A\cup C} + \mathcal{L}_{B\cup C} - \mathcal{L}_{A\cup B\cup C} - \mathcal{L}_C \right].$$

We now utilize the result given in eq.(15), in the $AdS_3/CFT_2$ scenario i.e $\mathcal{L}^{(1/2)} = \chi_2 \mathcal{L}$, to rewrite the above expression as follows

$$\mathcal{E} = \frac{1}{8G_N} \left[ \mathcal{L}_{A\cup C}^{(1/2)} + \mathcal{L}_{B\cup C}^{(1/2)} - \mathcal{L}_{A\cup B\cup C}^{(1/2)} - \mathcal{L}_C^{(1/2)} \right] \tag{18}$$

$$= \frac{1}{2} \left[ S^{(1/2)}(A \cup C) + S^{(1/2)}(B \cup C) - S^{(1/2)}(A \cup B \cup C) - S^{(1/2)}(C) \right], \tag{19}$$

where, $S^{(1/2)}(Y)$ in the above equation denotes the Renyi entropy of order half for the subsystem $Y$. In order to arrive at the last line of the above equation we have used eq.(14). Following the same procedure as above we may re-express the holographic conjecture for the entanglement negativity of the adjacent intervals in [79] as

$$\mathcal{E} = \frac{1}{2}\left[S^{(1/2)}(A) + S^{(1/2)}(B) - S^{(1/2)}(A \cup B)\right]. \tag{20}$$

Similarly, the holographic conjecture for the entanglement negativity of a single interval [76] may be expressed as follows

$$\mathcal{E} = \lim_{B_1 \cup B_2 \to A^c} \frac{1}{2}\left[2S^{(1/2)}(A) + S^{(1/2)}(B_1) + S^{(1/2)}(B_2) - S^{(1/2)}(A \cup B_1) - S^{(1/2)}(A \cup B_2)\right]. \tag{21}$$

Inspired by the above construction, we will propose below the island contribution to the entanglement negativity for various pure and mixed state configurations in terms of a combination of the generalized Renyi entropies of order half. However, before we discuss our island proposals, we briefly review the generalized Renyi entanglement entropy and comment on the analytic continuation to $n = \frac{1}{2}$.

### 3.1.1 Generalized Renyi Entropy

Here we provide a concise review of the island construction for the generalized Renyi entropy considered in [35]. Consider a quantum field theory coupled to gravity defined on a hybrid manifold $\mathcal{M} = \mathcal{M}^{\text{fixed}} \cup \mathcal{M}^{\text{bulk}}$, where $\mathcal{M}^{\text{fixed}}$ is non-gravitating with a fixed background metric, while $\mathcal{M}^{\text{bulk}}$ contains dynamical gravity. We assume that the quantum matter fields extend freely to the fluctuating geometry of $\mathcal{M}^{\text{bulk}}$ as well. The generalized Renyi entropy is computed through a path integral on a replica geometry $\mathcal{M}_n = \mathcal{M}_n^{\text{fixed}} \cup \mathcal{M}_n^{\text{bulk}}$ obtained by taking a branched cover of the original manifold with branch cuts along the subsystem $A$ on each copy. The Renyi entropy is then given by

$$(1-n)S_{\text{gen}}^{(n)}(A) = \log \text{Tr}\rho_A^n = \log \frac{\mathbf{Z}[\mathcal{M}_n]}{(\mathbf{Z}[\mathcal{M}_1])^n}, \tag{22}$$

where $\rho_A$ is the reduced density matrix and $\mathbf{Z}[\mathcal{M}_n]$ and $\mathbf{Z}[\mathcal{M}_1]$ corresponds to the path integral on the replicated and the original manifold respectively.

Assuming that the bulk geometry can be treated semiclassically, we can make a saddle point approximation to the gravitational path integral to write

$$\mathbf{Z}[\mathcal{M}_n] = e^{-I_{\text{grav}}[\mathcal{M}_n^{\text{bulk}}]}\mathbf{Z}_{\text{mat}}[\mathcal{M}_n], \tag{23}$$

where $I_{\text{grav}}$ corresponds to the Euclidean semiclassical gravitational action and $\mathbf{Z}_{\text{mat}}[\mathcal{M}_n]$ denotes the path integral for the quantum matter fields on the manifold $\mathcal{M}_n$.

Next we assume that the bulk geometry retains the boundary replica symmetry, and consider the theory on the quotient manifold $\tilde{\mathcal{M}}_n = \mathcal{M}_n/Z_n$. For a Hawking type saddle, the quotiented geometry $\tilde{\mathcal{M}}_n$ has conical defects on the branch cut along $A$ with deficit angle $\Delta\phi_n = 2\pi(1 - 1/n)$ sourced by a backreacted cosmic brane $\gamma_A$ homologous to the subsystem $A$ on the boundary. Supposing that the backreaction is small enough such that the replicated geometry is still a saddle to the gravitational path integral [35, 54, 55], we have:

$$I_{\text{grav}}[\mathcal{M}_n^{\text{bulk}}] \approx n\,I_{\text{grav}}[\tilde{\mathcal{M}}_1^{\text{bulk}}] + \frac{n-1}{4G_N}\mathcal{A}^{(n)}(\gamma_A), \tag{24}$$

where $\mathcal{A}^{(n)}(\gamma_A)$ is related to the area of the backreacted cosmic brane as described by eq.(13). Substituting equations (23) and (24) in eq.(22) we obtain the Renyi entropy as [35]

$$
\begin{aligned}
S_{\mathrm{gen}}^{(n)}(A) &= \frac{\mathcal{A}^{(n)}(\gamma_A)}{4G_N} + \frac{1}{1-n} \log \frac{\mathbf{Z}_{\mathrm{mat}}[\mathcal{M}_n]}{(\mathbf{Z}_{\mathrm{mat}}[\mathcal{M}_1])^n} \\
&= \frac{\mathcal{A}^{(n)}(\gamma_A)}{4G_N} + S_{\mathrm{eff}}^{(n)}(A).
\end{aligned}
\tag{25}
$$

In the above equation $S_{\mathrm{eff}}^{(n)}$ corresponds to the effective Renyi entropy of the bulk quantum matter fields. Note that in order to arrive at the last equation we simply used the definition for the Renyi entropy of the effective quantum matter fields. We emphasize here that the correct Renyi entropy is obtained by extremizing the above generalized entropy with respect to the position of the cosmic brane $\gamma_A$.

The authors in [14, 15] demonstrated that when the effective matter entropy is comparable to the gravitational entropy, another saddle arising from the spacetime replica wormhole may provide the dominant contribution to the gravitational path integral. Whenever this new saddle becomes dominant, the generalized Renyi entropy gets non-perturbative instanton-like contributions. Assuming the replica symmetry remains unbroken, the quotiented geometry corresponding to this non-trivial saddle point has no conical singularity at the boundaries of $A$. Instead, there will be additional $Z_n$ fixed points on the replica wormhole, which are the boundaries of a new region within the bulk manifold $\mathcal{M}_n^{\mathrm{bulk}}$, called the entanglement island. Therefore for the replica wormhole saddle, the analog of eq.(24) is [35]:

$$
I_{\mathrm{grav}}[\mathcal{M}_n^{\mathrm{bulk}}] \approx n\, I_{\mathrm{grav}}[\tilde{\mathcal{M}}_1^{\mathrm{bulk}}] + \frac{n-1}{4G_N} \mathcal{A}^{(n)}(\partial \mathrm{Is}(A)).
\tag{26}
$$

Once again in order to arrive at the above equation it was assumed that the backreaction is small enough to keep the replica manifold a solution to Einstein's field equations. This leads to the following form for the generalized Renyi entropy

$$
S_{\mathrm{gen}}^{(n)}(A) = \frac{\mathcal{A}^{(n)}[\partial \mathrm{Is}(A)]}{4G_N} + S_{\mathrm{eff}}^{(n)}(A \cup \mathrm{Is}(A)),
\tag{27}
$$

where $\mathrm{Is}(A)$ corresponds to the island of $A$ and $S_{\mathrm{eff}}^{(n)}$ corresponds to the effective Renyi entropy of the quantum matter fields coupled to semiclassical gravity. One consistency check of the above derivation is that it reproduces the well known island formula for the entanglement

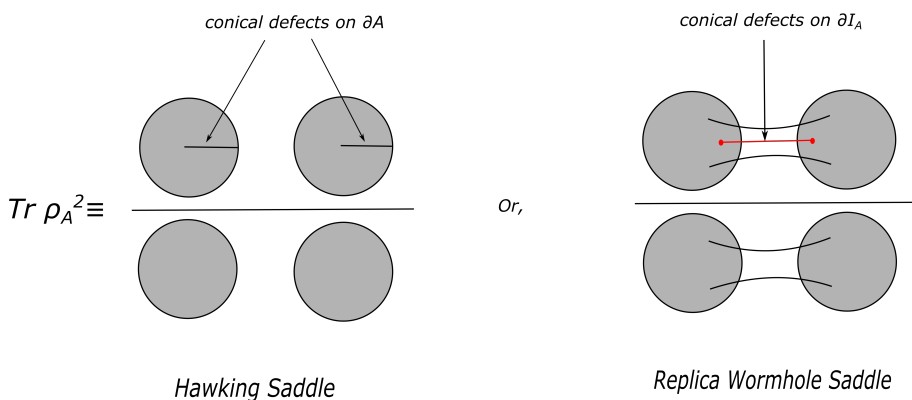

Figure 2: Schematic for the path integral representation of the replicated manifold for the generalied Renyi entropy. Figure modified from [35]

entropy in the limit $n \to 1$ [9, 35]. The analytic continuation of the above expression for $n \to \frac{1}{2}$ gives the following result for the generalized Renyi entropy of order half

$$S_{\text{gen}}^{(1/2)}(A) = \frac{\mathcal{A}^{(1/2)}[\partial \text{Is}(A)]}{4G_N} + S_{\text{eff}}^{(1/2)}(A \cup \text{Is}(A)) , \tag{28}$$

where $S_{\text{eff}}^{(1/2)}$ corresponds to the effective Renyi entropy of order half of the quantum matter fields coupled to semiclassical gravity.

**Proposal-I**

We will now develop our construction to obtain the island contributions to the entanglement negativity for various pure and mixed state configurations in terms of the generalized Renyi entropy of order half derived in eq.(28). For the case of mixed state of disjoint subsystems we propose that the island contribution to the entanglement negativity is therefore given by

$$\mathcal{E}_{\text{gen}}(A : B) = \frac{1}{2} \Big[ S_{\text{gen}}^{(1/2)}(A \cup C) + S_{\text{gen}}^{(1/2)}(B \cup C) - S_{\text{gen}}^{(1/2)}(A \cup B \cup C) - S_{\text{gen}}^{(1/2)}(C) \Big]$$
$$\mathcal{E}(A : B) = \min(\text{ext}_{Q''}\{\mathcal{E}_{\text{gen}}(A : B)\}), \tag{29}$$

where $C$ is the subsystem sandwiched between $A$ and $B$, $Q'' = \partial \text{Is}_{\mathcal{E}}(A) \cap \partial \text{Is}_{\mathcal{E}}(B)$ and $S_{\text{gen}}^{(1/2)}$ indicates the generalized Renyi entropy of order half given in eq.(28). $\text{Is}_{\mathcal{E}}(A)$ and $\text{Is}_{\mathcal{E}}(B)$ correspond to the islands for the entanglement negativity of $A$ and $B$ respectively. Note that $\text{Is}_{\mathcal{E}}(A)$ and $\text{Is}_{\mathcal{E}}(B)$ after extremization need not in general coincide with the islands for entanglement entropies of $A$ and $B$. However, as in the case of reflected entropy [68, 69], they obey the condition that $\text{Is}_{\mathcal{E}}(A) \cup \text{Is}_{\mathcal{E}}(B) = \text{Is}(A \cup B)$. We also emphasize here that $\mathcal{E}_{\text{gen}}(A : B)$ defined above naively appears to have lot more parameters than the one coming from $Q''$ but we will see that in all the cases we consider here the entanglement negativity will indeed depend only on the parameter $Q''$ which needs to be fixed by extremization as stated in our proposal above. Note that the combination of the subsystems appearing in eq.(29) is exactly same as that for the holographic entanglement negativity described by eq.(19) as proposed in [81, 82]. Note that the subsystem $C$ is an interval in a $\text{CFT}_2$. However for generic disjoint subsystems $A$ and $B$ in higher dimensions, the choice of $C$ needs more careful examination.

The result for the adjacent subsystems may be obtained by sending $C \to \emptyset$ (where $\emptyset$ denotes null set ) in eq.(29) which leads to[4]

$$\mathcal{E}_{\text{gen}}(A : B) = \frac{1}{2} \Big[ S_{\text{gen}}^{(1/2)}(A) + S_{\text{gen}}^{(1/2)}(B) - S_{\text{gen}}^{(1/2)}(A \cup B) \Big]$$
$$\mathcal{E}(A : B) = \min(\text{ext}_{Q''}\{\mathcal{E}_{\text{gen}}(A : B)\}), \tag{30}$$

where $Q'' = \partial \text{Is}_{\mathcal{E}}(A) \cap \partial \text{Is}_{\mathcal{E}}(B)$. For the case of a single connected subsystem, we propose that the generalized entanglement negativity is given by a combination of the generalized Renyi entropies of order half as follows

$$\mathcal{E}_{\text{gen}}(A : B) = \lim_{B_1 \cup B_2 \to A^c} \frac{1}{2} \Big[ 2S_{\text{gen}}^{(1/2)}(A) + S_{\text{gen}}^{(1/2)}(B_1) + S_{\text{gen}}^{(1/2)}(B_2)$$
$$- S_{\text{gen}}^{(1/2)}(A \cup B_1) - S_{\text{gen}}^{(1/2)}(A \cup B_2) \Big]$$
$$\mathcal{E}(A : B) = \min(\text{ext}_{Q''}\{\mathcal{E}_{\text{gen}}(A : B)\}), \tag{31}$$

---

[4]Note that recently in [93], the authors explicitly computed the entanglement negativity for the quantum states of a random tensor network which are toy models for a restrictive set of holographic states described as *fixed area states* which exhibit a flat entanglement spectrum. Observe that when the entanglement spectrum is flat, all the Renyi entropies corresponding to a given subsystem $X$ are simply equal $S^n(X) = S(X)$. In particular, when we consider the analytic continuation to $n = \frac{1}{2}$ which results in $S^{(1/2)}(A) = S(A)$, our proposal expressed in eq.(30) exactly reduces to the result obtained in eq.(4.14) of [93].

where $B_1$ and $B_2$ are two auxiliary subsystems on either side of $A$ and the entanglement negativity is computed in the bipartite limit $B_1 \cup B_2 \to A^c$, $Q'' = \partial \mathrm{Is}_{\mathcal{E}}(A) \cap \partial \mathrm{Is}_{\mathcal{E}}(B)$. Once again, note that this particular combination of the generalized Renyi entropies of order half is inspired by the conjecture for the holographic entanglement negativity of a single interval proposed in [76, 84] which involves a combination of entanglement entropies given by eq.(21). Observe that the auxiliary systems $B_1$ and $B_2$ are intervals in a CFT$_2$, however, for a generic subsystem-$A$ in a higher dimensional theory, the choice of $B_1$ and $B_2$ needs more careful examination.

## 3.2 Proposal-II: Islands for Entanglement Negativity from EWCS

In this subsection, we propose an alternative construction to obtain the island contribution to the entanglement negativity which is inspired by the bulk quantum corrected expression for the holographic entanglement negativity as described in [90, 91]. According to [90] the quantum corrected holographic entanglement negativity is given as follows

$$\mathcal{E}(A:B) = \min \mathrm{ext}_Q \left\{ \frac{\langle \mathcal{A}^{(1/2)}[Q = \partial a \cap \partial b] \rangle}{4G_N} \right\} + \mathcal{E}_{\mathrm{bulk}}(a:b),$$
(32)

where $\mathcal{A}^{(\frac{1}{2})}$ corresponds to the area of the backreacted cosmic brane on the EWCS and $\mathcal{E}_{\mathrm{bulk}}$ refers to the entanglement negativity of quantum matter fields in the bulk regions across EWCS. Note that this proposal is analogous to the quantum corrected holographic entanglement entropy construction proposed by Faulkner, Lewkowycz, and Maldacena in [54]. However for the island construction, we would need an analog of the Engelhardt-Wall prescription for the quantum corrected holographic entanglement entropy [55], which for the holographic entanglement negativity, we propose to be as follows

$$\mathcal{E}(A:B) = \min \mathrm{ext}_Q \left\{ \frac{\langle \mathcal{A}^{(1/2)}[Q = \partial a \cap \partial b] \rangle}{4G_N} + \mathcal{E}_{\mathrm{bulk}}(a:b) \right\},$$
(33)

where $a$ and $b$ correspond to the bulk co-dimension one regions separated by the EWCS.

We now generalize the above formula to obtain the entanglement negativity by considering the island contributions as follows

$$\mathcal{E}_{\mathrm{gen}}(A:B) = \frac{\mathcal{A}^{(1/2)}\left(Q'' = \partial \, \mathrm{Is}_{\mathcal{E}}(A) \cap \partial \, \mathrm{Is}_{\mathcal{E}}(B)\right)}{4G_N} + \mathcal{E}_{\mathrm{eff}}(A \cup \mathrm{Is}_{\mathcal{E}}(A) : B \cup \mathrm{Is}_{\mathcal{E}}(B))$$
$$\mathcal{E}(A:B) = \min(\mathrm{ext}_{Q''}\{\mathcal{E}_{\mathrm{gen}}(A:B)\}),$$
(34)

where $\mathrm{Is}_{\mathcal{E}}(A)$ and $\mathrm{Is}_{\mathcal{E}}(B)$ correspond to the islands for entanglement negativity of $A$ and $B$ respectively, obeying the condition that $\mathrm{Is}_{\mathcal{E}}(A) \cup \mathrm{Is}_{\mathcal{E}}(B) = \mathrm{Is}(A \cup B)$. Note that in general, the individual islands for the entanglement negativity $\mathrm{Is}_{\mathcal{E}}(A)$ and $\mathrm{Is}_{\mathcal{E}}(B)$ need not correspond to the islands for entanglement entropy $\mathrm{Is}(A)$ and $\mathrm{Is}(B)$ or the islands for the reflected entropy $\mathrm{Is}_R(A)$ and $\mathrm{Is}_R(B)$. However in most of the cases we consider here, we will see that they indeed match with the islands for the reflected entropy. The quantity $\mathcal{E}_{\mathrm{eff}}$ in the above equation corresponds to the effective entanglement negativity of the quantum matter fields coupled to semiclassical gravity.

In the above construction, we utilized the quantum corrected version of the holographic entanglement negativity in [90] given by eq.(33) to develop a proposal for island contribution to the entanglement negativity described by eq.(34). However, the holographic entanglement negativity proposal of [90] was concisely stated in [91] to be given by the Renyi reflected entropy of order half. A natural question then arises whether we can state our island proposal in eq.(34) more concisely in terms of the generalized Renyi reflected entropy of order half.

Before we describe this island proposal, we first review the path integral representation of the generalized Renyi reflected entropy considered in [68] and discuss its analytic continuation to $n = \frac{1}{2}$.

### 3.2.1   Generalized Renyi Reflected Entropy

We now focus on the replica construction for computing the generalized Renyi reflected entropy for a bipartite system $AB$ on a hybrid manifold $\mathcal{M} = \mathcal{M}^{\text{fixed}} \cup \mathcal{M}^{\text{bulk}}$ as before in section 3.1.1. However, for the gravity dual of the Renyi reflected entropy $S_{R\,\text{gen}}^{n,m}(A:B)$ one resorts to a more complicated replica technique than that for the Renyi entropy [54, 67]. To this end, one prepares the purifier state $|\rho_{AB}^{m/2}\rangle$ which is a replicated version of $|\sqrt{\rho_{AB}}\rangle$ discussed in section 2.2, by first performing a replication of the geometry in the index $m \in 2\mathbf{Z}$. One may then compute the Renyi entropy of $AA^*$ after performing another replication in the Renyi index $n \in \mathbf{Z}$. The gravitational path integral on the resulting replica geometry $\mathcal{M}_{m,n} = \mathcal{M}_{m,n}^{\text{fixed}} \cup \mathcal{M}_{m,n}^{\text{bulk}}$ comprising of $mn$ copies of the original geometry is then computed as follows

$$\mathbf{Z}[\mathcal{M}_{m,n}] = \text{Tr}_{AA^*}\left[\text{Tr}_{BB^*}\left(|\rho_{AB}^{m/2}\rangle\langle\rho_{AB}^{m/2}|\right)\right]^n. \tag{35}$$

We are now going to focus on the replica wormhole saddle assuming that the replica symmetry $Z_m \times Z_n$ remains unbroken. Analogous to the case of the Renyi entropy, the quotient manifold $\tilde{\mathcal{M}}_{m,n} = \mathcal{M}_{m,n}/(Z_m \times Z_n)$ has conical defects at the positions of the $m$-type and $n$-type cosmic branes corresponding to replications in different directions. There are $n$ $m$-type branes for $n$-replicas of $m$ purifier manifolds which land on the boundaries of the entanglement island $\text{Is}(A \cup B)$, and two $n$-type branes corresponding to the remnant $Z_2$ CPT symmetry which land on the cross-section of the islands [68]. Therefore, the analog of (23) in this case reads

$$\log \mathbf{Z}\big[\mathcal{M}_{m,n}\big](A \cup B) = -I_{\text{grav}}[\mathcal{M}_{m,n}^{\text{bulk}}] + \log \mathbf{Z}_{\text{mat}}[\mathcal{M}_{m,n}]$$

$$\approx -mn\, I_{\text{grav}}[\tilde{\mathcal{M}}_{1,1}^{\text{bulk}}] - n\,\frac{m-1}{4G_N}\mathcal{A}_m\big[\partial\big(\text{Is}_R(A) \cup \text{Is}_R(B)\big)\big] \tag{36}$$

$$- 2\,\frac{n-1}{4G_N}\mathcal{A}^{(n)}\big[\partial\,\text{Is}_R(A) \cap \partial\,\text{Is}_R(B)\big] + \log \mathbf{Z}_{\text{mat}}[\mathcal{M}_{m,n}].$$

This leads to the following expression for the generalized Renyi reflected entropy given by

$$\begin{aligned}
S_{R\,\text{gen}}^{n,m}(A:B) &= \frac{1}{1-n}\log\frac{\mathbf{Z}\big[\mathcal{M}_{m,n}\big]}{\big(\mathbf{Z}\big[\mathcal{M}_{m,1}\big]\big)^n} \\
&= \frac{\mathcal{A}^{(n)}(\partial\,\text{Is}_R(A) \cap \partial\,\text{Is}_R(B))}{2G_N} + \frac{1}{1-n}\log\frac{\mathbf{Z}_{\text{mat}}[\mathcal{M}_{m,n}]}{\big(\mathbf{Z}_{\text{mat}}[\mathcal{M}_{m,1}]\big)^n} \\
&= \frac{\mathcal{A}^{(n)}(\partial\,\text{Is}_R(A) \cap \partial\,\text{Is}_R(B))}{2G_N} + S_{R\,\text{eff}}^{n,m}(A \cup \text{Is}_R(A) : B \cup \text{Is}_R(B)),
\end{aligned} \tag{37}$$

which gives the island formula for the reflected entropy in [68] for $m \to 1$, $n \to 1$. Observe that in order to arrive at the last line of the above expression, we have simply utilized the definition of the effective Renyi reflected entropy $S_{R\,\text{eff}}^{n,m}$ of the quantum matter fields.

Note that the relative order of the analytic continuation in the replica indices $m$ and $n$ is important from the dual field theory side. In the large central charge limit of the $\text{CFT}_2$ the dominant channel for the conformal block could change if the order is reversed [98]. For the bulk calculations, we inherently assume that the dominant channel has already been chosen and the precise entanglement wedge is prepared first. Therefore, we choose to analytically continue $m \to 1$ first, which treats the $m$-type branes in the probe limit. Subsequently, the

analytic continuation $n \to \frac{1}{2}$ which involves the backreactions from the $n$-type branes alone, leads to the following form for the generalized Renyi reflected entropy of order half

$$S_{R\,\text{gen}}^{(1/2)}(A:B) = \frac{\mathcal{A}^{(1/2)}(\partial\text{Is}_R(A) \cap \partial\text{Is}_R(B))}{2G_N} + S_{R\,\text{eff}}^{(1/2)}(A \cup \text{Is}_R(A) : B \cup \text{Is}_R(B)). \tag{38}$$

Inspired by [91], we propose that the island contribution to the entanglement negativity of a quantum field theory coupled to semiclassical gravity is obtained by extremizing half the generalized Renyi reflected entropy of order half, which corresponds to a different analytic continuation of eq.(37) which is $m \to 1$, $n \to \frac{1}{2}$.

$$\mathcal{E}_{\text{gen}}(A:B) = \frac{S_{R\,\text{gen}}^{(1/2)}(A:B)}{2},$$
$$\mathcal{E}(A:B) = \min(\text{ext}_{Q'}\{\mathcal{E}_{\text{gen}}(A:B)\}), \tag{39}$$

where $Q' = \partial\text{Is}_R(A) \cap \partial\text{Is}_R(B)$ and $S_{R\,\text{gen}}^{1/2}(A:B)$ is the generalized Renyi reflected entropy which we obtained in eq.(38). Note that the area term in eq.(34) and eq.(39) are identical only if the islands for the reflected entropy and the entanglement negativity are exactly the same. We will see that this condition holds for the cases that we will consider in this article providing a consistency check for our proposal-II.

This concludes the description of our proposals. We now turn our attention to the application of our proposals to systems involving the baths coupled to the extremal and the non-extremal black holes in JT gravity with matter described by a large-c $\text{CFT}_2$. We will demonstrate that eq.(34) and eq.(39) give exactly the same results for the entanglement negativity in all the cases we consider in the present article, providing a consistency check for our proposal-II. Furthermore, we will demonstrate that the island contributions to the entanglement negativity obtained from proposal-I and proposal-II match precisely for all the pure and mixed states we consider. This will provide substantial evidence to our proposals.

# 4 Extremal Black Hole in JT Gravity Coupled to a Bath

## 4.1 Review of the model

Having described our proposal for the island contributions to the entanglement negativity in a quantum field theory coupled to semiclassical gravity, we now proceed to apply our conjectures to various pure and mixed state configurations in a bath coupled to an extremal black hole in JT gravity with matter described by a $\text{CFT}_2$, as considered in [10]. This model consists of a zero temperature black hole coupled to a bath described by the same $\text{CFT}_2$ as that of the quantum matter, living only on one half of two dimensional Minkowski space. The bath is coupled to the extremal black hole through transparent boundary conditions at the interface of the asymptotic boundary of $\text{AdS}_2$ and the half Minkowski space. Furthermore, we will consider the $\text{CFT}_2$ to be in its large-c limit so that we could utilize the factorizations of higher point correlation functions as considered for reflected entropy in [68]. The action for this model is given by

$$I = \frac{1}{4\pi} \int d^2x \sqrt{-g}\,[\phi R + 2(\phi - \phi_0)] + I_{\text{CFT}}, \tag{40}$$

where $\phi$ corresponds to the dilaton field and $\phi_0$ denotes a constant that contributes to the topological entropy and $I_{\text{CFT}}$ is the $\text{CFT}_2$ action of the matter coupled to JT black hole. The

metric in the Poincare coordinates and the dilaton profile are given by the following expressions

$$ds^2 = \frac{-4dx^+ dx^-}{(x^- - x^+)^2}, \quad \phi = \phi_0 + \frac{2\phi_r}{(x^- - x^+)}. \tag{41}$$

The authors in [10], obtained the following expression for the generalized entropy which characterizes the fine grained entanglement entropy of a single interval [0,b] in the bath

$$S_{\text{gen}}(a) = \phi_0 + \frac{\phi_r}{a} + S_{\text{eff}}, \quad S_{\text{eff}} = \frac{c}{6} \log\left[\frac{(a+b)^2}{a}\right] + \text{constant}. \tag{42}$$

In the above equation, the interval $[-\infty, a]$ is the island located outside the JT black hole, which corresponds to a single interval [0,b] in the bath. The end point of the island denoted as $a$ in the above equation can be found by extremizing the above expression for the generalized entropy. Note that in the above equation, the authors in [10] have utilized the units in which $4G_N = 1$. We will be using the same units for the rest of our article as well.

## 4.2 On the Computation of $S_{\text{gen}}^{(1/2)}$

Note that in order to use our proposal in eq.(29), we need a general expression for $S_{\text{gen}}^{(1/2)}$ defined by eq.(28) analogous to $S_{\text{gen}}$ given by eq.(42). As the matter $CFT_2$ is in its large-$c$ limit, $S_{\text{gen}}^{(1/2)}$ has two possibilities depending on whether the interval is large or small. Consider the configuration in which the interval denoted by $[c_1, c_2]$ is large. In this case, we obtain the generalized Renyi entropy of order half as

$$S_{\text{gen}}^{(1/2)}([c_1, c_2]) = \mathcal{A}^{(1/2)}(a(c_1)) + \mathcal{A}^{(1/2)}(a(c_2)) + S_{\text{eff}}^{(1/2)}([c_1, a(c_1)] \cup [c_2, a(c_2)])$$
$$= \mathcal{A}^{(1/2)}(a(c_1)) + \mathcal{A}^{(1/2)}(a(c_2)) + S_{\text{eff}}^{(1/2)}([c_1, a(c_1)]) + S_{\text{eff}}^{(1/2)}([c_2, a(c_2)]), \tag{43}$$

where $a(c_1)$ and $a(c_2)$ denote the end points of the island corresponding to the interval $[c_1, c_2]$. Observe that in the last step we have used the result that when the interval is large enough, the leading contribution to the four point function of twist operators characterizing $S_{\text{eff}}^{(1/2)}([c_1, a(c_1)] \cup [c_2, a(c_2)])$ factorizes into the product of two 2-point functions as described in [9]. Hence this leads to

$$S_{\text{gen}}^{(1/2)}([c_1, c_2]) = 2\phi_0 + \frac{3\phi_r}{2}\left(\frac{1}{a(c_1)} + \frac{1}{a(c_2)}\right) + \frac{c}{4}\left[\log\frac{(a(c_1) + c_1)^2}{a(c_1)\epsilon} + \log\frac{(a(c_2) + c_2)^2}{a(c_2)\epsilon}\right]. \tag{44}$$

Note that in order to arrive at the above result we have used the following result for $\mathcal{A}^{(1/2)}$ and $S_{\text{eff}}^{(1/2)}([c_1, a(c_1)]$

$$\mathcal{A}^{(1/2)}(x) = \phi_0 + \frac{3\phi_r}{2x}, \tag{45}$$

$$S_{\text{eff}}^{(1/2)}([c_1, a(c_1)] = \frac{c}{4}\left[\log\frac{(a(c_1) + c_1)^2}{a(c_1)\epsilon}\right], \tag{46}$$

where $x$ is the point in JT gravity whose area is being computed and $S_{\text{eff}}^{(1/2)}[c_1, a(c_1)]$ was obtained through the twist correlator with appropriate Weyl transformation factors taken into account as described in [10]. The justification for the above expression for $\mathcal{A}^{(1/2)}(x)$ comes from the computation of the entanglement negativity of a TFD state dual to a bulk eternal black hole in JT gravity explicitly computed in the Appendix A.1. We have demonstrated in the appendix that the topological part of $\mathcal{A}^{(1/2)}$ remains the same as that of $\mathcal{A}$. However,

the dynamical part of $\mathcal{A}^{(1/2)}$ is proportional to the corresponding dynamical part of $\mathcal{A}$ with the proportionality constant given by $\mathcal{X}_2 = \frac{3}{2}$. We assume that the same result holds for the extremal black hole case, as it is a purely geometric relation.

Consider now the configurations in which the interval $[c_1, c_2]$ is small enough to have no island contribution to its generalized Renyi entropy. Hence, in such cases the area term in eq.(28) vanishes and we get a simple result for the generalized Renyi entropy of order half, for a single interval in a $CFT_2$ which is given as follows

$$S_{\text{gen}}^{(1/2)}([c_1, c_2]) = S_{\text{eff}}^{(1/2)}([c_1, c_2]) = \frac{c}{2} \log[\frac{c_1 - c_2}{\epsilon}]. \tag{47}$$

As argued above depending on the length of the intervals $|c_1 - c_2|$, there are only two possibilities for $S_{\text{gen}}^{(1/2)}([c_1, c_2])$. This can be better understood from the double holography picture. When the interval is large, $S_{\text{gen}}^{(1/2)}$ gets an island contribution and hence becomes sum of the lengths of two backreacting cosmic branes, whereas when the interval is small, the contribution is received only from the length of a single backreacting cosmic brane.

## 4.3 Disjoint Intervals in the Bath

In this section, we compute the entanglement negativity for the mixed state of two disjoint intervals in a bath, in the model described above. Here we consider different phases by taking into account distinct possibilities for the size of the subsystems and the distance between them as in [68]. We compute the entanglement negativity for this configuration using two different methods. First one involves an algebraic sum of the generalized entropies of order half inspired by [81, 82]. The second method involves the sum of the area of a backreacted cosmic brane and the effective entanglement negativity of quantum matter fields which was inspired by [90]. We will demonstrate that the entanglement negativities computed from both proposals match exactly for all the phases. Following this we will also compute the same utilizing the island generalization of Renyi Reflected entropy of order half based on [91]. We will show that once again the result determined exactly reproduces the entanglement negativity obtained using the above mentioned proposals.

**Phase-I**

Consider the disjoint intervals $A \equiv [b_1, b_2]$ and $B \equiv [b_3, b_4]$ in the bath. In this phase, we consider $A$ and $B$ to be large enough to have a connected entanglement island described by $[a, a']$ and $[a', a'']$ respectively, with a non trivial entanglement wedge cross-section similar to [68]. However, note that in phase-I, the two intervals $A$ and $B$ are in proximity. This implies that the interval $C \equiv [b_2, b_3]$ between $A$ and $B$ is very small, and therefore, does not admit an island.

**Proposal-I**

We now utilize our proposal given in eq.(29) to compute the island contribution to entanglement negativity in phase-I described above. Utilizing the results in eq.(43) or eq.(47) depending on the size of various intervals in eq.(29), we get the following expression for the

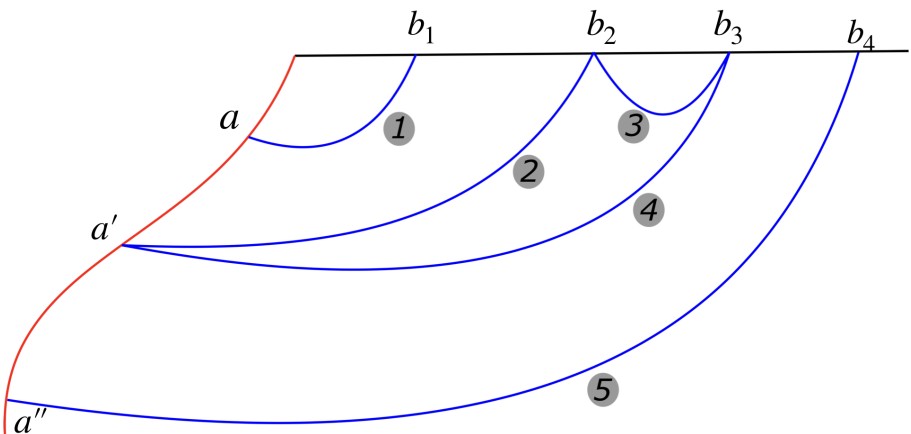

Figure 3: Schematic for the islands proposal-I for the entanglement negativity of the mixed state configuration for disjoint intervals in phase-I. The circled curves denote the backreacted cosmic branes in the double holographic picture of our proposal-I.

generalized Renyi entropies of order half

$$
\begin{aligned}
S_{\text{gen}}^{(1/2)}(A \cup C) &= \mathcal{A}^{(1/2)}(a) + \mathcal{A}^{(1/2)}(a') + S_{\text{eff}}^{(1/2)}([b_1, a]) + S_{\text{eff}}^{(1/2)}([b_3, a']), \\
S_{\text{gen}}^{(1/2)}(B \cup C) &= \mathcal{A}^{(1/2)}(a') + \mathcal{A}^{(1/2)}(a'') + S_{\text{eff}}^{(1/2)}([b_2, a']) + S_{\text{eff}}^{(1/2)}([b_4, a'']), \\
S_{\text{gen}}^{(1/2)}(A \cup B \cup C) &= \mathcal{A}^{(1/2)}(a) + \mathcal{A}^{(1/2)}(a'') + S_{\text{eff}}^{(1/2)}([b_1, a]) + S_{\text{eff}}^{(1/2)}([b_4, a'']), \\
S_{\text{gen}}^{(1/2)}(C) &= S_{\text{eff}}^{(1/2)}([b_2, b_3]) = \frac{c}{2} \log[\frac{b_2 - b_3}{\epsilon}].
\end{aligned}
\tag{48}
$$

Quite interestingly, in the double holography picture we may visualize each of the above expressions for the generalized Renyi entropies of order half to be a particular sum of the length of backreacted cosmic branes which are depicted by circled numbers in fig.[3].

We now substitute the appropriate expressions for $S_{\text{eff}}^{(1/2)}$ and $\mathcal{A}^{(1/2)}$ given by eq.(45) and (46) respectively, in eq.(48) to obtain the required generalized Renyi entropies of order half. Substituting thus obtained expressions in our proposal given by eq.(29) we obtain the generalized entanglement negativity to be as follows

$$
\mathcal{E}_{\text{gen}} = \phi_0 + \frac{3\phi_r}{2a'} + \frac{c}{4} \left[ \log(b_3 + a') + \log(b_2 + a') - \log a' - \log(b_3 - b_2) \right].
\tag{49}
$$

Note that in order to obtain the above results we utilized the fact that the subsystem $C$ is very small due to the proximity limit of $A$ and $B$. This in turn implies that the subsystem-$C$ does not admit any island. However, $A$ and $B$ are considered to be large enough to admit their respective islands, and $a'$ can be found by extremizing the above expression. The resulting equation could be solved in the limit $b_2 \to b_3$. This leads to the following

$$
2a' = b_3 + 6\frac{\phi_r}{c} + \sqrt{b_3^2 + 36\frac{\phi_r}{c}b_3 + 36\frac{\phi_r^2}{c^2}}.
\tag{50}
$$

Note that $a'$ ends on the EWCS and the above expression matches exactly with the island for reflected entropy in [68].

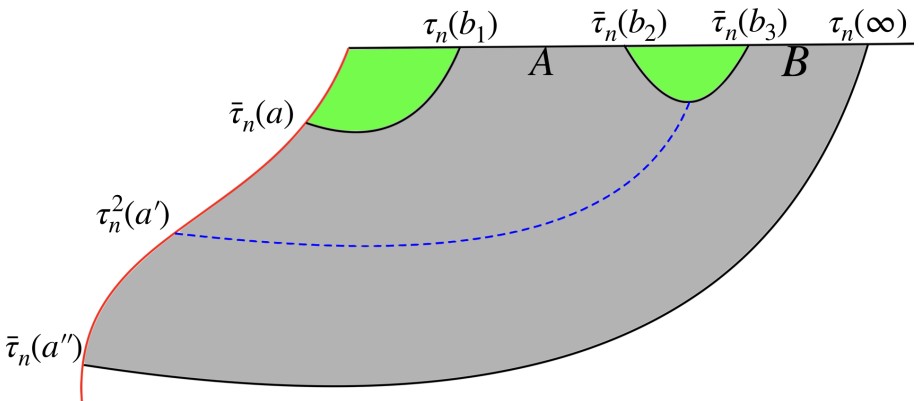

Figure 4: Schematic for island proposal-II for the entanglement negativity of the mixed state configuration of disjoint intervals in phase-I. The gray region here and for the rest of the figures corresponds to the entanglement wedge of the subsystem-$AB$. The blue dotted line denotes the EWCS in the double holographic picture. The above figure is modified from [68].

**Proposal-II**

Having computed the entanglement negativity for the disjoint interval configuration in phase-I using our island proposal-I, we now proceed to obtain the same utilizing proposal-II given in eq.(34). The area term in eq.(34) is given by

$$\mathcal{A}^{(1/2)}(a') = \phi_0 + \frac{3\phi_r}{2a'}, \tag{51}$$

where we have used eq.(45). Apart from the area term, we need to determine the effective entanglement negativity of quantum matter fields whose computation we describe below.

### $\mathcal{E}_{\text{eff}}$ through the emergent twist operators

Here we will compute the effective entanglement negativity through the replica technique of [72,73] by considering the emergent twist fields arising due to the appearance of an island. The configuration for two disjoint intervals with appropriate twist operators corresponding to the subsystem and the island, is as shown in figure below Note that in this phase the subsystem-$B$ is very large which essentially renders one of the end point of the corresponding island $a'' \to -\infty$. Since, the infinities of the bath and the JT brane are identified as in [10], the twist and the anti-twist operators located at these two infinities merge to give identity to the leading order in the OPE. This identification is depicted in the figure above by the black line.

The effective entanglement negativity $\mathcal{E}_{\text{eff}}(A \cup \text{Is}_{\mathcal{E}}(A) : B \cup \text{Is}_{\mathcal{E}}(B))$ can be written in terms of twist operators as follows

$$\mathcal{E}_{\text{eff}}(A \cup \text{Is}_{\mathcal{E}}(A) : B \cup \text{Is}_{\mathcal{E}}(B)) = \lim_{n_e \to 1} \log \left\langle \tau_{n_e}(b_1) \overline{\tau}_{n_e}(a) \overline{\tau}_{n_e}(b_2) \overline{\tau}_{n_e}(b_3) \tau_{n_e}^2(a') \right\rangle, \tag{52}$$

where $\tau_{n_e}$ denotes the twist operators described in [72,73] and $n_e$ denotes that the Renyi index is even. The limit $n_e \to 1$ has to be understood as an analytic continuation of even sequences

in $n_e$ to $n_e = 1$. The above five point correlator factorizes in the limit of large c for the channel corresponding to the phase I configuration as follows

$$
\begin{aligned}
\log\Big\langle &\tau_{n_e}(b_1)\overline{\tau}_{n_e}(a)\overline{\tau}_{n_e}(b_2)\overline{\tau}_{n_e}(b_3)\tau^2_{n_e}(a')\Big\rangle \\
&= \log\Big\langle \tau_{n_e}(b_1)\overline{\tau}_{n_e}(a)\Big\rangle\Big\langle \overline{\tau}_{n_e}(b_2)\overline{\tau}_{n_e}(b_3)\tau^2_{n_e}(a')\Big\rangle \\
&= \log\Bigg[ \Omega^{\Delta_{\tau^2_{n_e}}}(a')\frac{1}{(a+b_1)^{2\Delta_{\tau_{n_e}}}}\frac{C_{n_e}}{(b_3+a')^{\Delta_{\tau^2_{n_e}}}(b_2+a')^{\Delta_{\tau^2_{n_e}}}(b_3-b_2)^{2\Delta_{\tau_{n_e}}-\Delta_{\tau^2_{n_e}}}} \Bigg],
\end{aligned}
$$

where the dimension of twist operators $\Delta_{\tau_{n_e}}$ and $\Delta_{\tau^2_{n_e}}$ are given by [73]

$$
\Delta_{\tau_{n_e}} = \frac{c}{24}\left(n_e - \frac{1}{n_e}\right), \quad \Delta_{\tau^2_{n_e}} = \frac{c}{12}\left(\frac{n_e}{2} - \frac{2}{n_e}\right). \tag{53}
$$

We now obtain the effective entanglement negativity for the configuration of disjoint interval in phase I using eq.(52) and (53) as

$$
\mathcal{E}_{\text{eff}} = \frac{c}{4}\Big[\log(b_3+a') + \log(b_2+a') - \log a' - \log(b_3-b_2)\Big] + const. \tag{54}
$$

Note that we have also included anti-holomorphic contribution in the above equation. Having obtained the effective entanglement negativity we may now utilize eq.(51) to obtain the area term. Substituting the area term in eq.(51) and the above result for the effective entanglement negativity in eq.(34) of our island proposal-II, we obtain the generalized entanglement negativity to be

$$
\mathcal{E}_{\text{gen}} = \phi_0 + \frac{3\phi_r}{2a'} + \frac{c}{4}\Big[\log(b_3+a') + \log(b_2+a') - \log 4a' - \log(b_3-b_2)\Big]. \tag{55}
$$

Note that this equation is exactly the same as the one obtained using proposal-I given by eq.(49). Hence, on extremizing the above equation as suggested in eq.(34), we get the same expression for $a'$ as derived in eq.(50).

### $\mathcal{E}(A:B)$ through the generalized Renyi reflected entropy

As described in [68], the Renyi reflected entropy of order "n" for the phase depicted in the figure above, is given by

$$
S^{(n,m)}_{R\,\text{eff}}(A:B) = \frac{1}{1-n}\log\frac{\Big\langle \sigma_{g_A}(b_1)\sigma_{g_A^{-1}}(a)\sigma_{g_A^{-1}}(b_2)\sigma_{g_B}(b_3)\sigma_{g_Ag_B^{-1}}(a')\Big\rangle_{mn}}{\Big\langle \sigma_{g_m}(b_1)\sigma_{g_m^{-1}}(a)\sigma_{g_m^{-1}}(b_2)\sigma_{g_m}(b_3)\Big\rangle^n_m}, \tag{56}
$$

where $\sigma_{g_A}$, $\sigma_{g_{A-1}}$ and $\sigma_{g_B}$, $\sigma_{g_{B-1}}$ are the twist operators at the end points of subsystems $A$ and $B$ respectively. $\sigma_{g_Ag_B^{-1}}$ is the intermediate operator that gives the dominant contribution in the OPE of $\sigma_{g_A}$ and $\sigma_{g_B^{-1}}$ as described in [67]. The twist operator correlation functions appearing in the above equation factorize in the large central charge limit in the required phase as follows

$$
\begin{aligned}
&\Big\langle \sigma_{g_A}(b_1)\sigma_{g_A^{-1}}(a)\sigma_{g_A^{-1}}(b_2)\sigma_{g_B}(b_3)\sigma_{g_Ag_B^{-1}}(a')\Big\rangle_{mn} \\
&= \Big\langle \sigma_{g_A}(b_1)\sigma_{g_B^{-1}}(a)\Big\rangle_{mn}\Big\langle \sigma_{g_A^{-1}}(b_2)\sigma_{g_g}(b_3)\sigma_{g_Ag_B^{-1}}(a')\Big\rangle_{mn} \\
&= \Omega^{2\Delta_n}(a')\frac{1}{(a+b_1)^{2n\Delta_m}}\frac{C_{n,m}}{(b_3-b_2)^{2n\Delta_m-2\Lambda_n}(b_3+a')^{2\Delta_n}}\frac{1}{(b_2+a')^{2\Delta}}
\end{aligned} \tag{57}
$$

Also,

$$
\Big\langle \sigma_{g_m}(b_1)\sigma_{g_m^{-1}}(a)\sigma_{g_n^{-1}}(b_2)\sigma_{g_m}(b_3)\Big\rangle_m = \frac{1}{(a+b_1)^{2\Delta_m}}\frac{1}{(b_3-b_2)^{2\Lambda_m}}. \tag{58}
$$

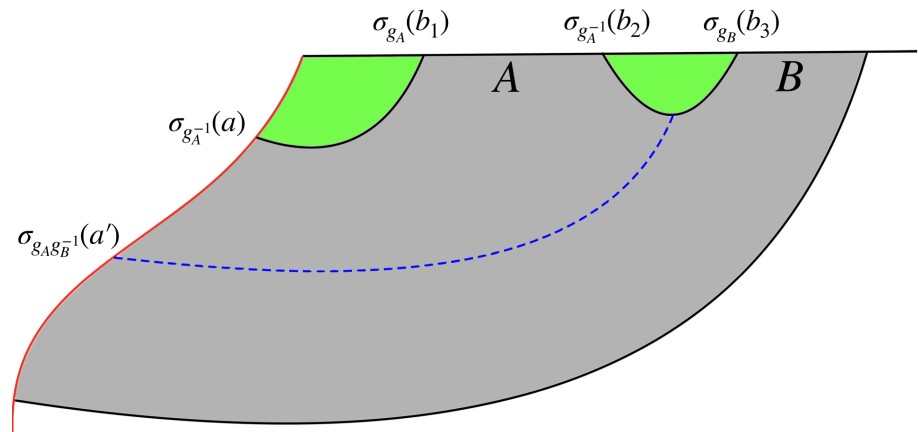

Figure 5: Schematic for the island proposal-II for the entanglement negativity of the disjoint intervals in phase-I. The emergent twist operators corresponding to the reflected entropy are depicted. Figure modified from [68]

Substituting eq.(57), eq.(58) in eq.(56) and taking the limit $m \to 1$ followed by $n \to \frac{1}{2}$, we obtain the effective Renyi reflected entropy of order half for the disjoint intervals in phase-I is given by

$$S_{R\,\text{eff}}^{(1/2)} = \frac{c}{2}\left[\log\left(b_3 + a'\right) + \log\left(b_2 + a'\right) - \log 4a' - \log\left(b_3 - b_2\right)\right]. \tag{59}$$

This gives the following expression for generalized Renyi reflected entropy of order half

$$S_{R\,\text{gen}}^{(1/2)}(A:B) = 2\left(\phi_0 + \frac{3\phi_r}{2a'}\right) + \frac{c}{4}\left[\log\left(b_3 + a'\right) + \log\left(b_2 + a'\right) - \log 4a' - \log\left(b_3 - b_2\right)\right]. \tag{60}$$

As described earlier, the islands for the reflected entropy and the entanglement negativity coincide, and hence, we have used eq.(51) for the area term in eq.(60) to arrive at the above equation. Upon utilizing the proposal described by eq.(39) along with eq.(60), we obtain the following expression for the generalized entanglement negativity

$$\mathcal{E}_{\text{gen}}(A:B) = \phi_0 + \frac{3\phi_r}{2a'} + \frac{c}{4}\left[\log\left(b_3 + a'\right) + \log\left(b_2 + a'\right) - \log 4a' - \log\left(b_3 - b_2\right)\right]. \tag{61}$$

Note that this expression precisely matches with the result obtained using our proposal given in eq.(34) for the entanglement negativity of disjoint intervals in phase-I described by eq.(55). Hence, the extremization once again leads to the same solution as in eq.(50). This demonstrates that the two equations for our proposal-II given by eq.(34) and eq.(39) for this phase exactly match as expected. Furthermore this result precisely agrees with the entanglement negativity obtained from proposal-I described by eq.(49).

**Phase-II**

We will now use the conjecture we have proposed to compute the entanglement negativity of disjoint interval in phase-II depicted in the figure above. Unlike phase-I, in this phase, the subsystem $A$ is small and hence does not have any island associated with it as was described in [68].

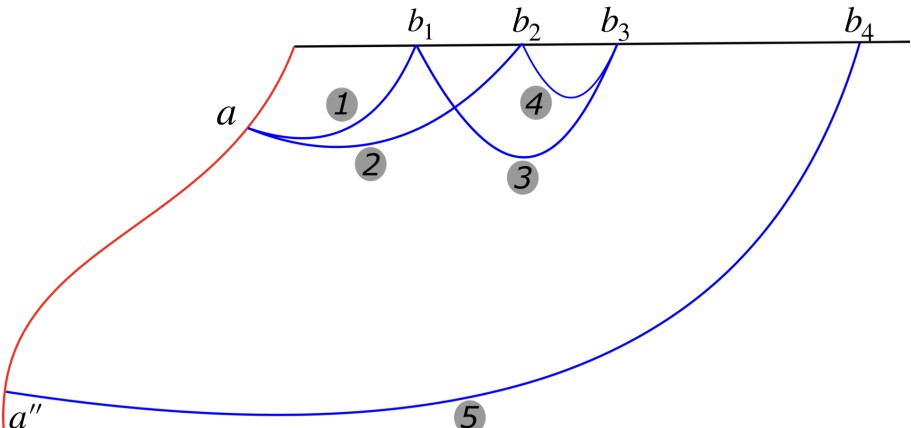

Figure 6: Schematic for the island proposal-I for the entanglement negativity of the mixed state configuration of the disjoint intervals in phase-II.

**Proposal-I**

In order to obtain the entanglement negativity using our proposal-I we first utilize the appropriate expressions for different subsystems depending on whether they are large or small as given by eq.(44) and eq.(47) respectively. Note that as the subsystem-$A$ is considered to be small in phase-II of the disjoint interval configuration. Since the subsystem-$C$ is already small due to the proximity approximation which we are using, the corresponding generalized Renyi entropy of order half for the subsystem $A \cup C$ is obtained by eq.(47). This is in contrast to the phase-I where $A \cup C$ was considered large enough to have an island. This leads to the following expressions for the required generalized Renyi entropies of order half

$$S_{\text{gen}}^{(1/2)}(A \cup C) = S_{\text{eff}}^{(1/2)}([b_1, b_2]) = \frac{c}{2} \log[\frac{b_1 - b_2}{\epsilon}]$$

$$S_{\text{gen}}^{(1/2)}(B \cup C) = \mathcal{A}^{(1/2)}(a) + \mathcal{A}^{(1/2)}(a'') + S_{\text{eff}}^{(1/2)}([b_2, a]) + S_{\text{eff}}^{(1/2)}([b_4, a''])$$

$$S_{\text{gen}}^{(1/2)}(A \cup B \cup C) = \mathcal{A}^{(1/2)}(a) + \mathcal{A}^{(1/2)}(a'') + S_{\text{eff}}^{(1/2)}([b_1, a]) + S_{\text{eff}}^{(1/2)}([b_4, a''])$$

$$S_{\text{gen}}^{(1/2)}(C) = S_{\text{eff}}^{(1/2)}([b_2, b_3]) = \frac{c}{2} \log[\frac{b_2 - b_3}{\epsilon}]. \tag{62}$$

Substituting the above expressions for various entropies in our conjecture described by eq.(29), we obtain the entanglement negativity in the proximity limit $b_2 \to b_3$ as

$$\mathcal{E}(A:B) = \frac{c}{4}\left[ \log(b_3 + a) + \log(b_3 - b_1) - \log(b_1 + a) - \log(b_3 - b_2) \right]. \tag{63}$$

Quite interestingly, the area terms in eq.(29), precisely cancel in this phase and hence do not contribute the entanglement negativity. Note that $a$ is simply determined by the island for entanglement entropy and the expression for it is once again given by eq.(50) with $b_3$ replaced by $b_1$ [10, 68]. We will see below that the vanishing of the area term has a nice physical interpretation from proposal II.

**Proposal-II**

In phase-II of the disjoint interval configuration, the area term vanishes as there is no intersection of the islands for $A$ and $B$ denoted as $Q''$ which is to be extremized over in proposal-II

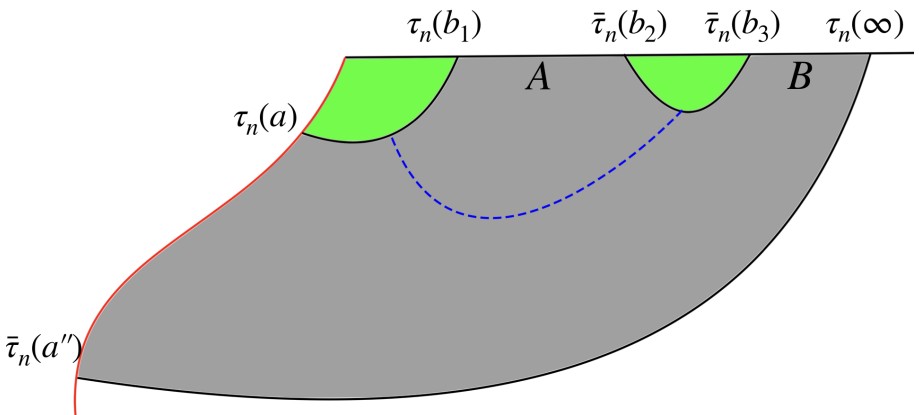

Figure 7: Schematic for the island proposal-II for the entanglement negativity for the mixed state configuration of the disjoint intervals in phase-II. Figure modified from [68].

as described by eq.(34). Since the matter $CFT_2$ is in its large-$c$ limit, $\mathcal{E}^{eff}$ in eq.(34) may be computed through the correlators of the corresponding emergent twist operators. We describe this computation below.

### $\mathcal{E}_{\text{eff}}$ through the emergent twist operators

In this phase, the subsystem-$A$ does not admit any island. The diagram for disjoint intervals in phase II is schematically shown in fig[7]. Then the effective entanglement negativity for this configuration may be written in terms of four point twist operators as

$$\mathcal{E}_{\text{eff}}(A \cup \text{Is}_\mathcal{E}(A) : B \cup \text{Is}_\mathcal{E}(B)) = \lim_{n_e \to 1} \log \left\langle \tau_{n_e}(b_1) \overline{\tau}_{n_e}(b_2) \overline{\tau}_{n_e}(b_3) \tau_{n_e}(a) \right\rangle. \tag{64}$$

Now, the effective entanglement negativity for two disjoint intervals in this phase may be calculated using the monodromy technique [81, 99], and is given by

$$\mathcal{E}_{\text{eff}}(A \cup \text{Is}_\mathcal{E}(A) : B \cup \text{Is}_\mathcal{E}(B)) = \frac{c}{4} \log\left[ \frac{(b_3 - b_1)(b_2 + a)}{(b_1 + a)(b_3 - b_2)} \right]. \tag{65}$$

Note that the area term is zero in this phase as there is no island for the subsystem $A$ in this phase as depicted in fig.[7]. Hence substituting the above expression for the effective entanglement negativity in our proposal-II described by eq.(34) we obtain the entanglement negativity to be

$$\mathcal{E}_{\text{gen}}(A : B) = \frac{c}{4} \log\left[ \frac{(b_3 - b_1)(b_2 + a)}{(b_1 + a)(b_3 - b_2)} \right]. \tag{66}$$

Observe that $a = a(b_1) \approx a(b_2)$ is simply determined by the island for entanglement entropy and the expression for it is once again given by eq.(50) with $b_3$ replaced by $b_1$ [10, 68].

### $\mathcal{E}(A : B)$ through the generalized Renyi reflected entropy

Here we will determine the effective Renyi reflected entropy of quantum matter fields for disjoint intervals in phase II and utilize it to compute the entanglement negativity through

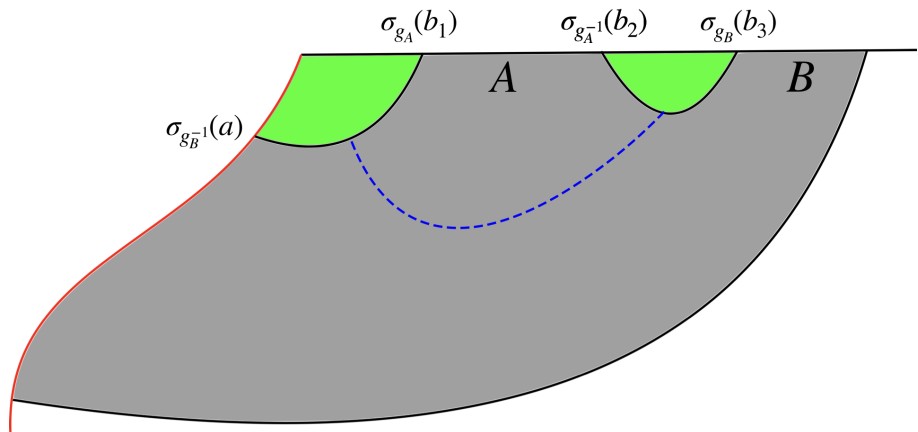

Figure 8: Schematic of the island proposal-II for the entanglement negativity of the disjoint intervals in phase-II. The emergent twist operators depicted for the reflected entropy. Figure modified from [68]

the generalized Renyi reflected entropy of order half. Following this we will demonstrate that the result obtained reproduces the entanglement negativity we computed above using our alternative island proposals.

The effective Renyi reflected entropy in this phase is given by [68]

$$S_{R\,\text{eff}}^{(n,m)}(A\cup\text{Is}_R(A):B\cup\text{Is}_R(B)) = \frac{1}{1-n}\log\frac{\left\langle\sigma_{g_B^{-1}}(a)\sigma_{g_A}(b_1)\sigma_{g_A^{-1}}(b_2)\sigma_{g_B}(b_3)\right\rangle_{mn}}{\left\langle\sigma_{g_m}(b_1)\sigma_{g_m^{-1}}(a)\sigma_{g_m^{-1}}(b_2)\sigma_{g_m}(b_3)\right\rangle_m^n}. \tag{67}$$

The conformal block $F(x,h,h_p)$ that gives the dominant contribution to the above four point function in the channel we are interested is as follows [67, 68]

$$\ln F(x,h,h_p) = -4h\log(x) + 2h_p\log\left(\frac{1+\sqrt{1-x}}{2\sqrt{x}}\right), \tag{68}$$

where $h$ is the scaling dimension of the twist operators and $h_p$ corresponds to the scaling dimension of the intermediate operator whose block gives dominant contribution to the four point function given as

$$h = \frac{cn(m^2-1)}{24m} \tag{69}$$

$$h_p = \frac{2c(n^2-1)}{24n}. \tag{70}$$

This leads to the following expression for the effective Renyi reflected entropy

$$\lim_{m\to 1}S_{R\,\text{eff}}^{(n,m)} = \frac{1}{1-n}2h_p\ln\left(\frac{1+\sqrt{1-x}}{2\sqrt{x}}\right). \tag{71}$$

Note that the order of limits of $m$ and $n$ are important in choosing the correct dominant block. However, once we choose the right block as we have done here then the order could be reversed for computational simplicity [98]. Taking the limit $n\to\frac{1}{2}$ in the above equation, we get the

following expression for the effective Renyi reflected entropy of order half in the limit $b_2 \to b_3$

$$S_{R\,\text{eff}}^{(1/2)}(A \cup \text{Is}_R(A) : B \cup \text{Is}_R(B)) = \frac{c}{2}\Bigg[ \log(b_3 + a) + \log(b_3 - b_1)$$
$$-\log(b_1 + a) - \log(b_3 - b_2) \Bigg]. \tag{72}$$

As shown in [68] there is no island for the reflected entropy corresponding to the subsystem-$A$ in this phase. This is analogous to what happens for the island corresponding to the entanglement negativity of $A$. Hence the area term in our island proposal given by eq.(39) vanishes. Therefore, we get the entanglement negativity of the bipartite system $AB$ by substituting the above equation in eq.(39) to be as follows

$$\mathcal{E}(A : B) = \frac{c}{4}\Bigg[ \log(b_3 + a) + \log(b_3 - b_1) - \log(b_1 + a) - \log(b_3 - b_2) \Bigg]. \tag{73}$$

Once again the expression for $a$ is given by eq.(50) with $b_3$ replace by $b_1$ [10, 68]. Observe that the above result matches precisely with the result we obtained using our proposal in eq.(34). Furthermore, the above expression for the entanglement negativity of disjoint interval in phase-II, matches precisely with the corresponding result obtained in eq.(63) utilizing proposal-I.

**Phase-III**

**Proposal-I**

In phase III depicted above the subsystems $C$ separating $A$ and $B$ is taken to be large. In this limit one could use the large interval result for the generalized Renyi entropy of order half given in eq.(44) for various subsystems appearing in the conjecture we have proposed in eq.(29). This leads to various cancellations leading to a vanishing result for the entanglement negativity in this phase

$$\mathcal{E}(A : B) = 0. \tag{74}$$

The double holographic picture of the above discussion is depicted in fig.[9a] and fig.[9b]. However, note that we have utilized the doubly holographic model only for the purpose of illustration and we are not performing any computations in the double holographic models in the present article.

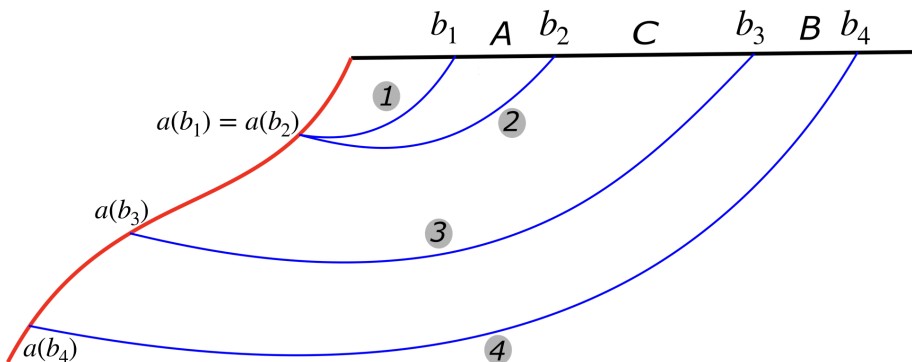

(a) Phase-III when subsystem-$A$ does not admit an island.

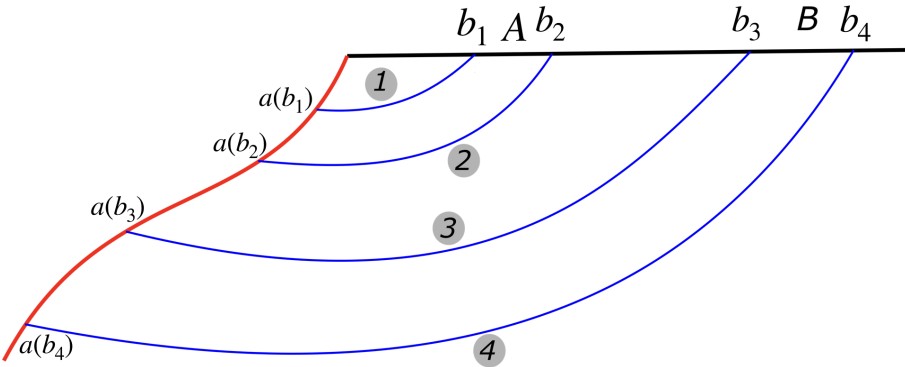

(b) Phase-III when subsystem-$A$ admits an island.

Figure 9: Schematic of island proposal-I for the entanglement negativity of the mixed state configuration of the disjoint intervals in phase-III.

**Proposal-II**

As described above in phase-III, the subsystems are separated by a large distance hence $\mathcal{E}_{\text{eff}} = 0$ and the entanglement wedge is disconnected as depicted in fig.[10a] and fig.[10b]. Hence, there is no intersection of the islands for the entanglement negativity of $A$ and $B$ and the area term in eq.(34) vanishes. Furthermore, the effective entanglement negativity is zero as the intervals are far apart from each other. This leads to the vanishing entanglement negativity $\mathcal{E}(A : B) = 0$ for this phase. This may be seen clearly from the double holography picture where the entanglement negativity is given by the area of the backreacted EWCS in the bulk, that may end on the JT brane. We will describe this in more details in section.6. Since, the entanglement wedge is disconnected in the bulk as shown in fig.[10a] and fig.[10b], it leads to $\mathcal{E}(A : B) = 0$. By similar arguments one may easily show that $S_R^{(1/2)}(A : B) = 0$. This implies that the results from two proposals once again match precisely in phase-III of the disjoint

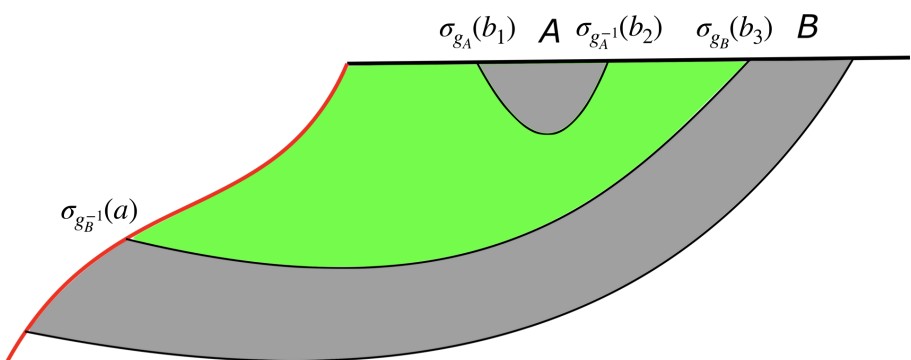

(a) Phase-III when subsystem-$A$ does not admit an island. Figure modified from [68].

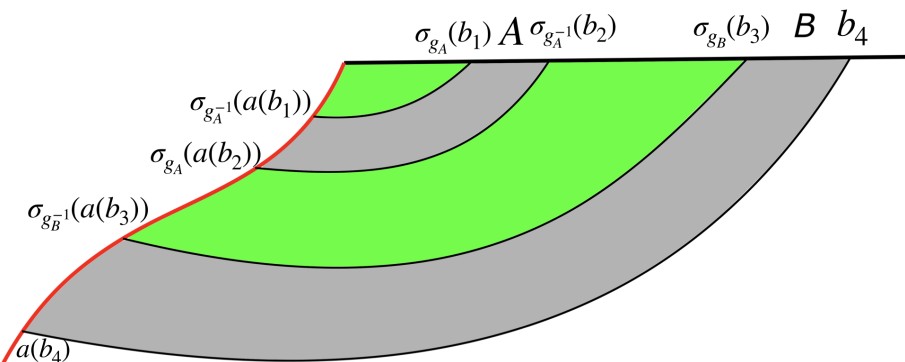

(b) Phase-III when subsystem-*A* admits an island.

Figure 10: Schematic of the island proposal-II for the entanglement negativity of the mixed state configuration of the disjoint intervals in phase-III.

interval configuration.

## 4.4 Adjacent Intervals in the Bath

In the previous subsection we computed the island contribution to the entanglement negativity for the mixed state of the disjoint intervals in the bath. In this subsection, we will determine the island contribution to the entanglement negativity for the case of adjacent intervals in the bath. We consider the configuration of adjacent intervals described by the subsystems $A \equiv [0, b_1]$ and $B \equiv [b_1, b_2]$. Note that the subsystem $A$ includes the origin. We will compute the entanglement negativity as a function of $b_1$ while keeping $b_2$ fixed. The subsystem $A$ always has an island while the existence of island for $B$ depends upon its size.

**Phase-I**

**Proposal-I**

Let us begin by computing the entanglement negativity for the adjacent intervals in phase -I, in which $b_1$ or the subsystem $A$ is small. We first utilize the appropriate expressions for the generalized Renyi entropy of order half for different subsystems depending on whether they are large or small as given by eq.(44) and eq.(47). Upon substituting thus obtained expression in our island proposal-I for the entanglement negativity of the adjacent interval case given by eq.(29), we obtain

$$\mathcal{E}_{\text{gen}}(A:B) = \phi_0 + \frac{3\phi_r}{2a(b_1)} + \frac{c}{4} \log \left[ \frac{(a(b_1) + b_1)^2}{\epsilon \, a(b_1)} \right]. \tag{75}$$

Note that in order to arrive at the above equation, we have used eq.(47) for the generalized Renyi entropy of order half corresponding to the interval $A$ as it is small and eq.(47) for the intervals $B$ and $A \cup B$ in our conjecture given in eq.(30). Furthermore $a(b_1)$ appearing in the above equation is once again determined by the entanglement island for $A$ and is obtained from eq.(50) by replacing $b_3$ with $b_1$, and $a'$ by $a(b_1)$ which leads to

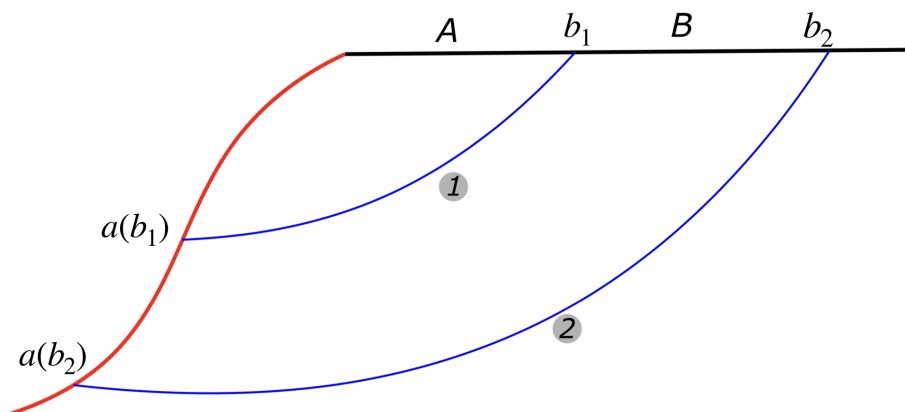

Figure 11: Schematic of the island proposal-I for the entanglement negativity of the mixed state of the adjacent intervals in phase-I

$$2a(b_1) = b_1 + 6\frac{\phi_r}{c} + \sqrt{b_1^2 + 36\frac{\phi_r}{c}b_1 + 36\frac{\phi_r^2}{c^2}}. \tag{76}$$

**Proposal-II**

After obtaining the result for the entanglement negativity of adjacent intervals in phase-I, using proposal-I, we now proceed to determine the same utilizing proposal-II. We begin by computing the effective entanglement negativity through the emergent twist operators on the JT brane. Following that we will calculate half the reflected entropy of order half through the corresponding emergent twist operators.

**$\mathcal{E}_{\text{eff}}$ through the emergent twist operators**

As described earlier in phase I, we consider $b_1$ to be small such that $B$ has an entanglement island associated with it. The required configuration with the appropriate twist operators is as shown in the above fig.[12]. The effective entanglement negativity $\mathcal{E}_{\text{eff}}(A \cup \text{Is}_{\mathcal{E}}(A) : B \cup \text{Is}_{\mathcal{E}}(B))$ for the configuration of adjacent intervals in phase I may be computed as follows

$$
\begin{aligned}
\mathcal{E}_{\text{eff}}(A \cup \text{Is}_{\mathcal{E}}(A) : B \cup \text{Is}_{\mathcal{E}}(A)) &= \lim_{n_e \to 1} \log \left\langle \tau_{n_e}^2(b_1) \overline{\tau}_{n_e}^2(a(b_1)) \right\rangle \\
&= \lim_{n_e \to 1} \log \left[ \Omega^{\Delta_{\tau_{n_e}^2}}(a(b_1)) \frac{1}{(b_1 + a(b_1))^{2\Delta_{\tau_{n_e}^2}}} \right] \\
&= \frac{c}{4} \log \left[ \frac{(a(b_1) + b_1)^2}{\epsilon \, a(b_1)} \right].
\end{aligned}
\tag{77}
$$

Note that we have also included the anti-holomorphic part in the above equation. We may now substitute the above result for the effective entanglement negativity in eq.(34) to obtain

$$\mathcal{E}(A : B) = \phi_0 + \frac{3\phi_r}{2a(b_1)} + \frac{c}{4} \log \left[ \frac{(a(b_1) + b_1)^2}{\epsilon \, a(b_1)} \right], \tag{78}$$

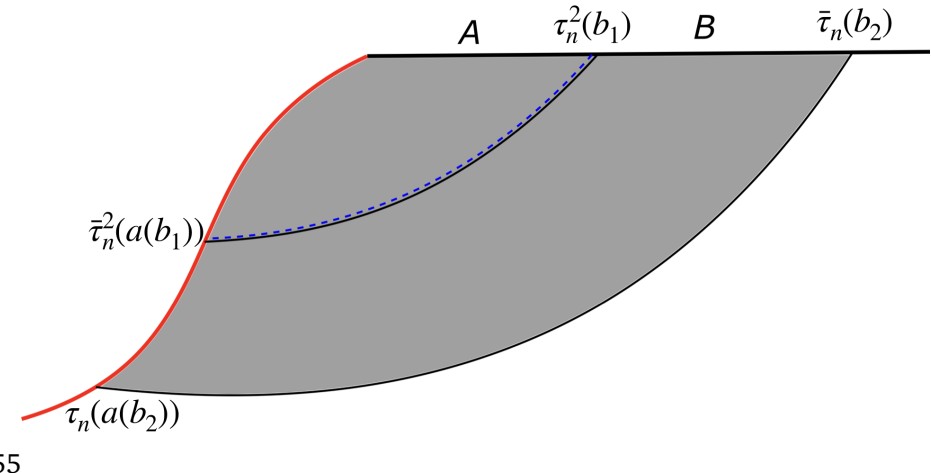

Figure 12: Schematic of the island proposal-II for the entanglement negativity of the mixed state of the adjacent intervals in phase-I. Figure modified from [68].

where, we have used the expression for the backreacted area of a point on the JT brane which we obtained by replacing $a'$ by $a(b_1)$ in eq.(51). Once again $a(b_1)$ is determined by the island for entanglement entropy given in eq.(76).

**$\mathcal{E}(A : B)$ through the generalized Renyi reflected entropy**

In this configuration the emergent twist operators are located as depicted in the figure above. The Renyi reflected entropy in this phase is given by

$$
\begin{aligned}
&S_{R\,\text{eff}}^{(n,m)}(A \cup \text{Is}_R(A) : B \cup \text{Is}_R(B)) \\
&= \frac{1}{1-n} \log \frac{\left\langle \sigma_{g_B^{-1}g_A}(a(b_1)) \sigma_{g_A^{-1}g_B}(b_1) \sigma_{g_B^{-1}}(b_2) \sigma_{g_B}(a(b_2)) \right\rangle_{mn}}{\left\langle \sigma_{g_m^{-1}}(b_2) \sigma_{g_m}(a(b_2)) \right\rangle_m^n}.
\end{aligned}
\tag{79}
$$

In the large-$c$ limit of the $\text{CFT}_2$, the above correlation function factorizes as follows

$$
\begin{aligned}
&\langle \sigma_{g_B^{-1}g_A}(a(b_1)) \sigma_{g_A^{-1}g_B}(b_1) \sigma_{g_B^{-1}}(b_2) \sigma_{g_B}(a(b_2)) \rangle_{mn} \\
&\approx \langle \sigma_{g_B^{-1}g_A}(a(b_1)) \sigma_{g_A^{-1}g_B}(b_1) \rangle \langle \sigma_{g_B^{-1}}(b_2) \sigma_{g_B}(a(b_2)) \rangle_{mn} \\
&= \frac{1}{\left(a(b_1) + b_1\right)^{4\Delta_n} \left(a(b_2) + b_2\right)^{2n\Delta_n}}
\end{aligned}
\tag{80}
$$

$$
\left\langle \sigma_{g_m^{-1}}(b_2) \sigma_{g_m}(a(b_2)) \right\rangle_m = \frac{1}{\left(a(b_2) + b_2\right)^{2\Delta_n}}.
\tag{81}
$$

Substituting the above correlations in eq.(79) we obtain the following expression for the effective Renyi reflected entropy of order half

$$
S_{R\,\text{eff}}^{(1/2)}(A \cup \text{Is}_R(A) : B \cup \text{Is}_R(B)) = \frac{c}{2} \log \left[ \frac{(a(b_1) + b_1)^2}{\epsilon\, a(b_1)} \right].
\tag{82}
$$

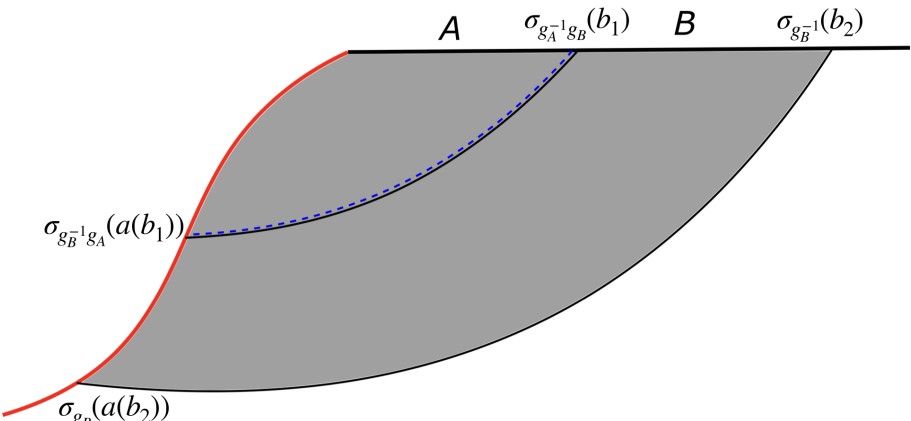

Figure 13: Schematic of the island proposal-II for the entanglement negativity of the adjacent interval configuration in phase-I. The emergent twist operators for the reflected entropy are depicted. Figure modified from [68].

Substituting the above expression for the effective Renyi reflected entropy and the area term given by eq.(51), in eq.(38) for the generalized Renyi reflected entropy we obtain

$$S_{R\,\text{gen}}^{(1/2)}(A:B) = 2(\phi_0 + \frac{3\phi_r}{2a(b_1)}) + \frac{c}{2} \log\left[\frac{(a(b_1)+b_1)^2}{\epsilon\,a(b_1)}\right]. \tag{83}$$

We may now utilize the above expression for the generalized Renyi reflected entropy of order half to compute the island contribution to the entanglement negativity of the adjacent interval in phase-I using our proposal in eq.(39) as follows

$$\mathcal{E}_{\text{gen}}(A:B) = \phi_0 + \frac{3\phi_r}{2a(b_1)} + \frac{c}{4} \log\left[\frac{(a(b_1)+b_1)^2}{\epsilon\,a(b_1)}\right], \tag{84}$$

where, $a(b_1)$ in the above equation is given by eq.(76). Observe that the above expression precisely matches with the result we obtained for the entanglement negativity in eq.(78) using our proposal in eq.(34). Furthermore, in this particular phase, the approximation $a(b_1) \approx a(b_2)$ holds and the results from the two proposals for the island contributions given by eq.(75) and eq.(84) match precisely for phase-I of the adjacent interval configuration.

## Phase-II

Having obtained the entanglement negativity for the adjacent intervals in phase-I we now now turn our attention to phase-II, where the length of the interval $A$ denoted by $b_1$ is taken to be large keeping $b_2$ fixed. This in turn reduces the size of the subsystem $B$ and hence, there is no island corresponding to it.

## Proposal-I

In order to utilize our conjecture we first notice that in this phase $b_1$ is close to $b_2$ as a result the interval $B$ is considered to be small. In this limit $a(b_1) \approx a(b_2)$. This leads to the following result for the entanglement negativity determined from our island proposal in eq.(30)

$$\mathcal{E}_{\text{gen}}(A:B) = \frac{c}{4} \log\left[\frac{(a+b_1)(b_2-b_1)}{\epsilon(a+b_2)}\right]. \tag{85}$$

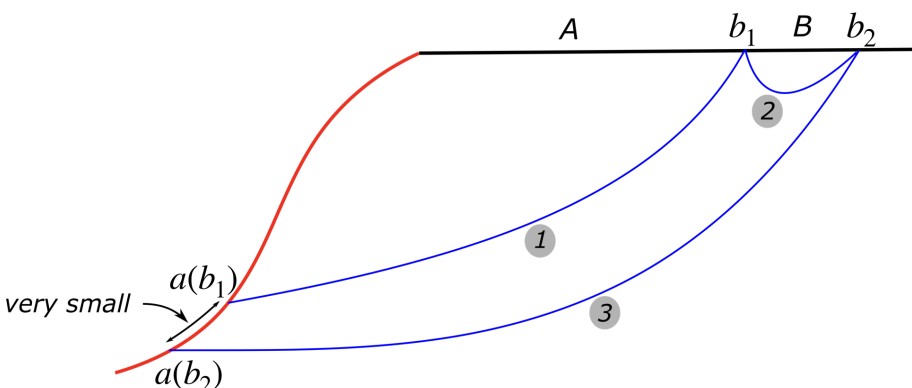

Figure 14: Schematic of the island proposal-I for the entanglement negativity of the mixed state of the adjacent intervals in phase-II

Note in this phase $a(b_1) \approx a(b_2)$ as depicted in fig.[14] and in this approximation the area term simply cancels out leading only to the effective term. Hence, $a = a(b_1) \approx a(b_2)$ in the above equation is given by eq.(76).

**Proposal-II**

Having obtained the entanglement negativity through proposal-I. We now proceed to determine the same through proposal-II. Note that the area term in the proposal-II described by eq.(34) does not given any contribution to the entanglement negativity as there is no island corresponding to $B$ in this phase. We will now compute the effective entanglement negativity contribution in this phase by considering the appropriate twist operators.

**$\mathcal{E}_{\text{eff}}$ through the emergent twist operators**

For this phase, $b_1$ is large such that the interval $B$ does not admit an island. The entire island belongs to $A$ and $a(b_1) \approx a(b_2)$. This configuration is shown in figure below. Then the effective entanglement negativity for this phase is given by

$$
\begin{aligned}
\mathcal{E}_{\text{eff}}(A \cup \text{Is}_{\mathcal{E}}(A) : B \cup \text{Is}_{\mathcal{E}}(B)) &= \lim_{n_e \to 1} \log \left\langle \tau_{n_e}^2(b_1) \overline{\tau}_{n_e}(a(b_2)) \overline{\tau}_{n_e}(b_2) \right\rangle \\
&= \frac{c}{4} \log \left[ \frac{(a(b_2) + b_1)(b_2 - b_1)}{\epsilon(a(b_2) + b_2)} \right] + const.
\end{aligned}
\tag{86}
$$

Note that in the above equation we have included the anti-holomorphic contribution. We may now compute the entanglement negativity using our island proposal-II given by eq.(34) to obtain

$$
\mathcal{E}(A : B) = \frac{c}{4} \log \left[ \frac{(a(b_2) + b_1)(b_2 - b_1)}{\epsilon(a(b_2) + b_2)} \right] + const.
\tag{87}
$$

Observe that the area term in eq.(34) did not contribute to the above result as the island for the subsystem-$B$ vanishes for this phase.

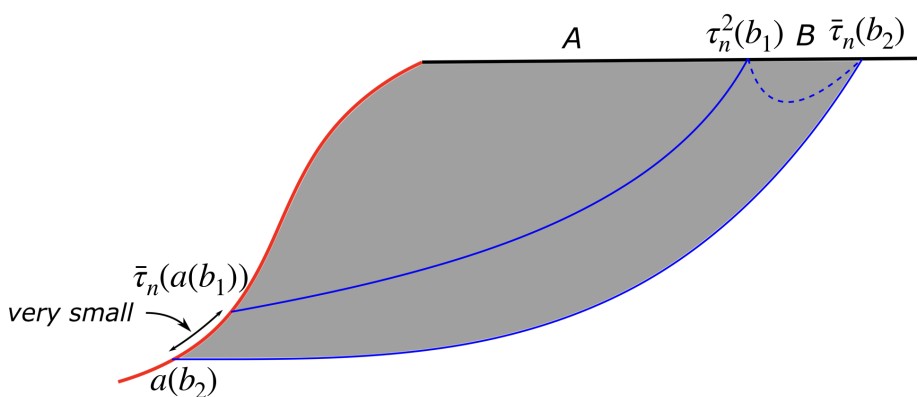

Figure 15: Schematic of the island proposal-II for the entanglement negativity of the mixed state of the adjacent intervals in phase-II. Figure modified from [68]

**$\mathcal{E}(A : B)$ through the generalized Renyi reflected entropy**

In this configuration the emergent twist operators are located as depicted in the figure above. The Renyi reflected entropy in this phase is given by

$$S_{R\,\text{eff}}^{(n,m)}(A \cup \text{Is}_R(A) : B \cup \text{Is}_R(B)) = \frac{1}{1-n} \log \frac{\left\langle \sigma_{g_A}(a(b_2))\sigma_{g_A^{-1}g_B}(b_1)\sigma_{g_B^{-1}}(b_2)\right\rangle_{mn}}{\left\langle \sigma_{g_m^{-1}}(b_2)\sigma_{g_m}(a(b_2))\right\rangle_m^n}. \tag{88}$$

The above three point function was obtained in [67] and is given as

$$\left\langle \sigma_{g_A}(a(b_2))\sigma_{g_A^{-1}g_B}(b_1)\sigma_{g_B^{-1}}(b_2)\right\rangle_{mn} = \frac{C_{n,m}}{\left(a(b_2)+b_1\right)^{2\Delta_n}\left(a(b_2)+b_2\right)^{2n\Delta_m-2\Delta_n}(b_1-b_2)^{2\Delta_n}}$$

$$\left\langle \sigma_{g_m^{-1}}(b_2)\sigma_{g_m}(a(b_2))\right\rangle_m = \frac{1}{(b_2+a(b_2))^{2\Delta_m}}. \tag{89}$$

Upon utilizing the above results for the correlation functions in eq.(88) we obtain the following expression for the effective Renyi reflected entropy of order half to be

$$S_{R\,\text{eff}}^{(1/2)}(A \cup \text{Is}_R(A) : B \cup \text{Is}_R(B)) = \frac{c}{2} \ln\left[\frac{4(b_2-b_1)(b_1+a(b_2))}{\epsilon(b_2+a(b_2))}\right], \tag{90}$$

where, we have re-introduced the UV cut-off $\epsilon$ to make the expression inside the logarithm dimensionless. Since, the area term in eq.(38) once again vanishes as the island for the subsystem is negligible we get the following expression for the entanglement negativity by substituting the above result in eq.(39).

$$\mathcal{E}(A : B) = \frac{c}{4} \ln\left[\frac{4(b_2-b_1)(b_1+a(b_2))}{\epsilon(b_2+a(b_2))}\right]. \tag{91}$$

Furthermore, notice that in this phase the area term in our proposal-II described by eq.(39) vanishes as there is no non trivial intersection of the islands for $A$ and $B$. Note that $a(b_2)$ in the above equation is obtained from eq.(76) by replacing $b_1$ with $b_2$. However as explained earlier in this phase we have $a(b_1) \approx a(b_2)$. Observe that the above result matches exactly

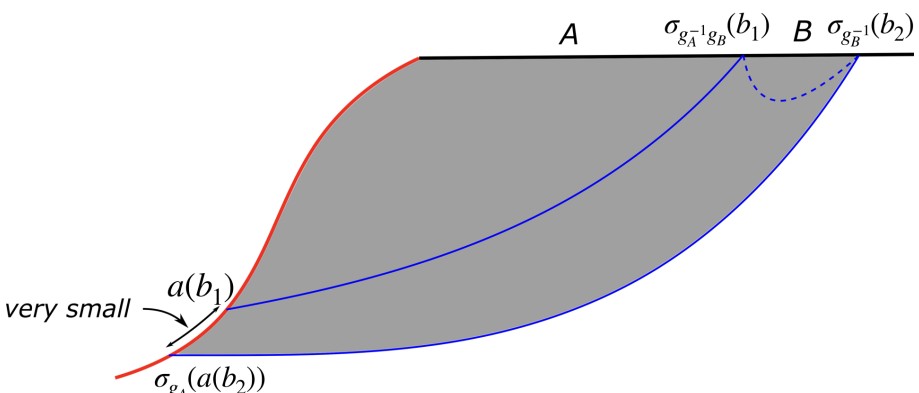

Figure 16: Schematic of the island proposal-II for the entanglement negativity of the mixed state configuration of the adjacent intervals in phase-II. The emergent twist operators required for the computation of the reflected entropy are depicted. Figure modified from [68]

with the expression for the entanglement negativity determined in eq.(87) using our proposal described by eq.(34). Furthermore, upon considering the approximation $a(b_1) \approx a(b_2)$ which is suitable for this phase, the above equation precisely matches with the result we obtained for the entanglement negativity of the adjacent interval configuration from proposal-I given by eq.(85) .

Note that the measure of reflected entropy for the configurations involving the disjoint and the adjacent intervals in various phases discussed above, was studied in [68]. The behavior of the entanglement negativity is quite similar to that of the reflected entropy for these cases. However, this is because the entangling surfaces involved have spherical symmetry and the area of the backreacted cosmic brane appearing in the expression for the entanglement negativity (see eq.(39)), is proportional to the area of the extremal surface without backreaction in the reflected entropy eq.(7). Same arguments hold for the effective terms in eq.(39) and eq.(7) as these also correspond to the area terms in the double holographic picture,. We will describe this issue in detail in section 6.2.

## 4.5 Single Interval in the Bath

**Phase-I**

Having obtained the island contributions to the entanglement negativity for the mixed state configuration of the disjoint and the adjacent intervals in the bath, we now proceed to determine the entanglement negativity of a single interval $A = [b_1, b_2]$ in the bath by considering the appropriate islands. We first describe the computations in phase-I, in which $A$ is considered to be large enough to receive contribution from the island corresponding to it in the JT spacetime. Note that if we include the boundary point which is at the interface of the JT and the bath, in the interval $B_1$ as depicted in fig.[17] below, then in the limit $B_1 \cup B_2 \rightarrow A^c$, the full system $A \cup B_1 \cup B_2$ is in a pure state. This is in contrast to the mixed state of the disjoint and the adjacent intervals we had considered previously.

**Proposal-I**

We first compute the entanglement negativity of a single interval using our proposal described by eq.(31) involving a combination of generalized Renyi entropies of order half. In order to obtain these generalized Renyi entropies for various subsystems in eq.(31), we need to examine the sizes of these intervals in question. Since in the bipartite limit $B_1 \cup B_2 \to A^c$ which describes the rest of the bath, we consider these two intervals to be large enough to admit their respective islands. Also in this phase, the interval $A$ is large, and hence it admits an island. We utilize the large interval limit of generalized Renyi entropy of order half described in eq.(44) to obtain each of the term in eq.(31). This leads to the following expression for the generalized entanglement negativity

$$\mathcal{E}_{\text{gen}}(A) = 2\phi_0 + \frac{3\phi_r}{2}\left(\frac{1}{a(b_1)} + \frac{1}{a(b_2)}\right) + \frac{c}{4}\log\left[\frac{(b_1 + a(b_1))^2 (b_2 + a(b_2))^2}{\epsilon^2 a(b_1)a(b_2)}\right] + const. \quad (92)$$

**Proposal-II**

We now turn our attention towards the entanglement negativity of a single interval in phase-I utilizing proposal-II described by eq.(34) and eq.(39). We begin by obtaining the effective entanglement negativity through the emergent twist operators which we explain below.

### $\mathcal{E}_{\text{eff}}$ through the emergent twist operators

As described above, for a single interval configuration in phase I, we take the interval $A$ to be large enough to have an island. This phase is shown in figure below along with the appropriate twist operators. Note that as depicted by light blue curve in the figure the infinities of the bath and the JT brane are identified. This in turn leads to the merging of twist and the anti-twist operators which to the leading order of the OPE expansion gives identity. This will hold for the rest of the figures in this article.

Then the effective entanglement negativity of quantum matter fields for this phase may be

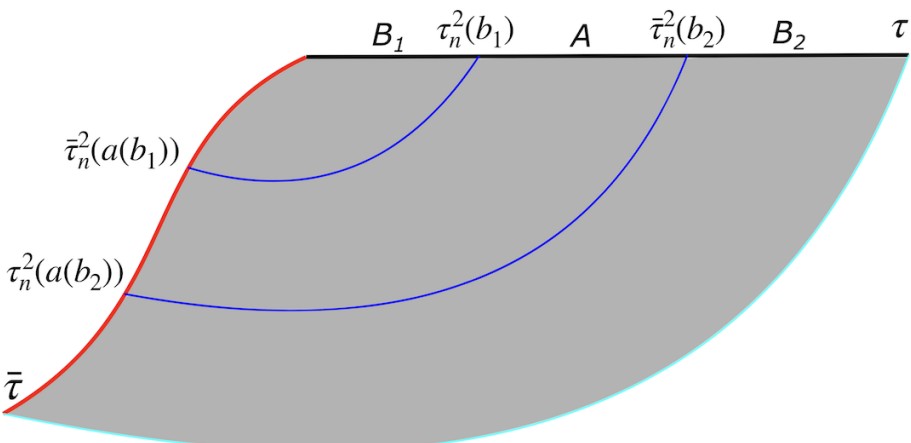

Figure 17: Schematic for the islands proposal -II for the entanglement negativity of a a single interval-*A* in phase-I.

computed as follows

$$
\begin{aligned}
\mathcal{E}_{\text{eff}}(A \cup \text{Is}_{\mathcal{E}}(A) : B \cup \text{Is}_{\mathcal{E}}(B)) &= \lim_{n_e \to 1} \log \left\langle \tau_{n_e}^2(b_1) \overline{\tau}_{n_e}^2(b_2) \tau_{n_e}^2(a(b_2)) \overline{\tau}_{n_e}^2(a(b_1)) \right\rangle \\
&= \lim_{n_e \to 1} \log \left\langle \tau_{n_e}^2(b_1) \overline{\tau}_{n_e}^2(a(b_1)) \right\rangle \left\langle \overline{\tau}_{n_e}^2(b_2) \tau_{n_e}^2(a(b_2)) \right\rangle \\
&= \frac{c}{4} \log \left[ \frac{(a(b_1) + b_1)^2}{\epsilon a(b_1)} \right] + \frac{c}{4} \log \left[ \frac{(a(b_2) + b_2)^2}{\epsilon a(b_2)} \right],
\end{aligned}
\tag{93}
$$

where we have also included the anti-holomorphic contribution. We now utilize proposal-II described by eq.(34) to obtain the entanglement negativity of a single interval in phase-I as follows

$$
\mathcal{E}(A:B) = 2\phi_0 + \frac{3\phi_r}{2} \left( \frac{1}{a(b_1)} + \frac{1}{a(b_2)} \right) + \frac{c}{4} \log \left[ \frac{(b_1 + a(b_1))^2 (b_2 + a(b_2))^2}{\epsilon^2 a(b_1) a(b_2)} \right] + const, \tag{94}
$$

where we have used the result in eq.(51) for the backreacted area term in eq.(34). We observe that in the limit $b_1 \to 0$, the entanglement negativity for single interval $A$ reduces to the entanglement negativity of adjacent intervals with the interval $B$ to be very large in phase I given by eq. (77).

### $\mathcal{E}(A:B)$ through the generalized Renyi reflected entropy

$$
S_{R\,\text{eff}}^{(n,m)}(A:B) = \frac{1}{1-n} \log \left\langle \sigma_{g_A g_{B_1}^{-1}}(b_1) \sigma_{g_{B_2} g_A^{-1}}(b_2) \sigma_{g_A^{-1} g_{B_1}}(a(b_1)) \sigma_{g_{B_2}^{-1} g_A}(a(b_2)) \right\rangle_{mn}. \tag{95}
$$

Note that unlike the earlier cases, in eq.(95) there is no denominator. This is due to the fusion of the twist and the anti-twist operators leading to the identity. We will consider $A$ to be large enough such that in the large-$c$ and large interval limit the above correlation function factorizes into

$$
\begin{aligned}
&\left\langle \sigma_{g_A g_{B_1}^{-1}}(b_1) \sigma_{g_{B_2} g_A^{-1}}(b_2) \sigma_{g_A^{-1} g_{B_1}}(a(b_1)) \sigma_{g_{B_2}^{-1} g_A}(a(b_2)) \right\rangle_{mn} \\
&\approx \langle \sigma_{g_A g_{B_1}^{-1}}(b_1) \sigma_{g_{B_2} g_A^{-1}}(b_2) \rangle_{mn} \langle \sigma_{g_A^{-1} g_{B_1}}(a(b_1)) \sigma_{g_{B_2}^{-1} g_A}(a(b_2)) \rangle_{mn}.
\end{aligned}
\tag{96}
$$

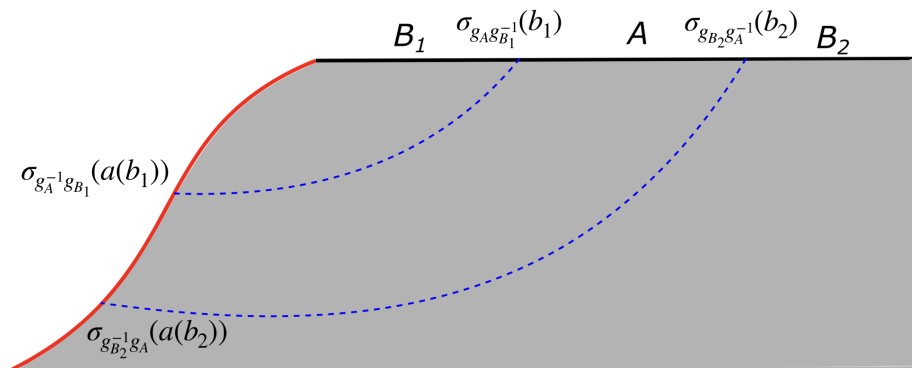

Figure 18: Schematic of the island proposal-II for the entanglement negativity of a single interval in phase-I.

Upon utilizing the above relation in eq.(95) we obtain the following expression for the Renyi reflected entropy in the limit $m \to 1$

$$S_{R\,\text{eff}}^{(n,1)}(A \cup \text{Is}_R(A) : B \cup \text{Is}_R(B)) = \frac{c(n+1)}{6n} \log\left[\frac{(b_1 + a(b_1))^2 (b_2 + a(b_2))^2}{16\epsilon^2 a(b_1)a(b_2)}\right], \qquad (97)$$

where $\epsilon$ is the UV cut-off introduced to make the argument of the log dimensionless. We may now obtain the effective Renyi reflected entropy of order half to be as follows

$$S_{R\,\text{eff}}^{(1/2)}(A \cup \text{Is}_R(A) : B \cup \text{Is}_R(B)) = \frac{c}{2} \log\left[\frac{(b_1 + a(b_1))^2 (b_2 + a(b_2))^2}{16\epsilon^2 a(b_1)a(b_2)}\right]. \qquad (98)$$

Upon using the above expression and the result for the backreacted area given by eq.(51), in eq.(38) we may obtain the generalized Renyi reflected entropy of order half to be as follows

$$S_{R\,\text{gen}}^{(1/2)}(A : B) = 4\phi_0 + 3\phi_r\left(\frac{1}{a(b_1)} + \frac{1}{a(b_2)}\right) + \frac{c}{2} \log\left[\frac{(b_1 + a(b_1))^2 (b_2 + a(b_2))^2}{16\epsilon^2 a(b_1)a(b_2)}\right]. \qquad (99)$$

Having obtained the generalized Renyi reflected entropy of order half, we may now express the entanglement negativity according to proposal-II described by eq.(39) as follows

$$\mathcal{E}_{\text{gen}} = 2\phi_0 + \frac{3\phi_r}{2}\left(\frac{1}{a(b_1)} + \frac{1}{a(b_2)}\right) + \frac{c}{4} \log\left[\frac{(b_1 + a(b_1))^2 (b_2 + a(b_2))^2}{16\epsilon^2 a(b_1)a(b_2)}\right]. \qquad (100)$$

Note that the above result matches precisely with eq.(94) for the entanglement negativity obtained using our proposal described by eq.(34). Furthermore, it is also to be observed that the above result precisely matches with eq.(92) which was determined utilizing proposal-I.

As described earlier, the quantum system of single interval-$A$ with its complement described by the rest of the system, forms a pure quantum state. One could easily understand this idea from the one dimensional point of view where the entire JT brane is replaced by its dual quantum mechanical $CFT_1$ coupled to the half line described by the bath $\text{CFT}_2$. In this case one expects that the entanglement negativity is given by the Renyi entropy of order half. To demonstrate this first observe that the generalized entanglement negativity we have obtained in eq.(100) is

$$\mathcal{E}_{\text{gen}}(A : B) = S_{\text{gen}}^{(1/2)}(A), \qquad (101)$$

where we have used the definition for the generalized Renyi entropy of order half given in eq.(28) to re-express the RHS of eq.(100). This in turn implies that upon extremization we obtain

$$\mathcal{E}(A : B) = S^{(1/2)}(A). \qquad (102)$$

For a quantum system in pure state, the above relation is expected to hold from quantum information theory as was demonstrated in [72, 73]. This serves as a strong consistency check for the result we have obtained.

**Phase-II**

In this subsection we compute the entanglement negativity of the single interval $A = [b_1, b_2]$ in the bath. We obtain the entanglement negativity when the length of the interval $A$ is small and hence it does not admit an island corresponding to it. We term this configuration as phase-II.

**Proposal-I**

We begin by considering the proposal-I for single interval described by eq.(31). As described above in phase-II length of the interval $A$ is small and hence the generalized Renyi entropy of order half corresponding to it is given by eq.(47) where as for the rest of the subsystems in eq.(31) we utilize eq.(44). In this approximation, the entanglement negativity of single interval comes out to be

$$\mathcal{E}(A) = \frac{c}{2} \log\left[\frac{b_2 - b_1}{\epsilon}\right]. \tag{103}$$

Once again since this is a result with no island contribution it is identical to the expression obtained in [72, 73].

**Proposal-II**

We will now proceed to compute the entanglement negativity of a single interval in phase-II using proposal-II.

### $\mathcal{E}_{\text{eff}}$ through the emergent twist operators

As explained above in this phase, we take $A$ to be small such that it does not admit an island. The figure for this phase is depicted below. Then the effective entanglement negativity is given by the two point twist correlators as

$$\begin{aligned}
\mathcal{E}_{\text{eff}} &= \lim_{n_e \to 1} \log\left\langle \tau_{n_e}^2(b_1) \overline{\tau}_{n_e}^2(b_2) \right\rangle \\
&= \frac{c}{2} \log\left[\frac{b_2 - b_1}{\epsilon}\right],
\end{aligned} \tag{104}$$

where the anti-holomophic contribution is also included. Notice that as there is no island corresponding to $A$ and hence, the area term in eq.(34) vanishes. Upon substituting the above expression for the effective entanglement negativity in eq.(34) for our proposal-II, we obtain

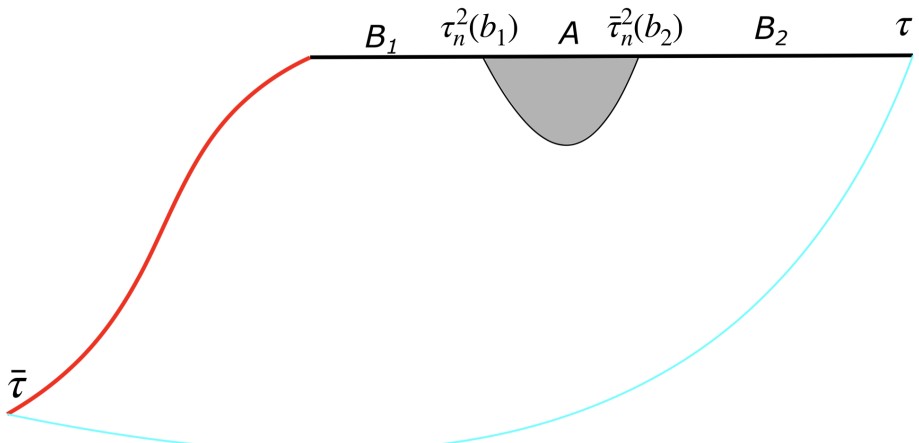

Figure 19: Schematic of the island proposal-II for the entanglement negativity of a single interval in phase-II.

the entanglement negativity to be

$$\mathcal{E}(A) = \frac{c}{2} \log\left[\frac{b_2 - b_1}{\epsilon}\right].$$

(105)

Observe that once again the result we obtained using our proposal-I which is given by eq.(103) matches precisely with the above expression obtained using proposal-II. Furthermore as can be easily checked the RHS of the above equation is simply the Renyi entropy of order half for a single interval for the no-island scenario as expected from quantum information theory [72, 73].

# 5 Eternal Black Hole in JT Gravity Coupled to a Bath

In this section, we proceed to apply our island proposals to determine the entanglement negativity of various mixed state configurations in a bath coupled to an eternal black hole solution in Jackiw-Teitelboim gravity with matter fields. The bath is described by matter fields on a separate rigid manifold, characterized by a two-dimensional conformal field theory [10]. In addition we will also assume that the $CFT_2$ is endowed with a large central charge, and we will utilize the large-c factorization of higher point twist correlators.

## 5.1 Review of the model

The model first considered in [10] consists of JT gravity living on a $AdS_2$ region, sewed together with two rigid Minkowski regions on each side which we refer to as the baths. In addition we consider a large-c $CFT_2$ living on the whole manifold which can pass freely through the AdS boundaries on which transparent boundary conditions are imposed. The action for two-dimensional Jackiw-Teitelboim (JT) gravity reads [10] (we set $4G_N = 1$)

$$I = -\frac{\phi_0}{4\pi}\left(\int_\Sigma R + 2\int_{\partial\Sigma} K\right) - \frac{1}{4\pi}\int_\Sigma \phi(R+2) - \frac{\phi_b}{4\pi}\int_{\partial\Sigma} 2K + I_{\text{CFT}},$$

(106)

where $\phi$ is the dilaton field, $\phi_b$ is its boundary value, $\Sigma$ denotes the $AdS_2$ region and $K$ is the trace of the extrinsic curvature. The term within the parenthesis is the usual Einstein-Hilbert action endowed with the proper boundary term which, in two dimensions, is topological and therefore $\phi_0$ measures the topological entropy. We focus on the two sided eternal black hole solution with the dilaton. The Penrose diagram of the model is shown in fig.[20].

Following [9, 68, 69], we may write down the metric and dilaton profiles for the black hole exteriors. We will choose two different coordinate charts to describe the geometry, namely the global coordinates $(w^+, w^-)$ which covers the entire patch, and the Rindler coordinates $(y^+, y^-)_{L/R}$ which cover respectively the left/right Rindler patches (the left BH exterior+the left bath and similarly for the right). In Rindler coordinates, the metrics for the black hole exterior and the respective baths are given by

$$ds_{\text{in}}^2 = -\frac{4\pi^2}{\beta^2}\frac{dy^+ dy^-}{\sinh^2\left[\frac{\pi}{\beta}(y^- - y^+)\right]}, \quad ds_{\text{out}}^2 = -\frac{dy^+ dy^-}{\epsilon^2},$$

(107)

where $\epsilon$ is the UV cut-off in the corresponding boundary theory. As already mentioned $y_L^\pm = t \mp z$ covers the left exterior and the bath while $y_R^\pm = t \pm z$ covers the right exterior and the corresponding bath. Under the transformation

$$w^\pm = \pm e^{\pm\frac{2\pi y_R^\pm}{\beta}}, \quad w^\pm = \mp e^{\mp\frac{2\pi y_L^\pm}{\beta}},$$

(108)

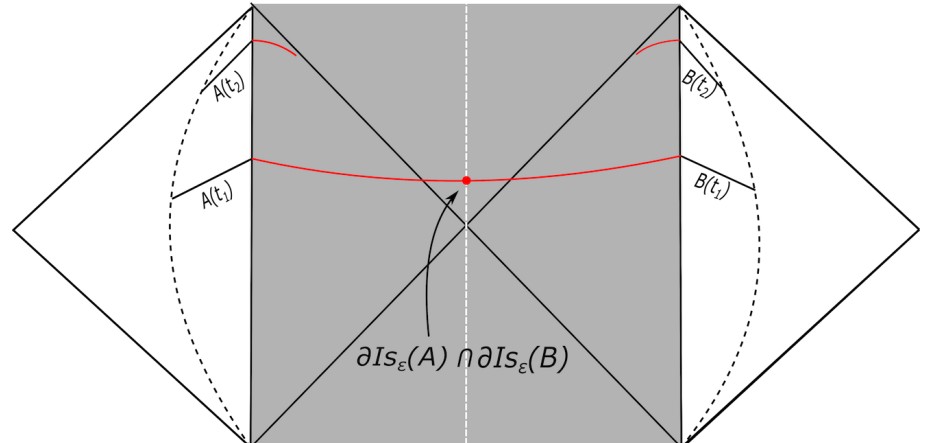

Figure 20: Penrose diagram for the eternal black hole coupled to two rigid Minkowski regions referred to as baths. Figure modified from [69].

the metrics become

$$ds^2_{\text{in}} = -\frac{4dw^+dw^-}{(1+w^+w^-)^2}, \quad ds^2_{\text{out}} = -\frac{\beta^2}{4\pi^2\epsilon^2}\frac{dw^+dw^-}{w^+w^-}. \tag{109}$$

By re-expressing the above metrics in a general form $ds^2 = -\Omega^{-2}dw^+dw^-$, the conformal factors can be read off immediately:

$$\Omega_{\text{in}} = \frac{1+w^+w^-}{2}, \quad \Omega_{\text{out}} = \frac{2\pi\epsilon}{\beta}\sqrt{w^+w^-}. \tag{110}$$

The dilaton is only defined in the gravity region $\Sigma$ and is given by

$$\phi = \phi_0 + \frac{2\pi\phi_r}{\beta}\coth\left[\frac{\pi}{\beta}(y^- - y^+)\right] = \phi_0 + \frac{2\pi\phi_r}{\beta}\frac{1-w^+w^-}{1+w^+w^-}, \tag{111}$$

with $\phi_b = \phi_r/\epsilon$ at the boundary.

## 5.2 On the Computation of $S^{(1/2)}_{\text{gen}}$

In this subsection we make some general comments on the computation of $S^{(1/2)}_{\text{gen}}$ for generic intervals in the above bulk AdS$_2$ plus the bath manifold, outside the black hole horizons. Relying on the large central charge behaviour of the matter CFT$_2$, we may again consider two possibilities depending on the size of the interval. For a large enough interval $[c_1, c_2]$ lying within the fixed geometry of the baths, we get an entanglement island $[a(c_1), a(c_2)]$. The effective Renyi entropy of order half adopts a similar factorization as eq.(43) owing to the large central charge behaviour of the four point twist correlator. Therefore, we may write down the generalized Renyi entropy of order half for this generic single interval configuration as

$$\begin{aligned}
S^{(1/2)}_{\text{gen}}([c_1, c_2]) = {} & 2\phi_0 + \frac{3\pi\phi_r}{2\beta}\left[\coth\left(\frac{2\pi a(c_1)}{\beta}\right) + \coth\left(\frac{2\pi a(c_2)}{\beta}\right)\right] \\
& + \frac{c}{4}\left[\log\left(\frac{2\beta}{\pi\epsilon}\frac{\sinh^2\left(\frac{\pi(a(c_1)+c_1)}{\beta}\right)}{\sinh\left(\frac{2\pi a(c_1)}{\beta}\right)}\right) + \log\left(\frac{2\beta}{\pi\epsilon}\frac{\sinh^2\left(\frac{\pi(a(c_2)+c_2)}{\beta}\right)}{\sinh\left(\frac{2\pi a(c_2)}{\beta}\right)}\right)\right].
\end{aligned} \tag{112}$$

Note that in writing the above expression we have used the fact that the geometric backreaction to the "Renyi area" in eq.(29) may be written in the form

$$\text{Area}^{1/2}(x) = \phi_0 + \frac{3\pi\phi_r}{2\beta} \coth\left(\frac{2\pi x}{\beta}\right). \tag{113}$$

We will demonstrate the above expression for the area of the backreacted region in JT gravity in appendix A.1. Furthermore, we note that the topological contribution $\phi_0$ does not get affected by the backreaction, while the backreaction of the dynamical part acquires a factor $\mathcal{X}_2 = \frac{3}{2}$.

Next we look at the configuration where the interval $[c_1, c_2]$ is much smaller and therefore does not admit an entanglement island. In that case, we get the standard $\text{CFT}_2$ result for a single interval at finite temperature

$$S_{\text{gen}}^{(1/2)}([c_1, c_2]) = S_{\text{eff}}^{1/2}([c_1, c_2]) = \frac{c}{2} \log\left[\frac{\beta}{\pi\epsilon} \sinh\left(\frac{2\pi(c_1 - c_2)}{\beta}\right)\right]. \tag{114}$$

We will make use of the results in Eqs.(112) and (114) for computing the entanglement negativity for various bipartite mixed state configurations involving different subsystems in the bath as well as the black hole exteriors in the following.

## 5.3 Disjoint Intervals in the Bath

In this section, we compute the entanglement negativity for the mixed state configuration of two disjoint intervals in the left and the right baths, respectively. First we compute the entanglement negativity using the proposal eq.(29) involving a specific algebraic sum of the generalized Renyi entropies of order half inspired by [81,82]. In this context, we will promote the matter $\text{CFT}_2$ to be holographic and digress into a doubly holographic picture [9] of the above configuration, arguing for the consistency of the formula used. Later we will also compute the entanglement negativity for the same configuration using our island proposal given in eq.(34) and demonstrate that the results match exactly.

**Proposal-I**

At $t = 0$ we consider the intervals $A = [-b, 0]$ and $B = [0, b]$ in the left and right Minkowski regions respectively. In this symmetric setup, at early times the corresponding entanglement islands in the black hole spacetime will be the entire bulk Cauchy slice, with a cross-section $a'$ splitting it ( see fig.[20]). At late times the entanglement islands of $A$ and $B$ becomes disconnected and therefore the corresponding island for the entanglement negativity disappears. We will be looking at the early time picture throughout this subsection. The double holography picture of the system under consideration is shown in fig.[21]. Note that in this case the subsystem $C$ sandwiched between $A$ and $B$ extends over parts of both the right and the left Minkowski regions; particularly the infinities of the Minkowski patches may be identified, which is depicted by the thin light blue curve. The justification of this construction comes from the fact that the $\text{CFT}_2$ in the entire bath region coupled to semiclassical gravity is together in the thermofield double state which is pure.

We compute the entanglement negativity between the subsystems $A$ and $B$ using the proposal in eq.(29). We will perform computations in global coordinates $(w^+, w^-)$. Let $w_1^\pm$ denote $a'$, $w_2^\pm$ denote the endpoint of the left bath, and $w_3^\pm$ denote the endpoint of the right bath. In addition we set $w_1^\pm = \delta$, by symmetry. Utilizing (112),(113) and (114) we can readily obtain

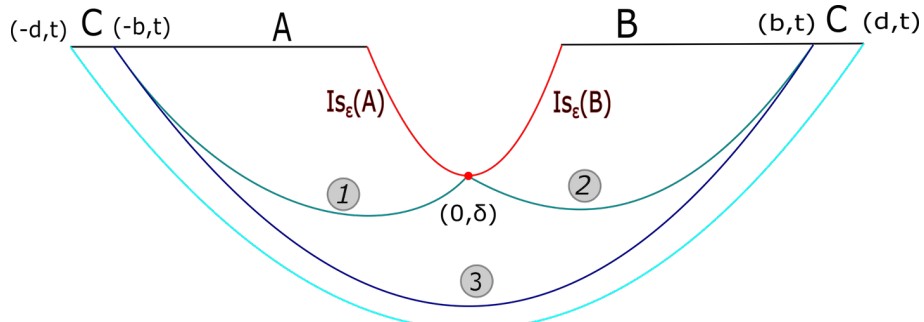

Figure 21: Schematic of the island proposal-I for the entanglement negativity of the mixed state of the disjoint intervals in the bath coupled to an eternal black hole

the generalized entanglement negativity as

$$
\mathcal{E}_{\text{gen}}(A:B) = \phi_0 + \frac{3\pi\phi_r}{2\beta}\frac{1-\delta^2}{1+\delta^2} + \frac{c}{4}\log 2
$$
$$
+ \frac{c}{8}\log\left[\frac{1}{\epsilon^2}\frac{e^{4\pi b/\beta}\left(e^{2\pi t/\beta} - \delta\,e^{-2\pi b/\beta}\right)^2\left(\delta\,e^{-2\pi b/\beta} - e^{-2\pi t/\beta}\right)^2}{(1+\delta^2)\cosh^2\left(\frac{2\pi t}{\beta}\right)}\right]. \tag{115}
$$

In writing this expression, we have used the fact that in taking the specific linear combinations of the generalized Renyi entropies in eq.(29), all the terms except those depending solely on $\delta$, get cancelled. Again this is an artifact of our proposal, which dictates that this expression has to be extremized with respect to the position of the intersection of the islands for the entanglement negativity $Q' = \partial\text{Is}_{\mathcal{E}}(A) \cup \partial\text{Is}_{\mathcal{E}}(B)$, which is nothing but the $\delta$ dependent term above. This provides a strong consistency check of our proposal. Also note that we have assumed that the subsystem $C$ sandwiched between $A$ and $B$ is very small conforming to the proximity limit $b \to \infty$, and therefore is denied an entanglement island.

We now extremize the expression (115) with respect to the position of the intersection of the islands for the entanglement negativity, namely over $\delta$, which leads to the symmetric limit $\delta \to 0$, in the proximity limit $b \to \infty$. Therefore, the total entanglement negativity for the symmetric setup of two disjoint intervals in the left and right baths, is given by

$$
\mathcal{E}(A:B) = \phi_0 + \frac{3\pi\phi_r}{2\beta} + \frac{c}{4}\log 2 - \frac{c}{8}\log\left[\frac{\cosh^2\left(\frac{2\pi t}{\beta}\right)}{\epsilon^2\,e^{4\pi b/\beta}}\right]. \tag{116}
$$

Next we will look at another consistency check of our formalism from the double holography picture in fig.[21] for which we take the $\text{CFT}_2$ matter fields to be holographic as well. From usual $\text{AdS}_3/\text{CFT}_2$ we know that the Renyi entropy for an interval in the $\text{CFT}_2$ may be computed in terms of the area of a backreacted cosmic brane homologous to the interval [96]. The backreacted cosmic branes corresponding to the different subsytem entropies are shown in fig.[21]. Reformulating the proposal in eq.(29) in terms of these bulk cosmic branes it is easy to see that the area contribution to the entanglement negativity manifestly shows the cancellation of the terms independent of the position of the intersection of the islands for the entanglement negativity $Q'$ and therefore provides a justification for the earlier calculations. The effective entanglement negativity is simply obtained from the linear combination of 3d bulk geodesic lengths as follows

$$
\mathcal{E}_{\text{eff}} = \frac{3}{4}\left(\mathcal{L}_2 + \mathcal{L}_1 - \mathcal{L}_3\right), \tag{117}
$$

which may be rewritten in terms of the different subsystem entropies and the final expression matches with the effective part of the entanglement negativity in eq.(115). Note that in writing eq.(117), we have already assumed the proximity limit given by $b \to \infty$, as well as the purity of the entire Cauchy slice of the AdS$_2$ bulk plus the bath.

Note that the double holography picture relies heavily on the dynamics of the end-of-the-world "Planck" brane as described in section 6.2. We may, therefore, consider modified 3d bulk geodesics in the dual locally AdS$_3$ spacetime which gets an endpoint contribution whenever they land on the end-of-the-world brane. Therefore, we can re-express the total holographic entanglement negativity as the linear combination of these modified 3d bulk geodesics. At this point, it is important to mention that we have used the doubly holographic model only for the purpose of illustration and we are not performing any computations in the double holographic models in the present article.

**Proposal-II**

Next we will compute the entanglement negativity of for the time-reflection symmetric configuration of two identical subsystems $A$ and $B$ in the left and the right baths using the conjecture in eq.(34). At early times $AB$ has an entire Cauchy slice of the gravity region as its entanglement island (shown in orange in fig.[20]). At late times the entanglement island is disconnected and $S_R(A : B) = 0$. We will be looking at the early time phase only, when we have a non-trivial intersection of the islands for the entanglement negativity Q$'$.

### $\mathcal{E}_{\text{eff}}$ through the emergent twist operators

We now compute the effective semiclassical entanglement negativity using the emergent twist operators which arise due to presence of the entanglement islands. Similar to subsection (4.3), the effective entanglement negativity $\mathcal{E}_{\text{eff}}(A \cup \text{Is}_{\mathcal{E}}(A) : B \cup Is_{\mathcal{E}}(B))$ for the connected phase of the entanglement island may be written in terms of twist operators as follows

$$
\begin{aligned}
\mathcal{E}_{\text{eff}} &= \lim_{n_e \to 1} \log \left\langle \Omega^{2\Delta_{\tau_{n_e}^2}} \tau_{n_e}(w_2) \overline{\tau}_{n_e}^2(w_1) \tau_{n_e}(w_3) \right\rangle \\
&= \frac{c}{8} \log \left[ \frac{4}{\epsilon^2} \frac{w_{12}^+ w_{12}^- w_{13}^+ w_{13}^-}{w_{23}^+ w_{23}^- (1 + w_1^+ w_1^-)^2} \right] + const. \\
&= \frac{c}{8} \log \left[ \frac{1}{\epsilon^2} \frac{e^{4\pi b/\beta} (\delta \, e^{-2\pi b/\beta} + e^{-2\pi t/\beta})^2 (\delta \, e^{-2\pi b/\beta} + e^{-2\pi t/\beta})^2}{(1 + \delta^2) \cosh^2\left(\frac{2\pi t}{\beta}\right)} \right] + const.,
\end{aligned}
\tag{118}
$$

where we have used eq.(108) and (110) for coordinate transformation and conformal factors, respectively. Substituting eq.(113) for $\mathcal{A}^{(1/2)}$ and the effective entanglement negativity we obtained above, in our proposal for the generalized entanglement negativity given by eq.(34), we have:

$$
\begin{aligned}
\mathcal{E}_{\text{gen}}(A : B) =\,& \phi_0 + \frac{3\pi\phi_r}{2\beta} \frac{1 - \delta^2}{1 + \delta^2} \\
&+ \frac{c}{8} \log \left[ \frac{1}{\epsilon^2} \frac{e^{4\pi b/\beta} (\delta \, e^{-2\pi b/\beta} + e^{-2\pi t/\beta})^2 (\delta \, e^{-2\pi b/\beta} + e^{-2\pi t/\beta})^2}{(1 + \delta^2) \cosh^2\left(\frac{2\pi t}{\beta}\right)} \right] + const.
\end{aligned}
\tag{119}
$$

Note that the above result for the generalized entanglement negativity matches exactly with the corresponding expression we obtained in eq.(115) through proposal-I.

**$\mathcal{E}(A:B)$ through the generalized Renyi reflected entropy**

Next we perform the computation of the entanglement negativity for the two disjoint intervals described above from the Renyi reflected entropy of order half. The effective Renyi reflected entropy for the connected phase of the entanglement islands corresponding to $A \cup B$ may be computed using techniques developed in [67]:

$$
\begin{aligned}
S_{R\,\mathrm{eff}}^{(n)}&(A\cup \mathrm{Is}_R(A) : B\cup \mathrm{Is}_R(B))\\
&= \frac{1}{1-n}\log\left[\frac{\Omega_1^{2\Delta_n}\langle \sigma_{g_A}(w_2)\sigma_{g_Bg_A^{-1}}(w_1)\sigma_{g_B^{-1}}(w_3)\rangle}{\langle \sigma_m(w_2)\sigma_m(w_3)\rangle}\right]\\
&= \frac{1}{1-n}\log\left[\frac{\Omega_1^{2\Delta_n}C_{n,m}\,w_{12}^{-4\Delta_n}w_{13}^{-4\Delta_n}w_{23}^{-4n\Delta_m+4\Delta_n}}{w_{23}^{-4n\Delta_m}}\right]\\
&= -\frac{c}{12}\left(1+\frac{1}{n}\right)\log\left[\Omega_1^2\left(\frac{-w_{23}^+w_{23}^-}{w_{12}^+w_{12}^-w_{13}^+w_{13}^-}\right)(2m)^{-2}\right],
\end{aligned}
\tag{120}
$$

where in the last step we have used the relations [67]

$$
2\Delta_n = \frac{c}{12}\left(n-\frac{1}{n}\right)\quad,\quad C_{n,m}=(2m)^{-4\Delta_n}.
\tag{121}
$$

Setting $m\to 1$ and using eq.(110) one gets for the Renyi reflected entropy in the state $|\sqrt{\rho_{AB}}\rangle$:

$$
S_{R\,\mathrm{eff}}^{(n)} = \frac{c}{12}\left(1+\frac{1}{n}\right)\log\left[\frac{4}{\epsilon^2}\frac{w_{12}^+w_{12}^-w_{13}^+w_{13}^-}{w_{23}^+w_{23}^-(1+w_1^+w_1^-)^2}\right]+\frac{c}{6}\left(1+\frac{1}{n}\right)\log 2,
\tag{122}
$$

where we have introduced the UV cut-off $\epsilon$ to make the argument of the log dimensionless. Now substituting for the coordinates of different points we get, after some simple algebra

$$
S_{R\,\mathrm{eff}}^{(n)} = \frac{c}{12}\left(1+\frac{1}{n}\right)\log\left[\frac{4}{\epsilon^2}\frac{e^{4\pi b/\beta}(\delta e^{-2\pi b/\beta}+e^{-2\pi t/\beta})^2(\delta e^{-2\pi b/\beta}+e^{-2\pi t/\beta})^2}{(1+\delta^2)\cosh^2\left(\frac{2\pi t}{\beta}\right)}\right].
\tag{123}
$$

Substituting eq.(113) for $\mathcal{A}^{1/2}$ and the above expression for the effective Renyi entropy of order half in eq.(38) we obtain

$$
\begin{aligned}
S_{R\,\mathrm{gen}}^{(1/2)}(A:B) =\,& 2\phi_0 + \frac{3\pi\phi_r}{\beta}\frac{1-\delta^2}{1+\delta^2}\\
&+\frac{c}{4}\log\left[\frac{4}{\epsilon^2}\frac{e^{4\pi b/\beta}(\delta e^{-2\pi b/\beta}+e^{-2\pi t/\beta})^2(\delta e^{-2\pi b/\beta}+e^{-2\pi t/\beta})^2}{(1+\delta^2)\cosh^2\left(\frac{2\pi t}{\beta}\right)}\right].
\end{aligned}
\tag{124}
$$

Therefore, the generalized entanglement negativity may now be obtained by substituting the above result in eq.(39) as:

$$
\begin{aligned}
\mathcal{E}_{\mathrm{gen}}(A:B) =\,& \phi_0 + \frac{3\pi\phi_r}{2\beta}\frac{1-\delta^2}{1+\delta^2}\\
&+\frac{c}{8}\log\left[\frac{4}{\epsilon^2}\frac{e^{4\pi b/\beta}(\delta e^{-2\pi b/\beta}+e^{-2\pi t/\beta})^2(\delta e^{-2\pi b/\beta}+e^{-2\pi t/\beta})^2}{(1+\delta^2)\cosh^2\left(\frac{2\pi t}{\beta}\right)}\right].
\end{aligned}
\tag{125}
$$

Note that the above expression for the generalized entanglement negativity matches precisely with the result we obtained in eq.(119) by an equivalent proposal in eq.(34). Furthermore it also matches with the entanglement negativity determined using proposal-I in eq.(115).

We must extremize this expression over the position of $a'$, namely over $\delta$. Note at this point that we must take the proximity limit $b \to \infty$, since otherwise we end up in the disconnected phase of the entanglement island. The extremization is fairly straightforward in this limit and corresponds to $\delta \to 0$, which is consistent with the symmetry of the setup. Therefore, the entanglement negativity between the subsystems $A$ and $B$ is given by

$$\mathcal{E} = \left( \phi_0 + \frac{3\pi\phi_r}{\beta} \right) - \frac{c}{8} \log \left[ \frac{\cosh^2\left(\frac{2\pi t}{\beta}\right)}{4\epsilon^2 \, e^{4\pi b/\beta}} \right], \tag{126}$$

which matches exactly with eq.(116) validating our proposals.

In [68] the authors computed the reflected entropy for the above configuration of two disjoint intervals in the left and the right baths, respectively. It is interesting to note that the entanglement negativity computed in eq.(126) or (116), looks quite similar to the corresponding result for the reflected entropy in [68]. This subtlety in the behaviour of the entanglement negativity arises from the fact that for spherical entangling surfaces, the area of the backreacted cosmic brane appearing in the entanglement negativity computations is proportional to the area of the original cosmic brane without backreaction [90].

## 5.4 Adjacent Intervals in the Bath and the Black Hole

In this subsection, we will look at a different scenario of two adjacent intervals, one inside the right bath and the other outside the black hole horizon. In fig.[22] the left/right quantum system is divided into two, $R_{L/R}$ and $B_{L/R}$, which we interpret as the subsystems in the bath and black hole exterior, respectively. We may identify $\tilde{I}_{L/R}$ as the islands for the entanglement negativity in the bath $R_{L/R}$. Note that $\tilde{I}_R \cup \tilde{B}_R$ constitute the whole right bulk, but in general $\tilde{I}_R$ and $I_R$ are not the same. Also the subsystem $\tilde{B}_R$ in the bulk has no notion of island. We will again compute the entanglement negativity for this configuration using the proposed formulae in eqs.(30) and (34).

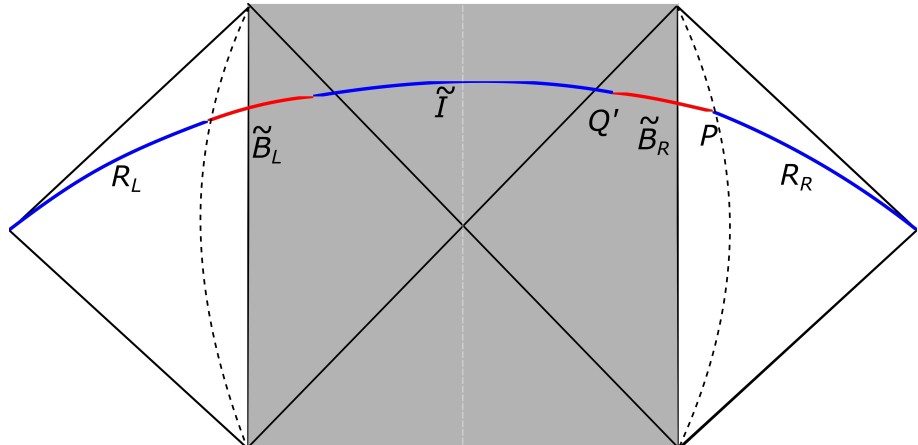

Figure 22: Schematic for the mixed state configuration of the adjacent intervals in the bath and the black hole. Figure modified from [69]

**Proposal-I**

We first compute the entanglement negativity our proposal in eqs.(30), by taking into the account for the fact that the subsystem $\tilde{B}_R$ in the AdS$_2$ bulk is lacking an entanglement island. As shown in fig.[22], $R_R$ joins $\tilde{B}_R$ at $P = [b, t]$. When the entanglement island is connected, $\tilde{I}_R$ meets $\tilde{B}_R$ at $Q' = [-a, t]$, which corresponds to the island for the entanglement negativity after extremization. It is easy to infer from fig.[22], that the islands for the entanglement negativity ceases to exist when the entanglement islands of the left and the right bath subsystems become disconnected as dictated by the phase transition of the entanglement entropy of $R_L$ and $R_R$. We will be interested in the connected phase of the entanglement island and therefore a non-trivial intersection of the islands for the entanglement negativity, in the following. We again compute in global coordinates, rendering the CFT$_2$ in its ground state, using the conformal map (108). Using equations (113),(112) and (114) we obtain the generalized entanglement negativity as

$$\mathcal{E}_{\text{gen}}(R_R : \tilde{B}_R) = \phi_0 + \frac{3\pi}{\beta} \frac{\phi_r}{\tanh\left(\frac{2\pi a}{\beta}\right)} + \frac{c}{4} \log\left[\frac{2\beta}{\pi\epsilon} \frac{\sinh^2\left(\frac{\pi(a+b)}{\beta}\right)}{\sinh\left(\frac{2\pi a}{\beta}\right)}\right]. \tag{127}$$

The extremization with respect to the position of $Q'$ leads to the following

$$\text{csch}\left(\frac{2\pi a}{\beta}\right) = \frac{\beta c}{12\pi\phi_r} \frac{\sinh\left(\frac{\pi(a-b)}{\beta}\right)}{\sinh\left(\frac{\pi(a+b)}{\beta}\right)}, \tag{128}$$

which incidentally is identical to the constraint on the entanglement island for a single interval $[0, b]$ inside the bath [15]. For $b \geq \frac{\beta}{2\pi}$ and $\frac{\phi_r}{\beta c} \gg 1$, an approximate solution to the above equation reads

$$a \simeq b + \frac{\beta}{2\pi} \log\left(\frac{24\pi\phi_r}{\beta c}\right), \tag{129}$$

showing that the islands for the entanglement negativity extend slightly outside the horizon.

Having computed the entanglement negativity for the above configuration of two adjacent intervals, we next promote the matter CFT$_2$ to be holographic and look at the doubly

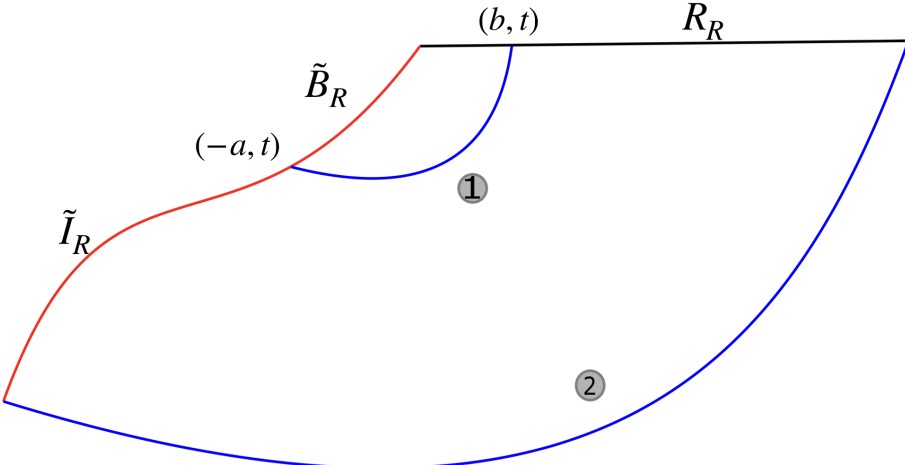

Figure 23: Schematic of island proposal-I for the entanglement negativity of the mixed state of the adjacent interval in the bath and the black hole

holographic model for this configuration. The double holography picture is shown in fig.[23], where the bulk geodesics corresponding to the different subsystem entropies are shown. Once again, the cancellation of the Renyi areas except that of $Q'$ is manifest from fig.[23], while the effective part of the entanglement negativity is given by the combination of 3d bulk geodesics as

$$
\begin{aligned}
\mathcal{E}_{\mathrm{eff}}(R_R \cup \tilde{I}_R : \tilde{B}_R) &= \frac{3}{4}\left[S_{\mathrm{eff}}(R_R \cup \tilde{I}_R) + S_{\mathrm{eff}}(\tilde{B}_R) - S_{\mathrm{eff}}(R_R \cup \tilde{I}_R \cup \tilde{B}_R)\right] \\
&= \frac{3}{4}\left((\mathcal{L}_1 + \mathcal{L}_2) + \mathcal{L}_1 - \mathcal{L}_2\right) \\
&= \frac{3}{2}\mathcal{L}_1.
\end{aligned}
\tag{130}
$$

The geodesic length $\mathcal{L}_1$ may be related to the corresponding subsystem entanglement entropy, which leads to the correct answer in eq.(127).

Once again, we may write down the total entanglement negativity in terms of the modified bulk geodesics in the braneworld scenario of double holography, similar to the case of two disjoint intervals. The modified geodesics pick up a contribution in terms of the area of a backreacted dilaton whenever they cross the Planck brane. We again stress on the fact that in the present article, we are only using the doubly holographic model as a visual aid, and not as a computational tool.

**Proposal-II**

We now turn our attention to compute the island contribution to the entanglement negativity of mixed state configuration of the adjacent intervals in the bath and the black hole, utilizing our proposal in eq.(34).

### $\mathcal{E}_{\mathrm{eff}}$ through the emergent twist operators

We consider two adjacent subsystems $\tilde{B}_R$ and $R_R$ similar to subsection (4.4) for the extremal black hole case. This configuration of two adjacent subsystems is illustrated in fig.[22]. Since $\tilde{B}_R$ lies in the black hole region, it has no island. The subsystem $R_R$ in the bath connects $\tilde{B}_R$ at $P \equiv [b, t]$ and its island $\tilde{I}_R$ joins $\tilde{B}_R$ at $Q' \equiv [-a, t]$. We will only do the computation here when the entanglement islands of the left and the right bath subsystems are connected. Then the effective entanglement negativity $\mathcal{E}_{\mathrm{eff}}(R_R \cup \tilde{I}_R : \tilde{B}_R)$ for the configuration of adjacent subsystems is given by

$$
\mathcal{E}_{\mathrm{eff}}(R_R \cup \tilde{I}_R : \tilde{B}_R) = \lim_{n_e \to 1} \log \left\langle \left(\prod_i \Omega_i^{2\Delta_i}\right) \tau_{n_e}(0) \overline{\tau}_{n_e}^2(Q') \tau_{n_e}^2(P) \overline{\tau}(\infty) \right\rangle,
\tag{131}
$$

where we have taken the arguments of twist operators in global coordinates. The above four point function can be factorized into the product of two 2-points functions in the large c limit as

$$
\left\langle \tau_{n_e}(0) \overline{\tau}_{n_e}^2(Q') \tau_{n_e}^2(P) \overline{\tau}(\infty) \right\rangle \approx \left\langle \tau_{n_e}(0) \overline{\tau}(\infty) \right\rangle \left\langle \overline{\tau}_{n_e}^2(Q') \tau_{n_e}^2(P) \right\rangle.
\tag{132}
$$

Substituting eq.(108), eq.(110) and (132) in eq. (131), the effective entanglement negativity may be obtained to be as follows

$$
\mathcal{E}_{\mathrm{eff}} = \frac{c}{4} \log \left[\frac{2\beta}{\pi\epsilon} \frac{\sinh^2\left(\frac{\pi(a+b)}{\beta}\right)}{\sinh\left(\frac{2\pi a}{\beta}\right)}\right].
\tag{133}
$$

Substituting eq.(113) for $\mathcal{A}^{(1/2)}$ and the effective entanglement negativity we determined above, in eq.(34), we obtain the generalized entanglement negativity as follows

$$\mathcal{E}_{\text{gen}}(R_R : \tilde{B}_R) = \phi_0 + \frac{3\pi}{\beta} \frac{\phi_r}{\tanh\left(\frac{2\pi a}{\beta}\right)} + \frac{c}{4} \log\left[\frac{2\beta}{\pi\epsilon} \frac{\sinh^2\left(\frac{\pi(a+b)}{\beta}\right)}{\sinh\left(\frac{2\pi a}{\beta}\right)}\right] . \tag{134}$$

Note that this matches exactly with the generalized entanglement negativity obtained through proposal-I in eq.(127).

### $\mathcal{E}(R_R : \tilde{B}_R)$ through the generalized Renyi reflected entropy

We now proceed to compute the effective contribution to the entanglement negativity for the above configuration of two adjacent intervals in the black hole exterior and the bath from the corresponding Renyi reflected entropy of order half. To proceed we will need to investigate a slightly different proposal [69] for the reflected entropy in the scenario of eternal black holes in JT gravity. The formula for the reflected entropy between the intervals $R_R$ and $\tilde{B}_R$, proposed in [69], reads

$$S_R(R_R : \tilde{B}_R) = \min \text{ext}_Q \left[\frac{2\,\mathcal{A}(Q' = \partial\tilde{I}_R \cap \partial\tilde{B}_R)}{4G_N} + S_R^{\text{eff}}(R_R \cup \tilde{I}_R : \tilde{B}_R)\right]. \tag{135}$$

As an illustration, we review the computation of the reflected entropy in [69], in terms of the global coordinates. Recall that the Rindler coordinates of the points $P$ and $Q'$ are $(b, t)$ and $(-a, t)$ respectively. The cross section term in eq.(135) is simply given by $(4G_N = 1)$

$$\mathcal{A}(Q' = \partial\tilde{I}_R \cap \partial\tilde{B}_R) = \left(\phi_0 + \frac{2\pi}{\beta} \frac{\phi_r}{\tanh\left(\frac{2\pi a}{\beta}\right)}\right). \tag{136}$$

The effective semiclassical part of the reflected entropy is nothing but the von Neumann entropy of $\tilde{B}_L \cup \tilde{B}_R$. Now using the map in (108), one can easily obtain

$$\begin{aligned}
S_{R\,\text{eff}}(R_R \cup \tilde{I}_R : \tilde{B}_R) &= S_{\text{eff}}(\tilde{B}_L \cup \tilde{B}_R) \\
&= \frac{c}{6} \log\left[\frac{(w^+(P) - w^+(Q'))(w^-(P) - w^-(Q'))}{\Omega(P)\Omega(Q')}\right] \\
&= \frac{c}{6} \log\left[\frac{\left(e^{-2\pi a/\beta} - e^{2\pi b/\beta}\right)^2}{\frac{1}{2}\left(1 - e^{-4\pi a/\beta}\right)\frac{2\pi\epsilon}{\beta}e^{2\pi b/\beta}}\right] \\
&= \frac{c}{3} \log\left[\frac{2\beta}{\pi\epsilon} \frac{\sinh^2\left(\frac{\pi(a+b)}{\beta}\right)}{\sinh\left(\frac{2\pi a}{\beta}\right)}\right].
\end{aligned} \tag{137}$$

This concludes the computation of the reflected entropy for the configuration of the two adjacent intervals. However, in order to compute the entanglement negativity between the subsystems in the radiation and the bulk, we are required to obtain the Renyi reflected entropy of order half. Following [67, 68], we may write down the effective Renyi reflected entropy in

the purified state $|\rho_{R_R \cup B_R}^{m/2}\rangle$ as

$$
\begin{aligned}
&S_{R\,\text{eff}}^{(m,n)}(R_R \cup \tilde{I}_R : \tilde{B}_R) \\
&= \frac{1}{1-n} \log \left[ \frac{\left(\prod_i \Omega_i^{2h_i}\right) \langle \sigma_{g_A}(0) \sigma_{g_B g_A^{-1}}(Q') \sigma_{g_A g_B^{-1}}(P) \sigma_{g_A^{-1}}(\infty)\rangle_{mn}}{\left[\left(\prod_i \Omega_i^{2h_i(n=1)}\right) \langle \sigma_{g_m}(0) \sigma_{g_m g_m^{-1}}(Q') \sigma_{g_m g_m^{-1}}(P) \sigma_{g_m^{-1}}(\infty)\rangle_m\right]^n} \right] \\
&= \frac{1}{1-n} \log \left[ \left(\Omega(P)\Omega(Q')\right)^{4\Delta_n} \frac{\langle \sigma_{g_A}(0) \sigma_{g_B g_A^{-1}}(Q') \sigma_{g_A g_B^{-1}}(P) \sigma_{g_A^{-1}}(\infty)\rangle_{mn}}{\left(\langle \sigma_{g_m}(0) \sigma_{g_m^{-1}}(\infty)\rangle_m\right)^n} \right],
\end{aligned}
\tag{138}
$$

where the arguments of the twist operators are in the global coordinates, $h_i = \frac{\Delta_i}{2}$ and $h_i\,(n=1)$ denotes $h_i$ evaluated at $n=1$. The above four-point function in the numerator factorizes into two 2-point functions in the large central charge limit as follows

$$
\langle \sigma_{g_A}(0) \sigma_{g_B g_A^{-1}}(Q') \sigma_{g_A g_B^{-1}}(P) \sigma_{g_A^{-1}}(\infty)\rangle_{mn} \approx \langle \sigma_{g_A}(0) \sigma_{g_A^{-1}}(\infty)\rangle_{mn} \langle \sigma_{g_B g_A^{-1}}(Q') \sigma_{g_A g_B^{-1}}(P)\rangle_{mn}.
\tag{139}
$$

The above expression is independent of the purifier index $m$ and hence setting $m=1$ leads to the effective Renyi reflected entropy as follows

$$
\begin{aligned}
S_{R\,\text{eff}}^{(n)}(R_R \cup \tilde{I}_R : \tilde{B}_R) &\approx \frac{1}{1-n} \log \left[ \left(\Omega(P)\Omega(Q')\right)^{4\Delta_n} \langle \sigma_{g_B g_A^{-1}}(Q') \sigma_{g_A g_B^{-1}}(P)\rangle_{mn} \right] \\
&= \frac{c}{6}\left(1+\frac{1}{n}\right) \log \left[ \frac{w_{PQ'}^2}{\Omega(P)\Omega(Q')} \right].
\end{aligned}
\tag{140}
$$

Now, we use the following expressions for the conformal factors,

$$
\Omega(P) = \Omega(w^\pm)\Big|_{y=b} = \frac{2\pi\epsilon}{\beta}, \quad \Omega(Q') = \Omega(w^\pm)\Big|_{y=-a} = \frac{1}{2}\left(1 - e^{-4\pi a/\beta}\right)
\tag{141}
$$

and $w_{PQ} = e^{-2\pi a/\beta} - e^{2\pi b/\beta}$, to obtain the fairly simple expression

$$
S_{R\,\text{eff}}^{(n)}(R_R \cup \tilde{I}_R : \tilde{B}_R) = \frac{c}{6}\left(1+\frac{1}{n}\right) \log \left[ \frac{2\beta}{\pi\epsilon} \frac{\sinh^2\left(\frac{\pi(a+b)}{\beta}\right)}{\sinh\left(\frac{2\pi a}{\beta}\right)} \right].
\tag{142}
$$

One consistency check for this expression is that setting $n=1$, we get back the expression in eq.(137), and therefore the twist operator analysis is consistent with the definition of the reflected entropy as the von Neumann entropy of one of the subsystems and its purifier. Note that the authors of [69] took the matter fields to be fermionic and therefore used a different formula from [100] to compute the effective contribution to the reflected entropy. Therefore the eq.(142) for the reflected entropy derived here and the corresponding expression in [69] look quite distinct.

Substituting the above determined effective Renyi reflected entropy for the adjacent intervals in eq.(38) we obtain the generalized Renyi reflected entropy of order half to be as follows

$$
S_{R\,\text{gen}}^{(1/2)}(R_R : \tilde{B}_R) = 2\phi_0 + \frac{6\pi}{\beta} \frac{\phi_r}{\tanh\left(\frac{2\pi a}{\beta}\right)} + \frac{c}{2} \log \left[ \frac{2\beta}{\pi\epsilon} \frac{\sinh^2\left(\frac{\pi(a+b)}{\beta}\right)}{\sinh\left(\frac{2\pi a}{\beta}\right)} \right],
\tag{143}
$$

where we have used eq.(113) for $\mathcal{A}^{(1/2)}$ in eq.(38). Finally, we may compute the generalized entanglement negativity using the proposal in eq.(39) as

$$\mathcal{E}_{\text{gen}}(R_R : \tilde{B}_R) = \phi_0 + \frac{3\pi}{\beta} \frac{\phi_r}{\tanh\left(\frac{2\pi a}{\beta}\right)} + \frac{c}{4} \log\left[\frac{2\beta}{\pi\epsilon} \frac{\sinh^2\left(\frac{\pi(a+b)}{\beta}\right)}{\sinh\left(\frac{2\pi a}{\beta}\right)}\right]. \qquad (144)$$

This expression is identical to the one computed through proposal-I and therefore the extremization with respect to the position of the intersection of the islands for the entanglement negativity yields the same result as before.

## 5.5 Single Interval in the Bath

In this subsection we will deal with the holographic entanglement negativity of a single interval in the bath outside the eternal black hole. For simplicity we take the subsystem $A$ inside the right bath to be sufficiently large so that the rest of the right bath, $B_1$, does not admit an island. The configuration is shown in fig.[24], where we choose a Cauchy slice at a given time and the corresponding subsystem on the left bath, denoted $B_2$, has an entanglement island. We will be interested in the early time phase where the entanglement islands corresponding to $A$ and $B_1$ are connected. The islands for the entanglement negativity $\text{Is}_{\mathcal{E}}(A)$ and $\text{Is}_{\mathcal{E}}(B_2)$ for the respective subsystems meet at the red blob which we interpret as the island cross-section for this configuration.

**Proposal-I**

Let us begin with the computation of the entanglement negativity for the above single interval in a bath coupled to an eternal black hole at a temperature $T = 1/\beta$. We take the interval $A = [0, b]$ and the corresponding entanglement island has the Rindler coordinates $[-a, 0]$. Once again we employ the global coordinates, in which $w_i^{\pm}$ $(i = 1, ..4)$ are the coordinates of respectively the left endpoint of $B_2$, the island cross section, right end of $A$ and right end of $B_1$, as shown in fig.[24]. Later we will take the bipartite limit $B \to A^c$, which corresponds to $w_{1,4} \to \infty$.

Using the proposal for single interval in eq.(31), we obtain for the generalized entangle-

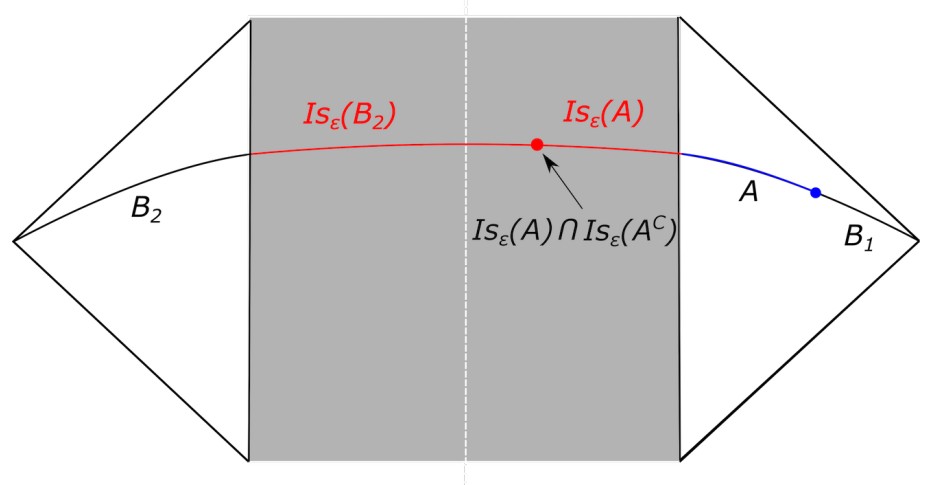

Figure 24: Schematic of a single interval in bath coupled to an eternal black hole.

ment negativity as

$$\mathcal{E}_{\text{gen}}(A) = \phi_0 + \frac{3\pi}{\beta}\frac{\phi_r}{\tanh\left(\frac{2\pi a}{\beta}\right)} + \frac{c}{4}\left[\log\left(\frac{2\beta}{\pi\epsilon}\frac{\sinh^2\left(\frac{\pi(a+b)}{\beta}\right)}{\sinh\left(\frac{2\pi a}{\beta}\right)}\right) - \log\left(\frac{2\pi\epsilon}{\beta}e^{2\pi b/\beta}\right)\right],$$

(145)

where we have used equations (112),(113) and (114). Extremization with respect to the position $a$ leads to the same constraint equation as given earlier in (128), and hence, in this case also it lies outside the horizon.

The double holography picture for this configuration is shown below. The 3d bulk geodesics corresponding to the different subsystem entropies are depicted and are numbered in order of the respective terms in the proposal eq.(31). Once again we see a cancellation of the terms except the one depending on the position of the intersection of the islands for the entanglement negativity. The effective part of the entanglement negativity may be obtained from the linear combination of the 3d bulk geodesic lengths as

$$\mathcal{E}_{\text{eff}} = = \frac{3}{4}\left(2\mathcal{L}_1 + \mathcal{L}_2 + \mathcal{L}_3 - \mathcal{L}_4 - \mathcal{L}_5\right).$$

(146)

A straightforward computation of the bulk geodesic lengths in terms of the $\text{CFT}_2$ twist correlators leads to the effective part of the generalized entanglement negativity in eq.(145) validating the consistency of our construction. As in the case of two disjoint or adjacent intervals, we can re-express the total holographic entanglement negativity by replacing the ordinary geodesics in eq.(146) by the modified geodesics in the locally $\text{AdS}_3$ bulk.

**Proposal-II**

Having computed the entanglement negativity for a single interval by taking into account the corresponding island contributions using proposal-I, we now determine the same using proposal-II and demonstrate that the results from both the proposals match exactly.

### $\mathcal{E}_{\text{eff}}$ through the emergent twist operators

The configuration for a single interval is described by $A \equiv [0, b]$ in the right bath and the rest of the system by $B \equiv B_1 \cup B_2$ similar to that of subsection 4.5. The size of the interval $B_1$ is taken

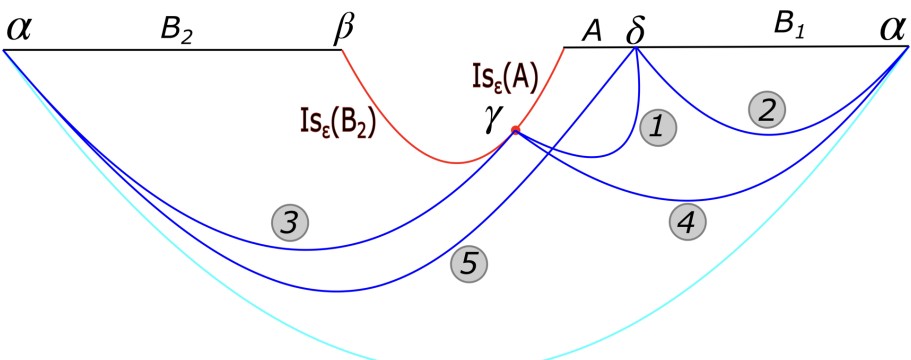

Figure 25: Schematic of the entanglement island proposal-I for the entanglement negativity of a single interval in a bath coupled to an eternal black hole.

to be small such that it does not admit an island and $B_2$ to be the whole of the left bath i.e $B_2 \equiv [-\infty, 0]$. The corresponding entanglement island of $A$ is $I_A \equiv [-a, 0]$. This configuration is illustrated in fig.[24]. Now the effective entanglement negativity $\mathcal{E}_{\text{eff}}(A \cup \text{Is}_{\mathcal{E}}(A) : B \cup \text{Is}_{\mathcal{E}}(B))$ may be written in terms of the emergent twist operators as

$$\mathcal{E}_{\text{eff}} = \lim_{n_e \to 1} \log \left\langle \Omega^{2\Delta_{\tau_{n_e}^2}}(w_2) \tau_{n_e}(w_1) \overline{\tau}_{n_e}^2(w_2) \tau_{n_e}^2(w_3) \overline{\tau}(w_4) \right\rangle. \tag{147}$$

The above four point function can be written in the large c limit as follows ($w_{1,4} \to \infty$)

$$\left\langle \tau_{n_e}(w_1) \overline{\tau}_{n_e}^2(w_2) \tau_{n_e}^2(w_3) \overline{\tau}(w_4) \right\rangle \approx \left\langle \tau_{n_e}(w_1) \overline{\tau}(w_4) \right\rangle \left\langle \overline{\tau}_{n_e}^2(w_2) \tau_{n_e}^2(w_3) \right\rangle. \tag{148}$$

Substituting eq.(108) and (148) in eq. (147), the effective entanglement negativity for the configuration of single interval is determined to be as follows

$$\mathcal{E}_{\text{eff}} = \frac{c}{4} \left[ \log \left( \frac{2\beta}{\pi\epsilon} \frac{\sinh^2\left(\frac{\pi(a+b)}{\beta}\right)}{\sinh\left(\frac{2\pi a}{\beta}\right)} \right) - \log \left( \frac{2\pi\epsilon}{\beta} e^{2\pi b/\beta} \right) \right]. \tag{149}$$

Note that in the above expression for the effective entanglement negativity, the second term is proportional to the thermal entropy which is subtracted from the first term proportional to the effective entanglement entropy, a feature also observed for the holographic entanglement negativity for this configuration in [76].

We now substitute eq.(113) for $\mathcal{A}^{(1/2)}$ and the effective entanglement negativity we determined above, in eq.(34) to obtain the generalized entanglement negativity as follows

$$\mathcal{E}_{\text{gen}}(A) = \phi_0 + \frac{3\pi}{\beta} \frac{\phi_r}{\tanh\left(\frac{2\pi a}{\beta}\right)} + \frac{c}{4} \left[ \log \left( \frac{2\beta}{\pi\epsilon} \frac{\sinh^2\left(\frac{\pi(a+b)}{\beta}\right)}{\sinh\left(\frac{2\pi a}{\beta}\right)} \right) - \log \left( \frac{2\pi\epsilon}{\beta} e^{2\pi b/\beta} \right) \right], \tag{150}$$

which matches exactly with the generalized entanglement negativity in eq.(145) obtained through proposal-I.

### $\mathcal{E}(A)$ through the generalized Renyi reflected entropy

In order to utilize our proposal-II in eq.(34), we compute the effective semiclassical contribution to the entanglement negativity of the single interval configuration. In most generality, an analysis of the reflected entropy for the above configuration needs the tools of multipartite reflected entropy and its holography [101, 102]. In this article, we avoid these complications by considering $B_1$ and $B_2$ to be parts of a single subsystem, $B_1 \cup B_2 = B$ and compute the reflected entropy between the subsystems $A$ and $B$. Finally we send $B \to A^c$, and interpret the result as the reflected entropy of $A$ with the rest of the Cauchy slice in the bath region.

Following [67, 68] we may write down the effective Renyi reflected entropy in the state $\rho_{A \cup B}^{m/2}$ for the above configuration as:

$$S_{R\,\text{eff}}^{(n)}(A \cup \text{Is}_R(A) : B \cup \text{Is}_R(B))$$
$$= \frac{1}{1-n} \log \left[ \frac{\Omega(w_2)^{4\Delta_n} \langle \sigma_{g_{B_2}}(w_1) \sigma_{g_A g_{B_2}^{-1}}(w_2) \sigma_{g_{B_1} g_A^{-1}}(w_3) \sigma_{g_{B_1}^{-1}}(w_4) \rangle_{mn}}{\left( \langle \sigma_{g_m}(w_1) \sigma_{g_m^{-1}}(w_4) \rangle_m \right)^n} \right]. \tag{151}$$

Eventually we are going to take the limit $w_{1,4} \to \infty$, so that the above expression computes the Renyi reflected entropy of $A$ with the complementary subsystem in the bath. Now using

standard OPE arguments the four point function in the numerator can be written as the product of the following two-point functions in the large central charge limit (also $w_{1,4} \to \infty$):

$$
\langle \sigma_{g_{B_2}}(w_1) \sigma_{g_A g_{B_2}^{-1}}(w_2) \sigma_{g_{B_1} g_A^{-1}}(w_3) \sigma_{g_{B_1}^{-1}}(w_4) \rangle_{mn}
$$
$$
\approx \langle \sigma_{g_{B_2}}(w_1) \sigma_{g_{B_1}^{-1}}(w_4) \rangle_{mn} \langle \sigma_{g_A g_{B_2}^{-1}}(w_2) \sigma_{g_{B_1} g_A^{-1}}(w_3) \rangle_{mn} . \tag{152}
$$

Therefore, again the Renyi reflected entropy for the mixed state $\rho_{AB}$ may be obtained by trivially setting the purifier index to $m \to 1$, and we obtain

$$
\begin{aligned}
&S_{R\,\mathrm{eff}}^{(n)}(A \cup \mathrm{Is}_R(A) : B \cup \mathrm{Is}_R(B)) \\
&= \frac{1}{1-n} \log \left[ \Omega(w_2)^{4\Delta_n} \left( w_{23}^+ w_{23}^- \right)^{-4\Delta_n} \right] \\
&= \frac{c}{6}\left(1 + \frac{1}{n}\right)\left[ \log\left( \frac{w_{23}^+ w_{23}^-}{\Omega(w_2)\Omega(w_3)} \right) - \log \Omega(w_3) \right],
\end{aligned} \tag{153}
$$

where in the last equality we can make an identification of the first term on the right as the von Neumann entropy of the interval $A$. Now setting $n = \frac{1}{2}$ and substituting the expressions for the warp factors from eq.(141), we obtain for the effective part of the Renyi reflected entropy of order half

$$
S_{R\,\mathrm{eff}}^{(1/2)}(A \cup \mathrm{Is}_R(A) : B \cup \mathrm{Is}_R(B)) = \frac{c}{2}\left[ \log\left( \frac{2\beta}{\pi\epsilon} \frac{\sinh^2\left(\frac{\pi(a+b)}{\beta}\right)}{\sinh\left(\frac{2\pi a}{\beta}\right)} \right) - \log\left( \frac{2\pi\epsilon}{\beta} e^{2\pi b/\beta} \right) \right] . \tag{154}
$$

Substituting the above determined effective Renyi reflected entropy for the adjacent intervals in eq.(38) we obtain the generalized Renyi reflected entropy of order half to be as follows

$$
S_{R\,\mathrm{gen}}^{(1/2)}(A : B) = 2\phi_0 + \frac{6\pi}{\beta} \frac{\phi_r}{\tanh\left(\frac{2\pi a}{\beta}\right)} + \frac{c}{2}\left[ \log\left( \frac{2\beta}{\pi\epsilon} \frac{\sinh^2\left(\frac{\pi(a+b)}{\beta}\right)}{\sinh\left(\frac{2\pi a}{\beta}\right)} \right) - \log\left( \frac{2\pi\epsilon}{\beta} e^{2\pi b/\beta} \right) \right] . \tag{155}
$$

Finally, using the above expression for the generalized Renyi reflected entropy of order half in eq.(39) we readily see that the expression for the generalized entanglement negativity of a single interval $A$ in the bath, is given by

$$
\mathcal{E}_{\mathrm{gen}}(A) = \phi_0 + \frac{3\pi}{\beta} \frac{\phi_r}{\tanh\left(\frac{2\pi a}{\beta}\right)} + \frac{c}{4}\left[ \log\left( \frac{2\beta}{\pi\epsilon} \frac{\sinh^2\left(\frac{\pi(a+b)}{\beta}\right)}{\sinh\left(\frac{2\pi a}{\beta}\right)} \right) - \log\left( \frac{2\pi\epsilon}{\beta} e^{2\pi b/\beta} \right) \right] . \tag{156}
$$

Note that as earlier in eq.(150), the term in the above expression within the brackets, for the effective entanglement negativity involves the subtraction of the thermal entropy from the effective entanglement entropy. Furthermore, we emphasize that the above expression is exactly identical to eq.(145) for the generalized entanglement negativity obtained using proposal-I.

## 6 Entanglement Negativity in the Double Holography Picture

In this section, we comment on the double holographic picture of our proposals for the island contributions to the entanglement negativity in quantum field theories coupled to semiclassical gravity. We begin by a very concise review of the double holography picture for the entanglement entropy [6, 9, 19] and the reflected entropy [68, 69] before proceeding to describe the same for the entanglement negativity.

## 6.1 Review of the Double Holographic Picture for the Entanglement entropy and the Reflected entropy

In the doubly holographic picture, the matter is described by a holographic $\text{CFT}_d$ ( for the case of JT gravity this is a $\text{CFT}_2$ ) which is coupled to a $d$ dimensional semiclassical gravity. The bulk dual of this configuration corresponds to a $(d + 1)$ dimensional locally *AdS* spacetime with a dynamical "*Planck*" brane at a finite boundary, similar to the Randall-Sundrum model as described in [6, 9, 19]. In this construction, the island contribution to the entanglement entropy for a subsystem-*A* in the $d$ dimensional bath CFT, to the leading order is determined by the RT/HRT surface in the higher dimensional bulk $\text{AdS}_{d+1}$ as follows

$$S(A) = \min_{\text{Is}(A)} \left\{ \text{ext}_{\text{Is}(A)} \left[ \frac{\text{Area}^{(d)}(\partial \text{Is}(A))}{4G_N^{(d)}} + \frac{\text{Area}^{(d+1)} \left[ X_{A \cup \text{Is}(A)} \right]}{4G_N^{(d+1)}} \right] \right\}, \qquad (157)$$

where $G_N^{(d)}$ and $G_N^{(d+1)}$ correspond to Newton's gravitational constants in $d$ and $(d + 1)$ dimensions respectively, and the superscript for the area term indicates the dimension of the ambient spacetime. In the above equation the area term has to be thought of as arising due to the "Planck" brane and the total sum should be considered as the area of a single RT/HRT surface $X_{A \cup \text{Is}(A)}$ in the dual bulk *AdS* spacetime. Furthermore, this was applied to the case of JT gravity with holographic matter in [9] to obtain the Page curve for the Hawking radiation in semiclassical gravity. Apparently, from the lower dimensional point of view the black hole interior seems completely disconnected from the bath, however, the double holographic scenario indicates that the two are connected via the entanglement wedge in the bulk $\text{AdS}_{d+1}$. Hence, the double holography picture also provides a manifestation of the ER=EPR proposal [103]. Following this, an application of the above construction to a higher dimensional example was considered in [19].

Similarly, a double holographic construction for the reflected entropy of a system involving a holographic matter $\text{CFT}_d$ coupled to d-dimensional semiclassical gravity was proposed in [68]. According to their proposal, the island contribution to the reflected entropy of a bipartite system-*AB* described in eq.(7) is obtained by the minimal EWCS in the higher dimensional bulk $\text{AdS}_{d+1}$ as follows

$$S_R(R_1 : R_2) = \min \left\{ \text{ext} \left[ \frac{\text{Area}^{(d)}(\partial \text{Is}_{R_1} \cap \partial \text{Is}_{R_2})}{4G_N^{(d)}} + \frac{\text{Area}^{(d+1)} \left[ \text{EWCS}(\text{Rad} \cup I) \right]}{4G_N^{(d+1)}} \right] \right\}, \qquad (158)$$

where the first area term has to be thought of as arising because the EWCS in the $\text{AdS}_{d+1}$ bulk spacetime, ends on the Planck brane. Therefore, the total sum in the above equation should be considered as a single EWCS in the dual bulk spacetime.

## 6.2 Double Holography for the Entanglement negativity

In this subsection, we develop a possible doubly holographic picture of our island proposals for the entanglement negativity in a bath coupled to a d-dimensional semiclassical gravitational theory with matter described by a holographic $\text{CFT}_d$.

**Proposal-I**

We now propose that the island contribution to the entanglement negativity of a bipartite system-*AB* is obtained by a the sum of the area of a combination of backreacted cosmic branes anchored on the subsystems/islands in a d-dimensional bath CFT coupled to semiclassical gravity, which extend into the higher dimensional bulk $\text{AdS}_{d+1}$ spacetime. For the case of entangling surfaces with spherical symmetry, the area of such a backreacted cosmic brane is

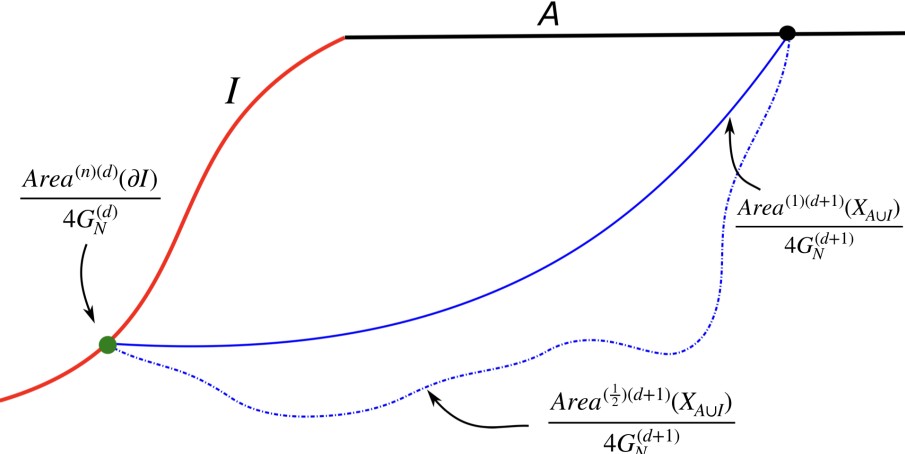

Figure 26: Schematic for double holography picture of our proposal-I. Note that the superscript in the first bracket in the area terms denotes the Renyi index, and the one in the second bracket indicates the dimension of the spacetime in which the surface is embedded.

proportional to the area of the corresponding RT/HRT surface with a dimension dependent constant that contains the information about the backreaction as described by eq.(15) [75, 90, 97]. Hence our conjecture for the disjoint, adjacent and single-connected subsystems are same as those described by eq.(29),(30) and (31) with the understanding that the generalized Renyi entropy of order half $S_{\text{gen}}^{(1/2)}(A)$ in these equations is given by

$$S_{\text{gen}}^{(1/2)}(A) = \mathcal{X}_d^{\text{hol}} \left[ \frac{\text{Area}^{(d)}[\partial \text{Is}(A)]}{4G_N^{(d)}} + \frac{\text{Area}^{(d+1)}[X_{A \cup \text{Is}(A)}]}{4G_N^{(d+1)}} \right]. \tag{159}$$

**Proposal-II**

Having described the doubly holographic construction for the entanglement negativity based on the proposal-I, we now proceed to describe an alternative construction based on proposal-II. We propose that the island contribution to the entanglement negativity for a bipartite system-$AB$ in a theory consisting of a holographic matter CFT$_d$ coupled to d-dimensional semiclassical gravity, is obtained by the area of a backreacted cosmic brane in the higher dimensional bulk AdS$_{d+1}$ anchored on the boundary of EWCS. For spherical entangling surfaces the area of such a backreacted cosmic brane is proportional to the area of the corresponding EWCS with a dimension dependent constant as described in eq.(15) [53, 90, 97]. Hence the island formula described by eq.(34) in the double holographic context may be expressed as follows

$$\mathcal{E}(A:B) = \mathcal{X}_d^{\text{hol}} \left[ \frac{\text{Area}^{(d)}[\text{EWCS}]}{4G_N^{(d)}} + \frac{\text{Area}^{(d+1)}[\text{EWCS}]}{4G_N^{(d+1)}} \right], \tag{160}$$

where the first term simply corresponds the area of the backreacted brane at the boundary of the EWCS ending on the Planck brane which is proportional to the area of $\partial \text{Is}_{\mathcal{E}}(A) \cap \partial \text{Is}_{\mathcal{E}}(B)$. Note that the entire sum inside the brackets in the above equation, is to be considered as the backreacted EWCS in the double holography set up.

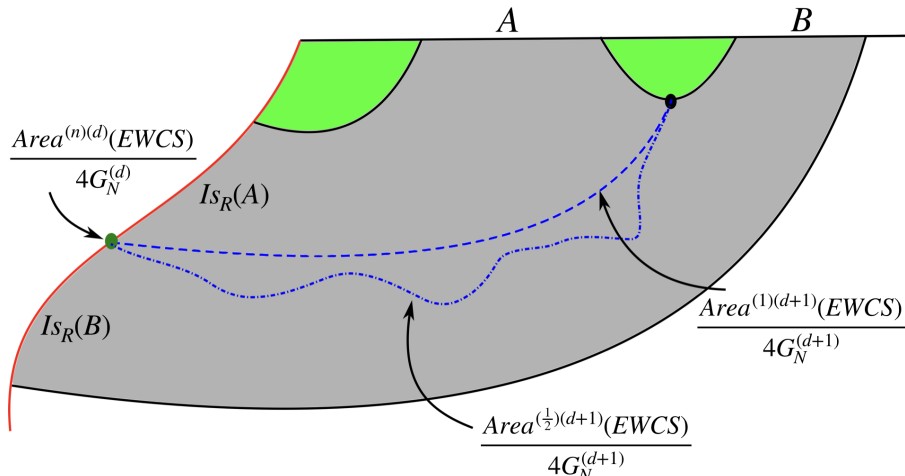

Figure 27: Schematic for double holography picture of our proposal-II. Note that the superscript in the first bracket in the area terms denotes the Renyi index, and the one in the second bracket indicates the dimension of the spacetime in which the surface is embedded.

## 7 Replica Wormholes and Islands for Entanglement Negativity

In this section, we will provide a derivation of the island formulae for the entanglement negativity in section 3. To this end, we recall the fact that the $k$-th Renyi generalization of the entanglement negativity for a bipartite mixed state $\rho_{AB}$ may be defined as [72, 73]

$$\mathcal{N}^{(k)}(A:B) = \text{Tr}\left[\left(\rho_{AB}^{T_B}\right)^k\right], \tag{161}$$

where the partial transpose $\rho_{AB}^{T_B}$ of the bipartite density matrix has been defined in eq. (9). The logarithmic entanglement negativity is then obtained through a replica trick which employs an even analytic continuation in the Renyi index $k$:

$$\mathcal{E}(A:B) = \lim_{n \to 1/2} \log \mathcal{N}^{(2n)}(A:B). \tag{162}$$

In order to formulate the replica wormhole calculations for the entanglement negativity in a setup of gravitational path integrals, we first consider a quantum field theory coupled to gravity on a hybrid manifold $\mathcal{M} \equiv \mathcal{M}^{\text{fixed}} \cup \mathcal{M}^{\text{bulk}}$, where $\mathcal{M}^{\text{fixed}}$ is non-gravitating with a fixed background metric, while $\mathcal{M}^{\text{bulk}}$ contains dynamical gravity. In the following we will assume that the quantum matter fields also extend into the fluctuating geometry $\mathcal{M}^{\text{bulk}}$ and the two parts of the geometry are joined smoothly across their boundary utilizing transparent boundary conditions. We wish to compute the generalized Renyi negativity between two generic regions $A$ and $B$ in the quantum field theory[5]. To proceed, we need to construct the replica manifold which computes the trace norm of the even powers of the partially transposed density matrix $\rho_{AB}^{T_B}$. Following [93], we will denote the above mentioned replica manifold as

$$\mathcal{M}_k^{A,B} = \mathcal{M}_k^{A,B\,(\text{fixed})} \cup \mathcal{M}_k^{A,B\,(\text{bulk})},$$

where $k$ is the Renyi index and the superscripts $A, B$ carry the idea that the replica geometry has to be constructed using a different cutting and gluing procedure than that for the Renyi

---

[5]Note that in the most general case $A$ and $B$ both may extend non-trivially inside the fluctuating geometry $\mathcal{M}^{\text{bulk}}$ as well.

entropy [72, 93]. The replica geometry $\mathcal{M}_k^{A,B}$ may be obtained form the original spacetime $\mathcal{M}_1 \equiv \mathcal{M}$ as follows:

**On the fixed geometry:** Following [72], we cut the $k$ copies of the original fixed manifold $\mathcal{M}^{\text{fixed}}$ along $A$ and $B$, and then glue them cyclically along $A$ and anti-cyclically along $B$. This fixes the topology of the replica manifold on the fixed background to be $\mathcal{M}_k^{A,B\,(\text{fixed})}$.

**On the fluctuating geometry:** In case of the fluctuating geometry, the task is a bit involved. Since the theory on this manifold contains gravity, we need to evaluate the full gravitational path integral $\mathbf{Z}[\mathcal{M}_k^{\text{bulk}}]$ in order to find the emergent geometry. Assuming that the bulk geometry can be treated semi-classically, we can perform a saddle-point approximation to the gravitational path integral:

$$\mathbf{Z}[\mathcal{M}_k^{\text{bulk}}] \approx e^{-I_{\text{grav}}[\mathcal{M}_k^{\text{bulk}}]} . \tag{163}$$

To determine the saddle-point solution, first we fix the topology utilizing the replica symmetry as well as the cutting and gluing procedure relevant to the computation of the Renyi negativity [72, 93]. Once we have fixed the topology, we impose the gravitational equations of motion to find the saddle-point solution $\mathcal{M}_k^{A,B\,(\text{bulk})}$. Note that while one fixes the the bulk topology of the fluctuating geometry, one must take care of the fact that in order to have a smooth replica manifold $\mathcal{M}_k^{A,B}$, the topology of the fixed geometry must coincide with that of the fluctuating geometry at the joining of the two portions of the manifold.

There are various possible ways to fix the topology of the saddle point solution $\mathcal{M}_k^{A,B}$ of the gravitational path integral as the fluctuating geometry need not obey the full $\mathbb{Z}_k$ replica symmetry. On the other hand, the fixed portion of the replica manifold $\mathcal{M}_k^{A,B\,(\text{fixed})}$ respects the complete replica symmetry [72].

We will now focus on the so called replica non-symmetric saddle [6] [93] for even $k = 2n$, denoted as $\mathcal{M}_{2n}^{A,B\,(\text{bulk, nsym})}$. As discussed before, the replica manifold for the fluctuating geometry has, as its asymptotic boundary, the fixed geometry $\mathcal{M}_{2n}^{A,B\,(\text{fixed, nsym})}$ which respects the full $\mathbb{Z}_{2n}$ replica symmetry. We will fix the topology of the bulk replica non-symmetric saddle utilizing the cutting and gluing procedure described in [93]. We will consider $2n$ copies of the original bulk manifold $\mathcal{M}_1^{\text{bulk}}$ and cut along three non-overlapping codimension-one homology hypersurfaces $\Sigma_A$, $\Sigma_B$ and $\Sigma_{\overline{AB}}$ satisfying the following homology conditions:

$$\partial \Sigma_X = X \cup \gamma_X , \tag{164}$$

where $X = A, B, \overline{AB}$ and $\gamma_X$ denotes a codimension-two hypersurface homologous to $X$. Along $\Sigma_A$, we glue odd numbered copies of $\mathcal{M}_1^{\text{bulk}}$ cyclically and even copies to themselves, along $\Sigma_B$, we glue even numbered copies anti-cyclically and odd copies to themselves and finally along $\Sigma_{\overline{AB}}$ we glue the copies pairwise, starting with the first two and finishing with the last two. This type of cutting and gluing manifestly breaks the replica symmetry group $\mathbb{Z}_{2n}$ to the subgroup $\mathbb{Z}_n$ [93]. Finally upon imposing the gravitational equations of motion onto this manifold, we obtain the full hybrid replica manifold:

$$\mathcal{M}_{2n}^{A,B} = \mathcal{M}_{2n}^{A,B\,(\text{fixed})} \cup \mathcal{M}_{2n}^{A,B\,(\text{bulk, nsym})} , \tag{165}$$

---

[6]As described in [93] if we consider the replica symmetric saddle in the bulk fluctuating geometry, the corresponding entanglement negativity turns out to be zero, and therefore this saddle does not give the dominant contribution to the gravitational path integral.

where as described earlier, the replica geometry on the fixed background $\mathcal{M}_{2n}^{A,B\,(\text{fixed})}$ respects the full $\mathbb{Z}_{2n}$ replica symmetry while the replication of the fluctuating geometry $\mathcal{M}_{2n}^{A,B\,(\text{bulk, nsym})}$ has only a residual $\mathbb{Z}_n$ replica symmetry. The partition function on this replica manifold is therefore factorized into a gravitational part and that computing the contributions from the quantum matter fields

$$
\begin{aligned}
\mathbf{Z}[\mathcal{M}_{2n}^{A,B}] &= \mathbf{Z}_{\text{grav}}[\mathcal{M}_{2n}^{A,B\,(\text{bulk, nsym})}]\,\mathbf{Z}_{\text{mat}}[\mathcal{M}_{2n}^{A,B}] \\
&= e^{-I_{\text{grav}}[\mathcal{M}_{2n}^{A,B\,(\text{bulk, nsym})}]}\,\mathbf{Z}_{\text{mat}}[\mathcal{M}_{2n}^{A,B}]\,,
\end{aligned}
\tag{166}
$$

where in the second equality we have made use of the saddle point approximation (163) for the gravitational partition function. In writing the contribution of the quantum matter fields to the partition function, we have utilized the fact that the quantum field theory extends over the full hybrid manifold $\mathcal{M}_{2n}^{A,B}$.

The generalized Renyi negativity between $A$ and $B$ may therefore be computed as the properly normalized replica partition function

$$
\begin{aligned}
\mathcal{N}_{\text{gen}}^{(2n)}(A:B) &= \frac{\mathbf{Z}[\mathcal{M}_{2n}^{A,B}]}{(\mathbf{Z}[\mathcal{M}_1])^{2n}} \\
&= e^{-I_{\text{grav}}[\mathcal{M}_{2n}^{A,B\,(\text{bulk, nsym})}]+2n\,I_{\text{grav}}[\mathcal{M}_1^{\text{bulk}}]}\,\frac{\mathbf{Z}_{\text{mat}}[\mathcal{M}_{2n}^{A,B}]}{(\mathbf{Z}_{\text{mat}}[\mathcal{M}_1])^{2n}}\,.
\end{aligned}
\tag{167}
$$

To proceed we first focus on the Hawking-type saddle [15, 35, 54, 55]. Since the bulk replica non-symmetric saddle retains the remnant replica symmetry $\mathbb{Z}_n$, it is natural to take a quotient of $\mathcal{M}_{2n}^{A,B\,(\text{bulk, nsym})}$ by the group $\mathbb{Z}_n$ to obtain the quotient manifold

$$
\hat{\mathcal{M}}_{2n}^{A,B\,(\text{bulk, nsym})} = \mathcal{M}_{2n}^{A,B\,(\text{bulk, nsym})}/\mathbb{Z}_n\,.
\tag{168}
$$

Note that the quotient manifold has the asymptotic boundary $\mathcal{M}_2^{A,B\,(\text{fixed})}$ [7]: a two-fold cover of the original fixed geometry $\mathcal{M}_1^{\text{fixed}}$ branched over $A$ and $B$. The quotient manifold has conical defects at $\gamma_{A_1}^{(n)}$ and $\gamma_{B_2}^{(n)}$, the loci of the fixed points of the residual replica symmetry. As usual, these conical defects are sourced by backreacting cosmic branes homologous to the subsystems $A$ and $B$, and come with deficit angles

$$
\Delta\phi_n = 2\pi\left(1-\frac{1}{n}\right)\,.
$$

The on-shell action of the fluctuating replica geometry may now be written in terms of the on-shell action of the quotient spacetime $I_{\text{grav}}\left(\mathcal{M}_2^{AB\,(\text{fixed})},\gamma_{A_1}^{(n)},\gamma_{B_2}^{(n)}\right)$ as

$$
I_{\text{grav}}[\mathcal{M}_{2n}^{A,B\,(\text{bulk, nsym})}] \equiv n\,I_{\text{grav}}\left(\mathcal{M}_2^{AB\,(\text{fixed})},\gamma_{A_1}^{(n)},\gamma_{B_2}^{(n)}\right)\,.
$$

Therefore, the generalized logarithmic Renyi negativity is given by

$$
\begin{aligned}
\mathcal{E}_{\text{gen}}^{(2n)}(A:B) &= \log\mathcal{N}_{\text{gen}}^{(2n)}(A:B) \\
&= -n\left[I_{\text{grav}}\left(\mathcal{M}_2^{AB\,(\text{fixed})},\gamma_{A_1}^{(n)},\gamma_{B_2}^{(n)}\right)-2\,I_{\text{grav}}[\mathcal{M}_1^{\text{bulk}}]\right]+\mathcal{E}_{\text{eff}}^{(2n)}(A:B)\,,
\end{aligned}
\tag{169}
$$

---

[7]Note that the two fold cover $\mathcal{M}_2^{A,B\,(\text{fixed})}$ computing the second Renyi negativity is the same as $\mathcal{M}_2^{AB\,(\text{fixed})}$, that computing the second Renyi entropy [93].

where the effective logarithmic Renyi negativity $\mathcal{E}_{\text{eff}}^{(2n)}(A:B)$ of quantum matter fields on the hybrid replica manifold $\mathcal{M}_{2n}^{A,B}$ is defined as

$$\mathcal{E}_{\text{eff}}^{(2n)}(A:B) = \log \frac{\mathbf{Z}_{\text{mat}}[\mathcal{M}_{2n}^{A,B}]}{(\mathbf{Z}_{\text{mat}}[\mathcal{M}_1])^{2n}} . \tag{170}$$

In order to evaluate the above expression we need to compute the on-shell action of the quotiented bulk geometry $\hat{\mathcal{M}}_{2n}^{A,B\,(\text{bulk, nsym})}$ which gets contributions from the cosmic branes sitting on $\gamma_{A_1}^{(n)}$ and $\gamma_{B_2}^{(n)}$ homologous to $A$ on the first copy of $\mathcal{M}_2^{AB}$ and to $B$ on the second copy. The authors in [15, 35, 54] had computed the contributions coming from such conical defects for $n \sim 1$ by expanding the on-shell action near $n = 1$, thereby assuming that the gravitational backreaction is small enough to keep the replica manifold a solution to the gravitational equations of motion. Here, instead, we follow the procedure in [104, 105] to find the solution away form $n = 1$ and obtain the effects of the gravitational backreaction comprehensively in terms of the areas of the backreacting cosmic branes sitting along $\gamma_{A_1}^{(n)}$ and $\gamma_{B_2}^{(n)}$. Therefore, the on shell action of the quotiented bulk geometry $\hat{\mathcal{M}}_{2n}^{A,B\,(\text{bulk, nsym})}$ can be written as

$$I_{\text{grav}}\left(\mathcal{M}_2^{AB\,(\text{fixed})}, \gamma_{A_1}^{(n)}, \gamma_{B_2}^{(n)}\right) = 2\,I_{\text{grav}}[\mathcal{M}_1^{\text{bulk}}] + \frac{\mathcal{A}^{(1/2)}(\gamma_{AB})}{4G}$$
$$+ \left(1 - \frac{1}{n}\right)\frac{\mathcal{A}^{(n)}(\gamma_A) + \mathcal{A}^{(n)}(\gamma_B)}{4G}, \tag{171}$$

where $\mathcal{A}^{(n)}(\gamma_X)$ is related to the area of the cosmic brane homologous to subsystem $X$ as in eq. (13):

$$n^2 \frac{\partial}{\partial n}\left(\frac{n-1}{n}\mathcal{A}^{(n)}\right) = \text{Area}\,(\text{ cosmic brane }_n) .$$

Therefore, utilizing eq. (171) we obtain the generalized logarithmic Renyi negativity between the subsystems $A$ and $B$ from eq. (170) as

$$\mathcal{E}_{\text{gen}}^{(2n)}(A:B) = -n\frac{\mathcal{A}^{(1/2)}(\gamma_{AB})}{4G} - (n-1)\frac{\mathcal{A}^{(n)}(\gamma_A) + \mathcal{A}^{(n)}(\gamma_B)}{4G} + \mathcal{E}_{\text{eff}}^{(2n)}(A:B) , \tag{172}$$

taking the $n \to 1/2$ limit, the generalized logarithmic negativity is given by

$$\mathcal{E}_{\text{gen}}(A:B) = \frac{\mathcal{A}^{(1/2)}(\gamma_A) + \mathcal{A}^{(1/2)}(\gamma_B) - \mathcal{A}^{(1/2)}(\gamma_{AB})}{8G} + \mathcal{E}_{\text{eff}}(A:B). \tag{173}$$

In the following, we will assume that the quantum field theory on the hybrid manifold $\mathcal{M}$ itself has a holographic description as well. Therefore, following [93] the effective logarithmic entanglement negativity $\mathcal{E}_{\text{eff}}(A:B)$ in eq. (173) may be written as the order half effective mutual information between $A$ and $B$ as [8]

$$\mathcal{E}_{\text{eff}}(A:B) = \frac{1}{2}\mathrm{I}_{\text{eff}}^{1/2}(A:B), \tag{174}$$

where we have used arguments similar to subsection 3.1 to restructure the expression on the right hand side in terms of the Renyi entropies of order half. Therefore, eq. (173) may be put in the comprehensive form in eq. (30), namely

$$\mathcal{E}_{\text{gen}}(A:B) = \frac{1}{2}\left[S_{\text{gen}}^{(1/2)}(A) + S_{\text{gen}}^{(1/2)}(B) - S_{\text{gen}}^{(1/2)}(A \cup B)\right] \equiv \frac{1}{2}\mathrm{I}_{\text{gen}}^{(1/2)}(A:B). \tag{175}$$

---

[8]Note that in [93] the on-shell action on the quotient manifolds were evaluated for the restrictive class of fixed area states corresponding to a flat entanglement spectrum. As a result, all the Renyi entropies were taken to be equal and correspondingly the Renyi version of the areas (cf. eq. (173)) reduced to ordinary areas. In this manuscript, however, we lift such restrictions and consider Renyi areas as in eq. (13).

Finally, following a similar procedure as in [55], the entanglement negativity between subsystems $A$ and $B$ in a quantum field theory coupled to gravity is obtained through the extremization of the generalized negativity as

$$\mathcal{E}(A:B) = \min\left[\underset{\gamma_A,\gamma_B}{\text{ext}} \; \mathcal{E}_{\text{gen}}(A:B)\right]. \tag{176}$$

Note that in the above expression, the correct entanglement negativity is obtained by extremizing with respect to the positions of the individual homology surfaces $\gamma_A$ and $\gamma_B$.

## 7.1 Replica Wormhole Saddle

As described in [15,35] for the case of entanglement entropy, when the effective entropy of the quantum matter fields becomes comparable to the gravitational area term, the spacetime replica wormhole starts to dominate the gravitational path integral. In the case of the generalized Renyi negativity, we propose that when the contribution from the effective matter negativity becomes comparable to the area contributions in eq. (169), the generalized Renyi negativity gets a non-perturbative instanton-like contribution from a $\mathbb{Z}_n$-symmetric replica wormhole saddle. Figure 28 shows a schematic picture of the replica wormhole saddles computing the Renyi entropy (left) and the Renyi negativity (right) for an even replica index $n = 4$. While for the case of the Renyi entropy, the asymptotic boundary on each copy is the original manifold $\mathcal{M}_1$, it is the two-fold cover $\mathcal{M}_2^{A,B\,(\text{fixed})}$ that serves as the asymptotic boundaries on each copy in the case of the replica wormhole saddle computing the Renyi negativity.

As in the case of generalized Renyi entropy, the quotient manifold $\hat{\mathcal{M}}_{2n}^{A,B\,(\text{bulk, nsym})}$ has no conical singularities of the homology surfaces of $A$ and $B$ [15,35]. Instead, the bulk replica wormhole contains additional $\mathbb{Z}_n$ fixed points. These are the boundaries of the corresponding islands of $A$, $B$ and $A\cup B$. Therefore, the on-shell action for the quotient manifold of the replica wormhole saddle can be written analogously to eq. (171) as

$$I_{\text{grav}}\left(\mathcal{M}_2^{AB\,(\text{fixed})}, \gamma_{A_1}^{(n)}, \gamma_{B_2}^{(n)}\right) = 2\,I_{\text{grav}}[\mathcal{M}_1^{\text{bulk}}] + \frac{\mathcal{A}^{(1/2)}(\partial\,\text{Is}(AB))}{4G}$$
$$+ \left(1 - \frac{1}{n}\right)\frac{\mathcal{A}^{(n)}(\partial\,\text{Is}(A)) + \mathcal{A}^{(n)}(\partial\,\text{Is}(B))}{4G} \,. \tag{177}$$

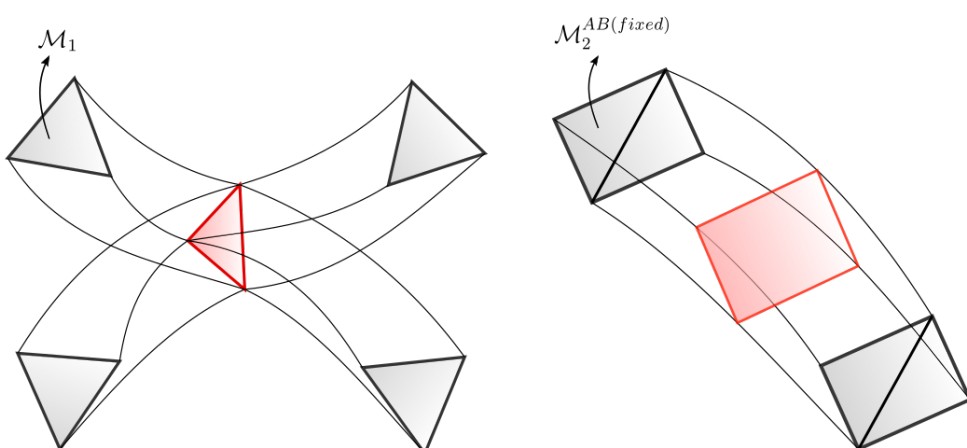

Figure 28: A cartoon picture of the replica wormhole saddle for the gravitational path integral for $\mathbb{Z}[\mathcal{M}_{2n}^{AB\,(\text{bulk})}]$ on the left and $\mathbb{Z}[\mathcal{M}_{2n}^{A,B\,(\text{bulk, nsym})}]$ on the right, is shown for $n = 2$.

Similar to the case of generalized Renyi entropy, the boundaries of these new island regions will have smooth twist operator insertions. As a result the effective Renyi negativity in eq. (170) will also get contributions from the quantum matter fields living on these island regions.

The genralized Renyi negativity between the subsystems $A$ and $B$ in eq. (169) may therefore be obtained utilizing eq. (177) as

$$\mathcal{E}_{\text{gen}}(A:B) = \frac{\mathcal{A}^{(1/2)}(\partial \text{Is}(A)) + \mathcal{A}^{(1/2)}(\partial \text{Is}(B)) - \mathcal{A}^{(1/2)}(\partial \text{Is}(AB))}{8G}$$
$$+ \mathcal{E}_{\text{eff}}(A \cup \text{Is}(A) : B \cup \text{Is}(B)). \qquad (178)$$

We will again assume that the quantum field theory on the hybrid manifold $\mathcal{M}$ itself has a holographic dual description and therefore the effective entanglement negativity of the quantum matter fields residing on the subsystems $A$ and $B$ as well as their corresponding entanglement islands may be written in terms of the order half effective mutual information between $A \cup \text{Is}(A)$ and $B \cup \text{Is}(B)$. Therefore the island formula (178) for the generalized entanglement negativity may equivalently be written as

$$\mathcal{E}_{\text{gen}}(A:B) = \frac{1}{2}\left[ S_{\text{gen}}^{(1/2)}(A) + S_{\text{gen}}^{(1/2)}(B) - S_{\text{gen}}^{(1/2)}(A \cup B) \right]$$
$$= \frac{1}{2} I_{gen}^{(1/2)}(A:B). \qquad (179)$$

Once again, employing the Engelhardt-Wall prescription [55] [9], we obtain the following formula for the entanglement negativity between two subsystems $A$ and $B$ in a quantum field theory coupled to gravity,

$$\mathcal{E}(A:B) = \min\left[ \underset{\partial \text{Is}(A), \partial \text{Is}(B)}{\text{ext}} \mathcal{E}_{\text{gen}}(A:B) \right]. \qquad (180)$$

However, as we have shown in the main body from pure geometric considerations, there is only one free parameter to extremize over in eq. (180), namely the island cross section for the entanglement negativity $Q'' \equiv \partial \text{Is}_{\mathcal{E}}(A) \cap \partial \text{Is}_{\mathcal{E}}(B)$. Therefore, the correct formula for the entanglement negativity between $A$ and $B$ is given by

$$\mathcal{E}(A:B) = \min\left[ \underset{Q''}{\text{ext}} \; \mathcal{E}_{\text{gen}}(A:B) \right]. \qquad (181)$$

This completes the derivation of the island formula for the entanglement negativity by considering the corresponding replica wormhole contributions based on techniques developed in [15, 35, 93]. Below we demonstrate the equivalence of the above obtained result from the replica wormhole saddle with our island proposal-I.

**Two adjacent intervals:**

Observe that eq.(179) and eq.(181) match exactly with our island proposal-I for the entanglement negativity of mixed state configuration involving two adjacent intervals in a quantum field theory coupled to gravity described by eq.(30).

**Two disjoint intervals:**

We will now demonstrate the equivalence between our conjecture in eq.(29) and the result in eq.(179) and eq.(181) which we derived from replica wormhole construction. To begin with

---

[9]Note that eq.(178) and eq.(179) clearly suggest that the Engelhardt-Wall like prescription for entanglement negativity would involve the extremization of the entire sum and not the individual generalized Renyi entropy terms.

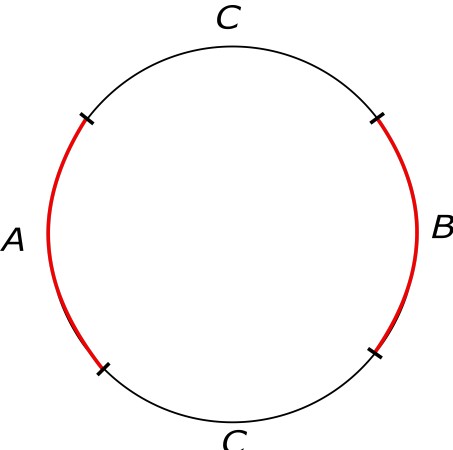

Figure 29: Schematic for the configuration of the disjoint intervals $A$ and $B$ separated by $C$ such that the full tripartite system $ABC$ is compact and is in a pure state .

let us consider the case of the disjoint intervals $A$ and $B$ separated by union of two disjoint intervals denoted as $C$ such that the full system $ABC$ is compact and is in a pure state[10] as depicted in fig.29. For a tripartite system $A \cup B \cup C$ which is in a pure quantum state, it is easy to show from quantum information that the following equality holds

$$\mathrm{I}(A : BC) = \mathrm{I}(A : B) + \mathrm{I}(A : C). \tag{182}$$

Note that as discussed earlier, for generic subregions $X$ and $Y$ in two dimensional holographic CFTs and for subsystems involving spherical entangling surfaces in higher dimensions, we have

$$\mathrm{I}^{(1/2)}(X : Y) = \chi_d \, \mathrm{I}(X : Y) \tag{183}$$

$$S^{(1/2)}(X) = \chi_d S_X. \tag{184}$$

Utilizing the above expressions, we can re-express eq.(182) by multiplying it on both sides by $\chi_d$ to obtain

$$\mathrm{I}^{(1/2)}(A : BC) = \mathrm{I}^{(1/2)}(A : B) + \mathrm{I}^{(1/2)}(A : C). \tag{185}$$

As explained earlier, the generalized Renyi entropies obey a relation similar to eq.(183) and hence, it is possible to generalize the above formula to subregions in a two dimensional QFTs coupled to gravity or for subsystems involving spherical entangling surfaces in higher dimensions, by considering the corresponding island contributions. Therefore, we have

$$\mathrm{I}^{(1/2)}_{gen}(A : BC) = \mathrm{I}^{(1/2)}_{gen}(A : B) + \mathrm{I}^{(1/2)}_{gen}(A : C). \tag{186}$$

Note that above formula could be re-expressed as follows

$$\frac{1}{2}\mathrm{I}^{(1/2)}_{gen}(A : B) = \frac{1}{2}\left[ \mathrm{I}^{(1/2)}_{gen}(A : BC) - \mathrm{I}^{(1/2)}_{gen}(A : C) \right]$$
$$= \frac{1}{2}\left[ S^{(1/2)}_{gen}(A \cup C) + S^{(1/2)}_{gen}(B \cup C) - S^{(1/2)}_{gen}(A \cup B \cup C) - S^{(1/2)}_{gen}(C) \right]. \tag{187}$$

---

[10]When the full system is non-compact and involves a mixed tripartite quantum state $\rho_{ABC}$, the gravitational path integral for the entanglement negativity and the corresponding replica wormhole contribution is more involved and possibly dominated by a different non-trivial replica symmetry breaking saddle. We leave this interesting problem for future investigation.

Alternatively one may obtain the above equation by utilizing the equalities of the entropies $S_{A \cup C} = S_B$, $S_{B \cup C} = S_A$, $S_{ABC} = 0$ and $S_C = S_{AB}$ for the tripartite pure state $\rho_{ABC}$.

Comparing the above result to eq.(179) which we derived utilizing the replica wormhole contribution leads to

$$\mathcal{E}_{\text{gen}}(A : B) = \frac{1}{2}\left[ S_{\text{gen}}^{(1/2)}(A \cup C) + S_{\text{gen}}^{(1/2)}(B \cup C) - S_{\text{gen}}^{(1/2)}(A \cup B \cup C) - S_{\text{gen}}^{(1/2)}(C) \right], \qquad (188)$$

$$\mathcal{E}(A : B) = \min\left[ \underset{Q''}{\text{extr}} \ \mathcal{E}_{\text{gen}}(A : B) \right]. \qquad (189)$$

The above expression matches exactly with our island proposal-I in eq.(29) for the entanglement negativity of the disjoint intervals in a QFT coupled to gravity. Hence, the computation above serves as a proof of our island proposal-I constructed by considering the corresponding replica wormhole contributions.

**Single interval in an infinite system:** In order to demonstrate the equivalence of our island proposal-I for the case of a single interval in an infinite system, to the replica wormhole result in eq.(179), we consider a tripartite pure state $A \cup B_1 \cup B_2$ with $\equiv A^c$, for which it is easy to show that

$$I(A : B_1 B_2) = I(A : B_1) + I(A : B_2).$$

This property was utilized in [92] to demonstrate the equivalence of the two alternative holographic proposals in [76] and [90] described earlier. As explained above, for subregions in two dimensional QFTs coupled to gravity and for systems involving the spherical entangling surfaces in higher dimensions, we can generalize the above expression to

$$I_{\text{gen}}^{(1/2)}(A : B_1 B_2) = I_{\text{gen}}^{(1/2)}(A : B_1) + I_{\text{gen}}^{(1/2)}(A : B_2) .$$

Finally utilizing eq. (175), we obtain the generalized entanglement negativity for a single interval in an infinite system as

$$\begin{aligned} \mathcal{E}_{\text{gen}}(A) &\equiv \frac{1}{2} I_{\text{gen}}^{(1/2)}(A : B_1 B_2) \\ &= \lim_{B_1 \cup B_2 \to A^c} \frac{1}{2}\left[ 2 S_{\text{gen}}^{(1/2)}(A) + S_{\text{gen}}^{(1/2)}(B_1) + S_{\text{gen}}^{(1/2)}(B_2) - S_{\text{gen}}^{(1/2)}(A \cup B_1) - S_{\text{gen}}^{(1/2)}(A \cup B_2) \right]. \end{aligned}$$

Upon extremization the above result obtained from the replica wormhole construction exactly matches with our island proposal-I for a single interval in an infinite system which was described in eq.(31). This concludes the proof of our island proposal-I for the entanglement negativity of all the configurations considered in the present article.

## 8 Summary and discussions

In this article we develop two alternative constructions for the island contributions to the entanglement negativity of various pure and mixed state configurations in quantum field theories coupled to semiclassical gravity. The first proposal involves a specific algebraic sum of the generalized Renyi entropies of order half. This is inspired by the holographic constructions described in [76, 79, 81]. The second proposal is motivated by the quantum corrected holographic formula for the entanglement negativity proposed in [90]. This involves an island construction for the entanglement negativity obtained through the sum of the area of a back-reacted cosmic brane spanning the EWCS, and the effective entanglement negativity of bulk

quantum matter fields. Following this, motivated by [91], we propose that the entanglement negativity of a bipartite system in a quantum field theory coupled to semiclassical gravity, is determined by extremizing the generalized Renyi reflected entropy of order half.

We applied our proposals to the case of JT gravity coupled to matter fields which are described by a $CFT_2$ with a large central charge. We computed the island contribution to the entanglement negativity for the pure and mixed state configurations involving disjoint, adjacent and single intervals in bath systems coupled to extremal black holes in JT gravity. The results from both the proposals match exactly for all the phases ( characterized by the size of the intervals ) of the configurations considered in this article. Furthermore, as discussed above in each case we determined the entanglement negativity from proposal-II using two different methods. Firstly, the entanglement negativity was computed through the extremization of the sum of the area of a back reacted cosmic brane and the effective entanglement negativity of bulk quantum matter fields, determined through the twist correlators in [72,73]. Following this, we obtained the entanglement negativity through the extremization of the generalized Renyi reflected entropy of order half. We demonstrated that the results from two methods match exactly, characterizing a consistency check of our proposal-II. We also showed that these results precisely match with the entanglement negativity determined from our proposal-I.

Note that the dynamical part of the entanglement negativity for the cases involving the disjoint and the adjacent interval configurations mentioned above, is proportional to that of the reflected entropy considered in [68]. We would like to emphasize that this is because of the spherical symmetry of the entangling surfaces involved, as they render the area of resulting backreacted extremal surfaces to be proportional to the non-backreacted area of corresponding extremal surfaces in the original geometry. However, for generic subsystems in higher dimensions, we expect that the reflected entropy and the entanglement negativity will have different behaviors.

Subsequently, we applied our island proposals to obtain the entanglement negativity of various pure and mixed state configurations involving disjoint, adjacent and single intervals in quantum system described by a bath coupled to an eternal black hole in JT gravity. In contrast to the extremal black hole case, where all the intervals were in the bath, the adjacent interval configuration considered in the non-extremal black hole scenario had one interval in the bath and the other on the JT brane. For the above described configurations we computed the island contribution to the entanglement negativity utilizing both of our proposals. We demonstrated that the entanglement negativity obtained using the generalized Renyi reflected entropy of order half, and that from the sum of the area of back reacted cosmic brane and the effective entanglement negativity matched exactly. Furthermore, these results matched precisely with the entanglement negativity obtained from our island proposal-I based on a combination of the generalized Renyi entropies of order half. Note that the reflected entropy was explored for some of the above configurations in [68,69]. For the case of disjoint intervals, we observed that once again the dynamical part of the entanglement negativity turns out to be proportional to that of the reflected entropy in [68] due to the spherical symmetry of the entangling surfaces involved. Furthermore, the reflected entropy for the case of the adjacent interval with one of the intervals in the bath and the other on the JT brane was explored in [69] for fermionic quantum matter fields utilizing the methods of [100]. Therefore, the results for reflected entropy derived in [69] for this case are different from our results.

Following this, we commented on a possible understanding of the above constructions in the double holography picture where one considers the quantum matter described by a holographic $CFT_d$ coupled to d-dimensional semiclassical gravity. Motivated by our island proposals we alluded to two alternative doubly holographic pictures for the entanglement negativity. As discussed above, our island proposal-I involved a specific algebraic sum of the generalized Renyi entropies of order half. In the context of double holography, the generalized Renyi en-

tropy of order half is given by the area of a backreacted extremal surface ( geodesic in AdS$_3$ ) anchored on the subsystem and extending in the dual bulk AdS$_{d+1}$ spacetime. Consequently, the double holographic picture of our proposal-I consisted of a specific combination of the areas of back reacted cosmic branes anchored on the subsystems/islands. On the other hand, the doubly holographic picture for our island proposal-II involved the minimal area of the backreacted EWCS in the dual bulk AdS$_{d+1}$ spacetime. The area of the backreacted cosmic brane is proportional to the area of its tensionless counterpart for a spherical entangling surface such that the backreaction effect is encoded in the proportionality constant. Finally, we provided a derivation of our island proposal-I for the pure and mixed state configurations considered by the computing the replica wormhole contribution to the partially transposed density matrix, in the gravitational path integral formulation through the techniques developed in [15, 35, 93].

Although the two proposals we have developed seem quite distinct, remarkably they lead to exactly the same results for all the cases examined here. It would be quite interesting to explore the reason for the equivalence of the two proposals in more details. The two alternative island proposals formulated in the present article are based on holographic constructions in [76, 79, 81] and [90, 91]. A possible mechanism to test the equivalence of the two proposals may involve a recently introduced measure termed the Markov gap examined in [106]. This measure is defined as the difference between the reflected entropy and the mutual information. The authors in [106] demonstrated that for a bipartite system in a holographic $CFT_2$, it is bounded from below by a $\mathcal{O}(\frac{1}{G_N})$ constant times the number of boundaries of the corresponding EWCS. Note that in the second proposal the entanglement negativity is related to the Renyi reflected entropy of order half whereas in the first proposal negativity is related to the Renyi mutual information of order half for compact systems. Furthermore, in a holographic $CFT_2$ and for subsystems involving spherical entangling surface in higher dimensions, the Renyi mutual information of order half and the Renyi reflected entropy of a given subsystem are proportional to the corresponding mutual information and the reflected entropy respectively. Hence, the two proposals give exactly the same answer when the Markov gap vanishes. In the cases we considered the results from the two proposals for entanglement negativity matched precisely. This is because the Markov gap in the configurations we examined can at most be a constant and hence the two proposals resulted in exactly the same functional form for the entanglement negativity. Therefore, it might be of crucial significance to further understand the Markov gap in various configurations to explore regimes where the two proposals might give different results. Furthermore, the two proposals have been examined in the language of tensor network in [90] and [93] utilizing which it might be possible to test their equivalence in various regimes. We hope to address these interesting issues in the near future.

We would like to emphasize that the entanglement negativity being a mixed state entanglement measure provides an insight into the structure of entanglement in the Hawking radiation. A more detailed study of our proposals into various other evaporating black hole scenarios may reveal deeper aspects of the Hawking radiation. It would be exciting to develop a possible proof of our island proposal-II for the entanglement negativity by considering an alternative replica symmetry braking saddle and the corresponding replica wormhole construction. It will be extremely fascinating to apply our proposals to spacetimes that are not asymptotically $AdS$ as done for entanglement entropy in [23, 24, 26]. Furthermore it has been recently shown in [35–38] that the island construction has significant implications to cosmology. It would be quite interesting to explore the implications of our proposals for the islands contributions to the entanglement negativity to the above scenarios.

# A    Renyi entropy of order half of the JT gravity

In this appendix we compute the Renyi entropy of order half of a thermal $CFT_1$ dual to a non-extremal JT black hole through two different techniques. In section A.1 we determine the Renyi entropy of order half by considering the purification of the thermal $CFT_1$ which is described by a thermofield double (TFD) dual to an eternal black hole in JT gravity. In section A.2 we obtain the same through the replica wormhole contribution to the corresponding gravitational path integral of the non-extremal JT black hole using the results of a recent article [105].

## A.1    Entanglement negativity of the TFD state dual to an eternal black hole in JT gravity

In this subsection we compute the entanglement negativity of the thermofield double (TFD) state dual to an eternal black hole in JT gravity. Note that since the TFD state is pure, the entanglement negativity is equal to Renyi entropy of order half of the thermal $CFT_1$ living on either the left or the right asymptotic boundary.

The TFD state is defined as

$$|\text{TFD}\rangle := \frac{1}{\sqrt{Z(\beta)}} \sum_k e^{-\frac{\beta E_k}{2}} |k\rangle |k\rangle. \tag{190}$$

The interesting property of the thermofield double is that the reduced density matrix of the left or the right subsystem is described by the thermal Gibbs state

$$\rho_L = \frac{e^{-\beta H}}{Z(\beta)}. \tag{191}$$

This in turn implies that the entanglement entropy of the left or the right subsystem is same as the thermal entropy of a Gibbs state i.e

$$S_{EE} = S_{th} = \left(1 - \beta \partial_\beta\right) \log Z(\beta). \tag{192}$$

It is clear from the above equation that we could obtain the entanglement entropy for the left or the right subsystem of a TFD state directly from the thermal partition function.

Similarly it was shown in [75] that the entanglement negativity of the bipartite system $LR$ is given by

$$\mathcal{E}(L:R) = \log \frac{Z\left(\frac{\beta}{2}\right)^2}{Z(\beta)} = \beta(F(\beta) - F(\beta/2)), \tag{193}$$

where $F$ denotes the free energy corresponding to the thermal partition function $Z(\beta)$ given by

$$F(\beta) = -\frac{1}{\beta} \log Z(\beta). \tag{194}$$

Note that, for a pure state the entanglement negativity should be same as the Renyi entropy of order half [72,73]. Since $LR$ together are in pure state described by the TFD,

$$\begin{aligned}
\mathcal{E}(L:R) = S_{EE}^{1/2}(L) &= S_{EE}^{1/2}(R) \\
&= \beta(F(\beta) - F(\beta/2)).
\end{aligned} \tag{195}$$

The free energy $F(\beta)$ depends on the partition function as described by eq.(194) and hence, analogous to the entanglement entropy, the entanglement negativity of the left or the right subsystem in the TFD state may also be obtained through the thermal partition function.

Now for the case of $JT$ gravity, the thermal partition function of the $CFT_1$ could be obtained by the dual bulk Euclidean classical action described by the Schwartzian as follows

$$Z[\beta] = \exp\left[S_0 + \frac{\phi_r}{8\pi G}\int du\, \text{Sch}(t,u)\right], \tag{196}$$

where, $S_0$ is the topological part and $\text{Sch}(t,u)$ is the Schwarzian derivative given by

$$\text{Sch}(t,u) = \frac{2t't''' - 3t''^2}{2t'^2}. \tag{197}$$

For the JT gravity

$$t(u) = \frac{\beta}{\pi}\tanh\left[\frac{\pi u}{\beta}\right]. \tag{198}$$

Substituting eq.(196) in eq.(192) leads to the following expression for the entanglement entropy

$$S(L) = \phi_0 + \frac{2\pi\phi_r}{\beta}, \tag{199}$$

where we are working in the units of $4G_N^{(2)} = 1$. On the other hand substituting eq.(196) in eq.(195) we obtain the following expression for the entanglement negativity of the TFD state

$$\mathcal{E}(L:R) = \phi_0 + \frac{3\pi\phi_r}{\beta} = S^{1/2}(L). \tag{200}$$

Now, note that the gravity dual of the Renyi entropy is related to the area of a backreacted cosmic brane spanning the RT surface in the bulk spacetime [96] as was described in eq.(14). For the present case the bulk spacetime corresponds to a two dimensional eternal black hole in JT gravity and hence the co-dimension two RT/HRT surface is a point. Therefore, from the above expression and eq.(14) we have

$$S^{1/2}(L) = \mathcal{A}^{(1/2)}(\gamma_L) = \phi_0 + \frac{3\pi\phi_r}{\beta}, \tag{201}$$

where, $\gamma_L$ is the point in the bulk where the backreacted RT surface is located. Observe by comparing eq.(199) and eq.(201) that the ratio of the dynamical part of the entanglement entropy of order one and that of order half is $3/2$. This number is precisely arising due to the backreaction $\mathcal{X}_2 = \frac{3}{2}$ defined in eq.(15). However, the topological term remains unaffected by the backreaction as expected.

## A.2 Renyi entropy of order $\frac{1}{2}$ through the replica wormhole construction

Recently in [105], the gravitation path integral for the replica wormhole saddle in JT gravity was performed to higher order in the replica parameter $n$ away from $n = 1$. The authors there obtained the following expression for the refined Renyi entropy

$$\tilde{S}^{(n)} = \sum_i \left[S_0 + \phi^{(n)}(\sigma_i)\right] + \tilde{S}^{(n)}_{\text{eff}},$$

$$\phi^{(n)}(\sigma_i) = -\frac{2\pi}{n\beta}\frac{\phi_r}{\tanh\left(\frac{2\pi\sigma_i}{n\beta}\right)}, \tag{202}$$

where $\sigma_i$ denote the positions of the corresponding quantum extremal surfaces, $\beta$ is the inverse temperature of the dilaton black hole and $\phi_r$ is the boundary value of the dilaton. In eq. (202) the effective refined Renyi entropy of the quantum matter is denoted by $\tilde{S}_{\text{eff}}^{(n)}$.

Following this, the Renyi entropy of order half for a subsystem $A \equiv [0, b]$ in a thermal $\text{CFT}_2$ bath coupled to a JT black hole of inverse temperature $\beta$, can be obtained as

$$
\begin{aligned}
S^{(n)} &= \frac{n}{n-1} \int_1^n d\hat{n} \, \frac{1}{\hat{n}^2} \, \tilde{S}^{(\hat{n})} \\
&= \frac{n}{n-1} \left[ S_0 \int_1^n d\hat{n} \, \frac{1}{\hat{n}^2} + \frac{2\pi\phi_r}{\beta} \int_1^n d\hat{n} \, \frac{1}{\hat{n}^3} \coth\left( \frac{2\pi a}{\hat{n}\beta} \right) \right] + S_{\text{eff}}^{(n)} \\
&= S_0 + \frac{n}{n-1} \frac{2\pi\phi_r}{\beta} \left( \frac{1}{2\alpha^2} \left[ \text{Li}_2\left( 1 - e^{\frac{2\alpha}{n}} \right) - \text{Li}_2\left( 1 - e^{2\alpha} \right) \right] + \frac{1-n^2}{2n^2} \right) + S_{\text{eff}}^{(n)},
\end{aligned} \tag{203}
$$

where $a$ describes the endpoint of the island region $I_A \equiv [-a, 0]$ corresponding to $A$, and we have defined $\alpha = 2\pi a/\beta$. In eq. (203), $\text{Li}_2(z)$ is the Spence's function, defined as

$$
\text{Li}_2(z) = - \int_0^z du \, \frac{\ln(1-u)}{u} \, .
$$

The effective Renyi entropy of the quantum matter fields $S_{\text{eff}}^{(n)}$ is obtained as

$$
S_{\text{eff}}^{(n)} = \frac{n}{n-1} \int_1^n d\hat{n} \, \frac{1}{\hat{n}^2} \, \tilde{S}_{\text{eff}}^{(\hat{n})} \, . \tag{204}
$$

Therefore, the Renyi entropy of order half for the subsystem $A \equiv [0, b]$ may be obtained through the analytic continuation $n \to \frac{1}{2}$ as

$$
S^{(1/2)} = S_0 - \frac{\phi_r}{a} \left( \frac{1}{2\alpha} \left[ \text{Li}_2\left( 1 - e^{4\alpha} \right) - \text{Li}_2\left( 1 - e^{2\alpha} \right) \right] + \frac{3\alpha}{2} \right) + S_{\text{eff}}^{(1/2)} \, . \tag{205}
$$

In the high temperature limit $\beta \to 0$, the area term in the above equation has the following expression

$$
\mathcal{A}^{(1/2)} = S_0 + \frac{3\pi\phi_r}{\beta} \, . \tag{206}
$$

Remarkably, this is the same expression obtained through an analysis of the entanglement negativity of the TFD state in appendix A.1.

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
