# Peer review of "Islands for Entanglement Negativity"

_SciPost Physics, doi:SciPost Phys. 12, 003 (2022)_

## Round 1 · Referee Report · Anonymous (Referee 1) · 2021-7-28

Report

The authors of "Islands for Entanglement Negativity" employ two complementary proposals for studying contributions to entanglement negativity as a result from the island prescription. They perform this for pure and mixed state configurations of a large c CFT coupled to a semi-classical eternal AdS_{2} black hole, described using semi-classical JT gravity. Proposal-I is constructed from a combination of generalized Renyi entropies. Proposal-II is constructed using the entanglement wedge cross section. They find the proposals agree and provide a clear brane world picture of the on/off-set of the quantum extremal surface and its contribution to the entanglement. This work provides good opportunities for new research.

Furthermore, sufficient details are provided, also reference wise, and both abstract and introduction reflect the content of the paper.

I have few remarks:

  • In the conclusion you refer to studying evaporating black holes. In terms of the setup you sketch in, e.g. the Penrose diagram in Figure 22, one will obtain a one sided black hole diagram with a black hole with finite lifetime. The time dependence will enter explicitly in the dilaton as well now as you're in the Unruh state. Where do you expect this will become difficult for you? In other words: what (if anything) is stopping you from doing this now?

  • I'd like to point out https://arxiv.org/abs/2007.15999 where using a brane world perspective semi-classical corrections to the eternal BTZ black hole are computed. This would provide a nice context to test proposal I and II against each other.

  • As you are looking at pure and mixed states I am wondering what you expect as an outcome for your proposals when applied to semi-classical two-dimensional de Sitter in the Bunch-Davies vacuum and different sides of cosmological horizon. Could you comment on that?

  • validity: -
  • significance: -
  • originality: -
  • clarity: -
  • formatting: -
  • grammar: -

Author:  Vinay Malvimat  on 2021-10-09  [id 1828]

(in reply to Report 1 on 2021-07-28)

We would like to thank the referee for the interesting remarks. Below is our response to each of the remarks.

  1. The single sided evaporating JT black hole that the referee is suggesting is presumably constructed by inserting the EOW brane behind one of the horizons of the eternal black hole. A toy version of such a model was examined in ref [14] of our revised manuscript where the authors provided a derivation of the island formula for entanglement entropy by considering such a EOW brane behind the horizon. However,a complete bulk computation of the entanglement entropy in the two dimensional effective theory from the island formula would require exactly solving the dilaton equation of motion in this newly constructed space time which is not understood in its full technicality. Hence, determining the entanglement entropy and negativity of subregions in bath/radiation coupled to an one sided evaporating JT black hole currently remains a highly non-trivial open question.
  2. We would like to thank the referee for pointing out this interesting direction. This is indeed a nice context to test the two proposals which we hope to address in the near future.
  3. This is an interesting direction suggested by the referee. However, this requires a careful technical analysis of the entanglement islands for negativity in de-Sitter space time. It would be quite interesting to examine such islands along the lines of ref [41,42,49] of our revised manuscript. We would like to explore this interesting direction in our future investigations.

---

## Round 1 · Referee Report · Anonymous (Referee 2) · 2021-9-21

Strengths

1 - Detailed, with lots of cases studied
2 - Pedagogical

Weaknesses

1 - No insight has been provided for why the two different proposals are giving the same answer. If this insight is in one of the older papers, it still deserved to be written out.

Report

The authors have presented two seemingly different proposals for computing the so-called entanglement negativity, a quantity that is a measure of quantum entanglement in mixed states. The authors have explained a lot of previous results, and have included a lot of details in many cases for which they computed the entanglement negativity using both their proposals.

Since the presence of entanglement islands for vN entropy is an exciting recent development, it makes sense to study island contributions to other entanglement measures. This paper is doing so.

I recommend publication after the minor changes below.

Requested changes

1 - Provide some plausible mechanism, perhaps to be proved in future, for why the two different proposals are giving the same answer.

2- In many places the italicization of multi-letter subscripts or superscripts is not consistent. The abbreviations gen, eff, or Is appear both in straight face and in italics. It would be better to adopt one consistent notation.

  • validity: top
  • significance: good
  • originality: good
  • clarity: top
  • formatting: excellent
  • grammar: perfect

Author:  Vinay Malvimat  on 2021-10-09  [id 1829]

(in reply to Report 2 on 2021-09-21)

We would like to thank the referee for the interesting comments. The two changes requested by the referee which are listed below have been incorporated in the revised version of the manuscript.

  1. We would like to emphasize that our two island proposals are based on two alternative holographic constructions for the entanglement negativity in ref [79,81] and ref [90,91] of our revised manuscript. The first holographic proposal has received substantial evidence from a recent article given in ref [93] of our revised manuscript, where the authors utilized a novel replica symmetry breaking saddle which dominates the corresponding gravitational path integral. This has been reviewed in section 7 of our manuscript. Furthermore, in the same section, we have also extended their construction to include replica wormhole contributions which serves as a possible derivation of our island proposal. On the other hand, the second island proposal is based on the holographic construction in ref [90,91] of our revised manuscript. In the context of $AdS_3/CFT_2$, the authors in ref [91] of our revised manuscript demonstrated that their proposal involving the equivalence between entanglement negativity and half the Renyi reflected entropy of order half, may be substantiated by comparing the dominating conformal blocks in various channels of the corresponding correlators.
    A possible mechanism to test the equivalence of the two proposals may involve a recently introduced measure termed the Markov gap examined in ref [106] of our revised manuscript. This measure is defined as the difference between the reflected entropy and the mutual information. The authors in ref [106] demonstrated that for a holographic $CFT_2$ it is bounded from below by a $\mathcal{O}(\frac{1}{G_N})$ constant times the number of boundaries of EWCS. Note that in the second proposal the entanglement negativity is related to the Renyi reflected entropy of order half whereas in the first proposal negativity is related to the Renyi mutual information of order half for compact systems. Furthermore, in a holographic $CFT_2$ and for subsystems involving spherical entangling surfaces in higher dimensions, the Renyi mutual information of order half and the Renyi reflected entropy of a given subsystem are proportional to the corresponding mutual information and the reflected entropy respectively . Hence, the two proposals give exactly the same answer when the Markov gap vanishes. In the cases we considered the results from the two proposals for entanglement negativity matched precisely. This is because the Markov gap for the configurations we examined can at most be a constant and hence the two proposals resulted in exactly the same functional form for the entanglement negativity. Therefore, it might be of crucial significance to further understand the Markov gap in various configurations to explore regimes where the two proposals might give different results. Furthermore, the two proposals have been examined in the language of tensor network in ref [90] and [93] utilizing which it might be possible to test their equivalence in various regimes. We hope to address these interesting issues in the near future. We have added the above discussion in the summary section of our revised manuscript.
  2. We thank the referee for pointing this out. All the abbreviations are now in straight face and the notation is now consistent in the revised manuscript.

---

## Round 2 · Referee Report · Anonymous · 2021-10-11

Report
The authors have addressed the previous concerns and the manuscript can now be published.
Anonymous on 2021-10-24 [id 1873]
As a referee I am satisfied and am of the opinion the manuscript is ready for publication.

---

## Round 2 · Author Response

List of changes
We have provided the complete list of changes made in the authors' response section.

You are currently on this page

---

## Round 2 · List of Changes

We have provided the complete list of changes made in the authors' response section.

You are currently on this page

---

## Editorial Decision

published